# The intrinsically disordered regions of organellophagy receptors are interchangeable and control organelle fragmentation, ER-phagy and mitophagy flux

Mikhail Rudinskiy [1,2], Carmela Galli[1], Andrea Raimondi[1] & Maurizio Molinari [1,3] ✉

Organellophagy receptors control the generation and delivery of portions of their homing organelle to acidic degradative compartments to recycle nutrients, remove toxic or aged macromolecules and remodel the organelle upon physiologic or pathologic cues. How they operate is not understood. Here we show that organellophagy receptors are composed of a membrane-tethering module that controls organellar and suborganellar distribution and by a cytoplasmic intrinsically disordered region (IDR) with net cumulative negative charge that controls organelle fragmentation and displays an LC3-interacting region (LIR). The LIR is required for lysosomal delivery but is dispensable for organelle fragmentation. Endoplasmic reticulum (ER)-phagy receptors' IDRs trigger DRP1-assisted mitochondrial fragmentation and mitophagy when transplanted at the outer mitochondrial membrane. Mitophagy receptors' IDRs trigger ER fragmentation and ER-phagy when transplanted at the ER membrane. This offers an interesting example of function conservation on sequence divergency. Our results imply the possibility to control the integrity and activity of intracellular organelles by surface expression of organelle-targeted chimeras composed of an organelle-targeting module and an IDR module with net cumulative negative charge that, if it contains a LIR, eventually tags the organelle portions for lysosomal clearance.

Nutrient restriction, pathologic and physiologic cues including accumulation of aged or toxic material, of misfolded proteins, pathogen attack and cell maturation or differentiation may activate lysosomal clearance of parts of intracellular organelles such as the endoplasmic reticulum (ER), the mitochondria and the Golgi complex[1–6]. These parts are physically separated from the bulk of the organelle, which must be preserved, and are delivered within degradative acidic organelles. ER-phagy, mitophagy and Golgiphagy (here collectively mentioned

as organellophagy) are driven upon activation of membrane-bound organellophagy receptors (for example, the ER-phagy receptors FAM134B[7], SEC62 (refs. 8,9), TEX264 (refs. 10,11), CCPG1 (ref. 12) and others (Fig. 1a); the mitophagy receptors FUNDC1 (refs. 13,14), BNIP3, BNIP3L/NIX[15,16] and others (Fig. 1b); and the Golgiphagy receptors YIPF3 and YIPF4 (refs. 17,18) (Fig. 1c)). All membrane-bound organellophagy receptors display a cytoplasmic intrinsically disordered region (IDR) module (Fig. 1a–c, green) that engages lipidated

[1]Institute for Research in Biomedicine, Università della Svizzera italiana, Bellinzona, Switzerland. [2]Department of Biology, Swiss Federal Institute of Technology, Zurich, Switzerland. [3]School of Life Sciences, École Polytechnique Fédérale de Lausanne, Lausanne, Switzerland. ✉e-mail: maurizio.molinari@irb.usi.ch

**Fig. 1 | Membrane-bound organellophagy receptors. a**, Schematics of the topology of ER-phagy receptors in mammalian (human, left) and yeast (*S. cerevisiae*, right) cells. RHD is shown in light blue, the blue rectangles show the conventional transmembrane domain, the green line shows the IDR and the LIR is shown in grey. **b**, Same as **a** for mitophagy receptors. IMS, intramembrane space.

**c**, Same as **a** for mammalian Golgiphagy receptors. **d**, The length of IDR modules in amino acids. AA, amino acid. **e**, The net cumulative charge of cytoplasmic IDR modules at physiologic pH calculated using https://www.biosynth.com/peptide-calculator.

LC3/GABARAP proteins via short consensus LC3-interacting region (LIR) motifs (Fig. 1a–c, grey boxes). The activation of organellophagy receptors has two major consequences: (1) the fragmentation of the homing organelle to separate portions to be degraded and (2) the delivery of the organelle fragments within acidic degradative organelles via macro-autophagy, micro-autophagy or LC3-dependent transport pathways[1,2,6,19–23]. While it has been clearly established that delivery of organelle portions within degradative compartments requires the engagement of LC3 proteins via cytoplasmic organellophagy receptors' LIRs, how organelles are portioned is unclear. For ER-phagy, a crucial role has been ascribed to the membrane remodelling function of the reticulon homology domains (RHD) that tether members of the FAM134 family at the ER membrane[7,23–28]. However, membrane remodelling is not a conserved trait of membrane-tethering modules of organellophagy receptors. In fact, other ER-phagy, mitophagy and Golgiphagy receptors[17,18] are anchored via conventional multi- or single-spanning transmembrane domains (Fig. 1a–c) that are not expected to promote membrane remodelling per se. Thus, a model assuming that the membrane-tethering modules of organellophagy receptors determine or give essential contribution to organelle fragmentation[23] cannot be

generalized. To clarify these issues, we investigated the consequences on ER and mitochondrial integrity of the expression at their limiting membranes of full-length, membrane-tethering or IDR modules of the ER-phagy receptors FAM134B, SEC62 and TEX264 and the mitophagy receptor FUNDC1.

Our study shows that the membrane-anchoring domains of organellophagy receptors can be replaced by exogenous membrane anchors with no impact on organelle fragmentation and lysosomal delivery. This implies, for example, that the membrane remodelling activity of FAM134B is dispensable for execution of ER-phagy. We report that ER and mitochondrial fragmentation is controlled by the cytoplasmic IDR modules of the organellophagy receptors. Consistent with the concept of functional conservation on sequence divergency, the IDR modules of organellophagy receptors have different sequences but maintain conserved traits. Among them, we identify a net cumulative negative charge at physiologic pH, a length above the 47 residues and a minimal distance of the LIR from the membrane of 24 residues. Conservation of these and possibly other features that remain to be characterized make them interchangeable. Thus, IDR modules of ER-phagy receptors expressed at the outer mitochondrial membrane

(OMM) trigger mitochondrial fragmentation and mitophagy; IDR modules of mitophagy receptors expressed at the ER membrane trigger ER fragmentation and ER-phagy.

## Results

### Membrane-anchoring and cytosolic IDR modules

The ER, the mitochondria and the Golgi complex display a large array of autophagy receptors at their limiting membrane[1,4,21,29] (Fig. 1a–c). Organellophagy receptors are composed of a membrane-tethering module that spans the organelle membrane once or multiple times (Fig. 1a–c, blue boxes) and a cytoplasmic IDR module (Fig. 1a–c, green lines) that contains the LIR domain engaging LC3/GABARAP proteins (Fig. 1a–c, grey boxes). Mammalian FAM134 family members and RTN3L are anchored at the ER membrane with RHDs that only partially span the lipid bilayer, with intrinsic membrane remodelling activity (shown in curved regions of the ER membrane in Fig. 1a). All other organellophagy receptors are anchored at membranes via conventional transmembrane domains (Fig. 1a–c).

We notice that the IDR modules of mammals and yeast ER-phagy, mitophagy and Golgiphagy receptors have a length between 47 and 250 residues (Fig. 1d) and include a single LIR (Fig. 1a–c, grey boxes). The LIR of the mitophagy receptor FUNDC1 is placed at 24 residue distance from the last residue of the transmembrane anchor. All other organellophagy receptors' LIRs are characterized by longer distances from the membrane of the homing organelles. The IDR module of RTN3L, a protein involved in endosome maturation[30], spans over 800 residues and contains multiple LIRs[1,4,5,20,21,31–33].

The IDR modules of organellophagy receptors have diverging sequences but are all characterized by net cumulative negative charge at physiologic pH (for example, −19.95 for SEC62$_{IDR}$, −33.94 for FAM134B$_{IDR}$, −13.96 for TEX264$_{IDR}$, −3.93 for FUNDC$_{IDR}$ and −8 and −11.11 for YIPF3$_{IDR}$ and YIPF4$_{IDR}$, respectively; Fig. 1e and https://www.biosynth.com/peptide-calculator). IDRs are often present in the proteome and a survey of the literature reveals that their functions include the processing of regulatory cues, molecular communication[34] and, notably, sensing and driving membrane curvature[35–37]. For their role in organellophagy, nothing is known beyond a putative role in ER-phagy as spacers to bridge the distance between the ER membrane in rough ER subdomains covered by ribosomes and LC3 lipidated at the membrane of phagophores (for macro-autophagic pathways) or of endolysosomes (EL, for micro-autophagy or LC3-mediated delivery pathways)[11].

### Expression of ER-phagy receptors triggers ER-phagy

ER-phagy is activated by pleiotropic (for example, nutrient restriction) and ER-centric cues (for example, luminal accumulation of misfolded proteins, ER stress and ribosome stalling)[1,20]. These cues induce expression of individual or multiple ER-phagy receptors and/or increase their local concentration and activity upon derepression, post-translational modifications and/or formation of hetero- or homo-meric clusters[8,12,24,25,27,38–55]. The ectopic expression of ER-phagy receptors induces ER-phagy bypassing the need for activating exogenous or intrinsic cues and has been used to characterize their functions, their interactors and their regulation (for example, FAM134B[7,53,56], SEC62 (refs. 8,9,57) and TEX264 (refs. 10,11,54)). Here, we first express Halo-tagged[58–60] versions of FAM134B, SEC62 and TEX264 (Fig. 2, left) to compare their capacity to induce ER-phagy. Please note that in all figures, the 4 nm × 3 nm GFP (Fig. 3, green circles) or HaloTag barrels (Fig. 2, green circles)[61] are drawn in approximate scale with the IDRs, where each amino acid contributes to the length of the unstructured IDR for an average of 0.38 nm (ref. 62). ER-phagy is monitored by confocal laser scanning microscopy (CLSM) in mouse embryonic fibroblasts (MEF) treated with Bafilomycin A1 (BafA1) to inhibit lysosomal hydrolases and preserve material (in this case, calnexin (CNX)-positive ER portions) delivered in the lumen of degradative LAMP1-positive EL[63].

Expression of FAM134B, SEC62 or TEX264 increases delivery of ER portions within LAMP1-positive EL (Fig. 2c,g,k) above the constitutive levels observed in mock-transfected cells (Fig. 2a). The variations in ER-phagy activity (that is, the accumulation of CNX-positive ER portions within LAMP1-positive EL upon inactivation of lysosomal hydrolases) are quantified with the deep learning tool LysoQuant developed in our lab[64] (Fig. 2e,i,m). The expression of the membrane-anchoring modules of the ER-phagy receptors (FAM134B$_{RHD}$ (Fig. 2b,e), SEC62$_{TMD}$ (Fig. 2f,i) or TEX264$_{TMD}$ (Fig. 2j,m)) does not trigger ER delivery within EL. Likewise, the expression of versions of the full-length ER-phagy receptors carrying mutations that prevent engagement of LC3 proteins (-$_{453}$DDFELLD$_{458}$- to -AAAAAAA- amino acid substitution for FAM134B$_{LIR}$ (ref. 7) (Fig. 2d,e), -$_{363}$FEMI$_{366}$- to -AAAA- for SEC62$_{LIR}$ (ref. 8) (Fig. 2h,i) and -$_{273}$FEEL$_{276}$- to -AAAA- for TEX264$_{LIR}$ (refs. 10,11) (Fig. 2l,m)) does not trigger ER delivery within EL. To confirm the activation of an autophagic flux also in cells, where the lysosomal activity has not been inhibited (that is, in absence of BafA1 exposure), we made use of the HaloTag assay, which has been developed in our lab[59,60,65–67]. Briefly, MEF expressing the ER-phagy reporter HT$_{17}$, a chimeric polypeptide of 37 kDa composed of a HaloTag tethered to the ER membrane with a 17-residue transmembrane domain[68] (Fig. 2n, second panel from the top, lanes 1–11) were mock-transfected (Fig. 2n, lane 1) or were transfected with the same modules analysed in Fig. 2b–m tagged with a monomeric superfolder (sf)GFP moiety[69]. Lysosomal delivery of ER portions and the subsequent hydrolytic processing of the HT$_{17}$ reporter generates a protease-resistant Halo fragment of about 33 kDa, whose fluorescence can be directly monitored in gel (Fig. 2n, third panel from the top)[59,60,65–67]. The HaloTag assay confirms the imaging data (Fig. 2a–m). Thus, expression of sfGFP-tagged FAM134B, SEC62 and TEX264 (Fig. 2n, top panel) triggers ER-phagy as shown by the generation of the Halo fragment (Fig. 2n, third panel from the top, lanes 3, 6 and 9), which requires hydrolytic lysosomal activity (that is, the cells exposure to BafA1 inhibits the generation of the Halo fragment, lane 11). The expression of FAM134B$_{RHD}$ (Fig. 2n, lane 2), SEC62$_{TMD}$ (lane 5), TEX264$_{TMD}$ (lane 8) or the expression of FAM134B$_{LIR}$ (Fig. 2n, lane 4), SEC62$_{LIR}$ (lane 7) and TEX264$_{LIR}$ (lane 10)) does not trigger ER-phagy as testified by the absence of the fluorescent Halo fragment.

Notably, as already shown by others in silico[26] and double checked here *in cellula* by room temperature-transmission electron microscopy (RT-TEM) (Extended Data Fig. 1a–e), the membrane-anchoring RHD module of FAM134B has the intrinsic property of remodelling the ER membrane. However, the compartment remains connected by thin tubules (Fig. 4f,g, fluorescence recovery after photobleaching (FRAP) experiments, and Extended Data Fig. 1c, red arrowheads) as it has been observed upon overexpression of members of the conserved family of ER remodelling proteins that generate high ER membrane curvature such as reticulons or REEP proteins[30,70–79]. The membrane-anchoring modules of SEC62 and TEX264 do not remodel the ER membrane as shown by the ER ultrastructure in mock-transfected cells (Extended Data Fig. 1b), which remains unchanged in cells expressing SEC62$_{TMD}$ and TEX264$_{TMD}$ (Extended Data Fig. 1d,e). Altogether, these data confirm that the expression of ER-phagy receptors competent for LC3 engagement at the ER membrane induces ER-phagy bypassing the needs of external cues. Importantly, the induction of ER-phagy occurs independent of the capacity of the membrane-anchoring modules of the receptors to remodel the ER membrane.

As a separate note, individual ER-phagy receptors are located in and control lysosomal clearance of distinct ER subdomains (FAM134B for ER tubuli, SEC62 and TEX264 for ER sheets and outer nuclear membrane[1,8–11,57,80,81]). Notably, the distinct suborganellar localization is also observed upon cellular expression of the membrane-anchoring domains of the ER-phagy receptors lacking the cytoplasmic IDR modules. For example, the endogenous interactome of FAM134B$_{RHD}$ reveals its distribution in subdomains hosting proteins of the reticulon and REEP families (as reported for the full-length receptor[56])

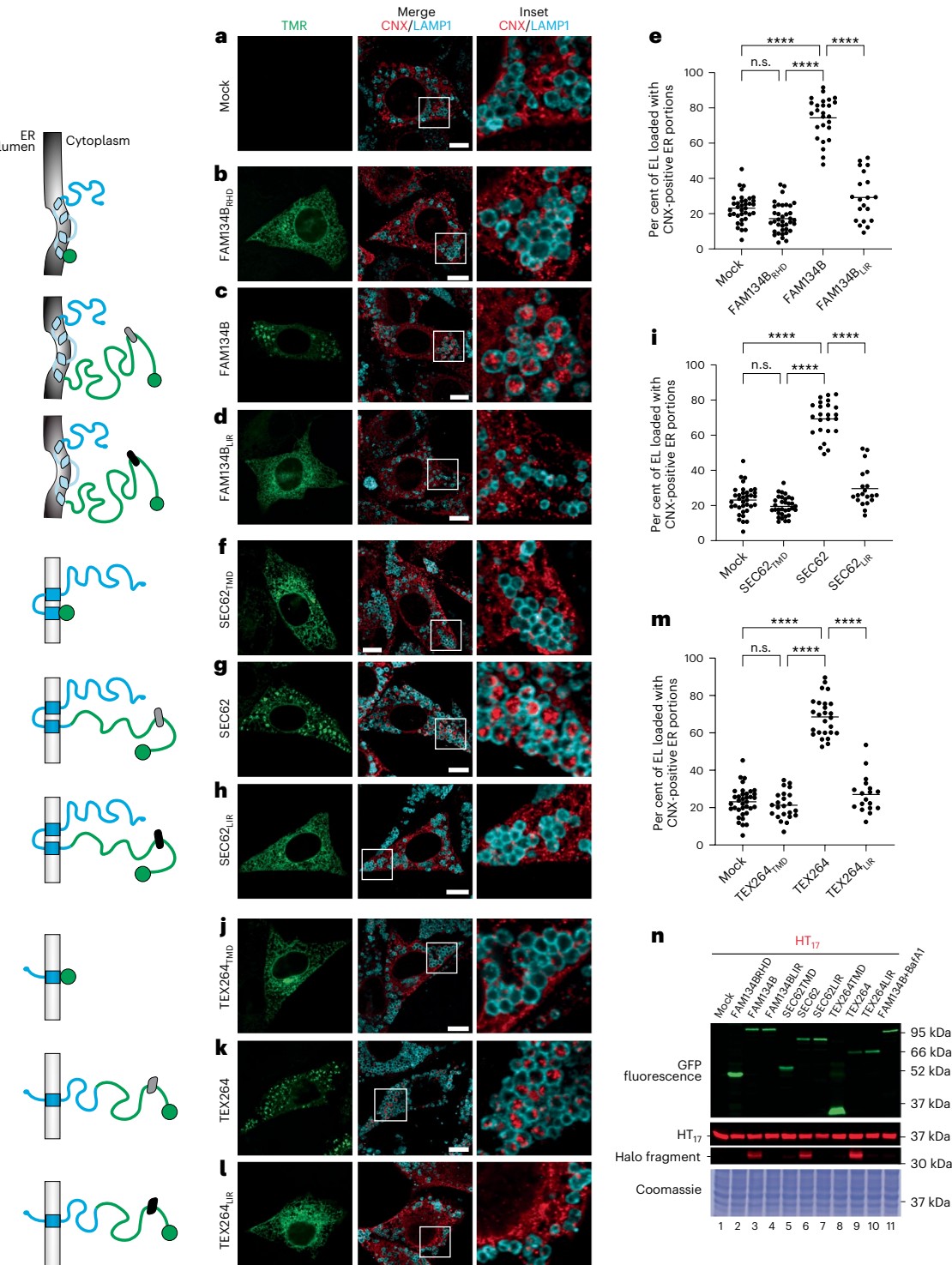

**Fig. 2 | CLSM analyses of ER-phagy induction. a**, Representative CLSM images of delivery of CNX-positive ER portions (red) to LAMP1-positive ELs (cyan) in mock-transfected MEF treated with 50 nM BafA1 and 100 nM TMR. Scale bar, 10 μm. The inset shows a 4× magnification of the merge image. **b**, Same as **a** in cells expressing FAM134B$_{RHD}$-HALO (green, TMR). **c**, Same as **b** in cells expressing FAM134B-HALO. **d**, Same as **c** for FAM134B$_{LIR}$-HALO. **e**, LysoQuant quantification of the percentage of CNX-loaded LAMP1-positive EL in **a**–**d** (mock (n = 30 cells), FAM134B$_{RHD}$-HALO (n = 32 cells), FAM134B-HALO (n = 26 cells), FAM134B$_{LIR}$-HALO (n = 20 cells)). N = 3 biological replicates. **f**–**h**, Same as **b**–**d** in cells expressing SEC62$_{TMD}$-HALO (n = 27 cells) (**f**), SEC62-HALO (n = 24 cells) (**g**) and SEC62$_{LIR}$-HALO (n = 20 cells) (**h**). **i**, LysoQuant quantification of the percentage of CNX-loaded LAMP1-positive EL in **f**–**h**. N = 3 biological replicates. Mock as in **e**. **j**–**l**, Same as **f**–**i** in cells expressing TEX264$_{TMD}$-HALO (n = 13 cells) (**j**), TEX264-HALO (n = 27 cells) (**k**) and TEX264$_{LIR}$-HALO (n = 18 cells) (**l**). **m**, LysoQuant quantification of the percentage of CNX-loaded LAMP1-positive EL in **j**–**l**. N = 3 biological replicates. Mock as in **e**. A one-way ANOVA with Turkey's multiple comparisons test was performed, with F = 190.1, 193.6 and 168 for **e**, **i** and **m**, respectively. Adjusted P value: ****P < 0.0001, n.s., not significant (P = 0.0643, 0.3035 and 0.8759 for n.s. in **e**, **i** and **m**, respectively). The mean bar is shown. **n**, In gel native GFP fluorescence (top) and HaloTag assay in cells expressing the ER-phagy reporter HT$_{17}$ (second from top) of constructs from **a**–**l** showing Halo fragment formation (high contrast) upon expression of LC3-binding-competent ER-phagy receptors FAM134B (lane 3), SEC62 (lane 6) and TEX264 (lane 9). N = 1 biological replicate. Bottom: the Coomassie loading control.

(Extended Data Fig. 1f,g). The interactions of the SEC62$_{TMD}$ and of the TEX264$_{TMD}$ reveal their localization in ER sheets also containing subunits of the translocation machinery including the oligosaccharyltransferase complex and chaperones assisting protein folding (as reported for the full-length receptors[8]) (Extended Data Fig. 1f,g).

## Expression of ER-phagy receptors IDR modules elicits ER-phagy

The finding that ER-phagy receptors expression induces ER-phagy (that is, ER fragmentation and delivery of ER portions within degradative LAMP1-positive organelles), independent of the capacity of their membrane-tethering modules to remodel the ER membrane, led us to verify the dispensability of the FAM134B, SEC62 and TEX264 membrane anchors to launch the catabolic programme. To verify this, we expressed in MEF the cytoplasmic IDR modules FAM134B$_{IDR}$, SEC62$_{IDR}$ or TEX264$_{IDR}$ anchored at the ER membrane with a single-spanning transmembrane domain of 17 residues (T$_{17}$)[68]. The chimeras were tagged with a luminal sfGFP (Fig. 3, schematics, and Extended Data Fig. 2). Induction of ER-phagy was assessed, as above, both with the imaging and with the HaloTag assays. Analyses by CLSM (Fig. 3a–g and Extended Data Fig. 2a–g for entire cells) and LysoQuant quantifications (Fig. 3h) show the enhanced lysosomal delivery of CNX-positive ER portions in cells expressing the three IDR modules (Fig. 3b,h for GT$_{17}$-FAM134B$_{IDR}$, Fig. 3c,h for GT$_{17}$-SEC62$_{IDR}$ and Fig. 3d,h for GT$_{17}$-TEX-264$_{IDR}$ and Extended Data Fig. 2b–d) compared with cells that express the ER membrane targeted sfGFP that does not display a cytoplasmic IDR (GT$_{17}$) (Fig. 3a,h and Extended Data Fig. 2a). The inactivation of the LC3-binding function of the three IDR modules abolishes delivery of ER portions within the LAMP1-positive EL (Fig. 3e–h and Extended Data Fig. 2e–g).

The dispensability of the ER-phagy receptors membrane-tethering modules for ER-phagy induction was confirmed with the HaloTag assay mentioned above that reports on the generation of the Halo fragment (Fig. 3i, third from the top) in cells with functional EL that express the membrane-bound, sfGFP-tagged IDR modules of SEC62 (Fig. 3i, lane 4), FAM134 (Fig. 3i, lane 6) and TEX264 (Fig. 3i, lane 8). The fragment is not produced in cells that express a membrane-tethered GT$_{17}$ control protein lacking cytoplasmic IDR (Fig. 3i, third from the top, lane 2) or the IDR modules that cannot engage LC3 (lanes 5, 7 and 9, respectively).

Finally, the competence of membrane-tethered IDR modules to fragment the ER and to deliver ER portions within EL was also monitored with anti-GFP immunoelectron microscopy (IEM) in MEF treated with BafA1. The gold particles reveal the localization of the GFP moiety of the membrane-tethered GT$_{17}$-SEC62$_{IDR}$ and of the GT$_{17}$-SEC62$_{IDRLIR}$ chimeras in the lumen of the ER and of ER portions in the cytoplasm (Fig. 3j,k, red arrowheads). Notably, the cells that express the IDR module that can engage LC3 show GFP-positive ER portions also within the lumen of EL (Fig. 3j, blue arrowheads), whereas the cells that express the IDR module that cannot engage LC3 do not show GFP-positive ER portions within the lumen of EL (Fig. 3k). This confirms that LC3 engagement is required for lysosomal delivery.

## Monitoring IDR modules-induced ER fragmentation

The detection of ER portions within LAMP1-positive EL in cells expressing full-length ER-phagy receptors (Fig. 2) or their IDR modules anchored at the ER membrane with a dissimilar transmembrane domain (Fig. 3 and Extended Data Fig. 2) indirectly reports on ER fragmentation, which is a prerequisite for ER portions being delivered within the degradative compartments. Direct monitoring of ER fragmentation (that is, the visualization of ER portions in the cytoplasm) is challenging due to the ultrastructure of the compartment and to the fact that the portions are delivered within EL for clearance. To overcome this hurdle and directly monitor the ER fragmentation function of ER-phagy receptor's IDR modules, the experiments shown in Fig. 3 were repeated in MEF lacking ATG7 (Fig. 4a), where the absence of LC3 lipidation[82] inhibits the delivery of ER portions within the LAMP1-positive degradative organelles[7–11], thus preserving them in the cytoplasm.

Analyses by RT-TEM of the ER ultrastructure in cells mock transfected (Fig. 4b), expressing membrane-bound sfGFP (GT$_{17}$, Fig. 4c) or membrane-bound sfGFP displaying the IDR modules of SEC62 that can (GT$_{17}$-SEC62$_{IDR}$, Fig. 4d) or cannot engage LC3 proteins (GT$_{17}$-SEC62$_{IDRLIR}$, Fig. 4e), reveal that only the expression of GT$_{17}$-SEC62$_{IDR}$ and of GT$_{17}$-SEC62$_{IDRLIR}$ drives the formation of ER-derived vesicles visible in the cytoplasm (Fig. 4d,e, red arrowheads).

A FRAP assay of ER lumen connectivity performed with the HaloTagged variants of the same constructs (HT$_{17}$, HT$_{17}$-SEC62$_{IDR}$ and HT$_{17}$-SEC62$_{IDRLIR}$) confirms a substantial inhibition of fluorescence recovery in cells expressing the IDR modules SEC62$_{IDR}$ and SEC62$_{IDRLIR}$ (Fig. 4f,g, green curves), as compared with mock-transfected cells (Fig. 4f,g, black curve) and with cells expressing HT$_{17}$ (Fig. 4f,g, violet curve). This is consistent with the RT-TEM images showing that expression of the IDR modules results in physical separation of ER portions from the bulk ER and shows that the LIR function of the IDR module is dispensable for organelle fragmentation. Notably, FRAP experiments also confirm that the expression of the RHD of FAM134B that induces ER remodelling and formation of ER constrictions (Extended Data Fig. 1c, red arrowheads) does not inhibit the recovery of the fluorescent signal, consistent with the incompetence of the RHD of FAM134B expressed alone *in cellula* to fragment the ER (Fig. 4f,g, blue curve).

## ER-phagy receptors IDRs-driven mitophagy

Motivated by these findings and by the fact that also autophagy receptors at the limiting membrane of other organelles display cytoplasmic IDR modules with similar features (Fig. 1), we verified whether the IDR modules of ER-phagy receptors encode a signal driving lysosomal delivery of mitochondrial portions, when exposed at the limiting membrane of mitochondria. To this end, FAM134B$_{IDR}$, SEC62$_{IDR}$ and TEX-264$_{IDR}$ were expressed at the OMM. Briefly, MEF were mock-transfected (Fig. 5a) or were transiently transfected with a plasmid for expression of a sfGFP moiety targeted at the OMM by the transmembrane domain of the mitochondrial protein TOMM20 (Mito-GFP)[83] (Fig. 5b). Alternatively, the cells were transfected with plasmids for expression of the same membrane-tethered sfGFP moiety displaying at the C-terminus SEC62$_{IDR}$, FAM134B$_{IDR}$ or TEX264$_{IDR}$ modules (Fig. 5c,e,g), or the same

**Fig. 3 | Membrane-associated IDRs trigger ER-phagy. a**, CLSM images of delivery of CNX-positive ER portions (red) to LAMP1-positive EL (cyan) upon the expression of GT$_{17}$ in MEF treated with 50 nm BafA1. GT$_{17}$ consists of a luminal GFP (G in the schematics on the left) tethered at the ER membrane with a transmembrane domain of 17 residues (T$_{17}$)[68]. Schematics of GT$_{17}$-FAM134B$_{IDR}$, GT$_{17}$-SEC62$_{IDR}$ and GT$_{17}$-TEX264$_{IDR}$ chimeras and the control GT$_{17}$ construct without the IDR are shown. Scale bars, 1 μm. **b**, Same as **a**, where cells express GT$_{17}$-FAM134B$_{IDR}$ (that is, the GT$_{17}$ moiety displaying the IDR module of FAM134B at the cytoplasmic face of the ER membrane (left). **c**, Same as **b** for GT$_{17}$-SEC62$_{IDR}$. **d**, Same as **b** for GT$_{17}$-TEX264$_{IDR}$. **e–g**, Same as **b–d** for cells expressing GT$_{17}$-FAM134B$_{IDRLIR}$ (**e**), GT$_{17}$-SEC62$_{IDRLIR}$ (**f**) and GT$_{17}$-TEX264$_{IDRLIR}$ (**g**) mutants. Please also refer to Extended Data Fig. 2a–g for micrographs of entire cells.

**h**, LysoQuant quantification of LAMP1-positive EL accumulating CNX-positive ER portions in **a–g** (n = 15, 21, 16, 19, 17, 16 and 19 cells), N = 3 biological replicates. A one-way ANOVA with Turkey's multiple comparisons test was performed, F = 64.15. Adjusted P value: ****P < 0.0001. The mean bar is shown. **i**, GFP native in gel fluorescence and HaloTag assay showing enhanced delivery of ER-phagy reporter HT$_{17}$ to hydrolytically active EL upon expression of GT$_{17}$-SEC62$_{IDR}$ (lane 4), GT$_{17}$-FAM134B$_{IDR}$ (lane 6) and GT$_{17}$-TEX264$_{IDR}$ (lane 8), as demonstrated by the increase in Halo fragment (high contrast). N = 1 biological replicate. Bottom: the Coomassie loading control. **j**, IEM showing the localization of gold-labelled GT$_{17}$-SEC62$_{IDR}$ in the lumen of ER and ER portions in the cytoplasm (red arrowheads) or within EL (blue arrowheads). The dashed line shows the EL limiting membrane. Scale bar, 500 nm. N = 1 biological replicate. **k**, Same as **j** for GT$_{17}$-SEC62$_{IDRLIR}$.

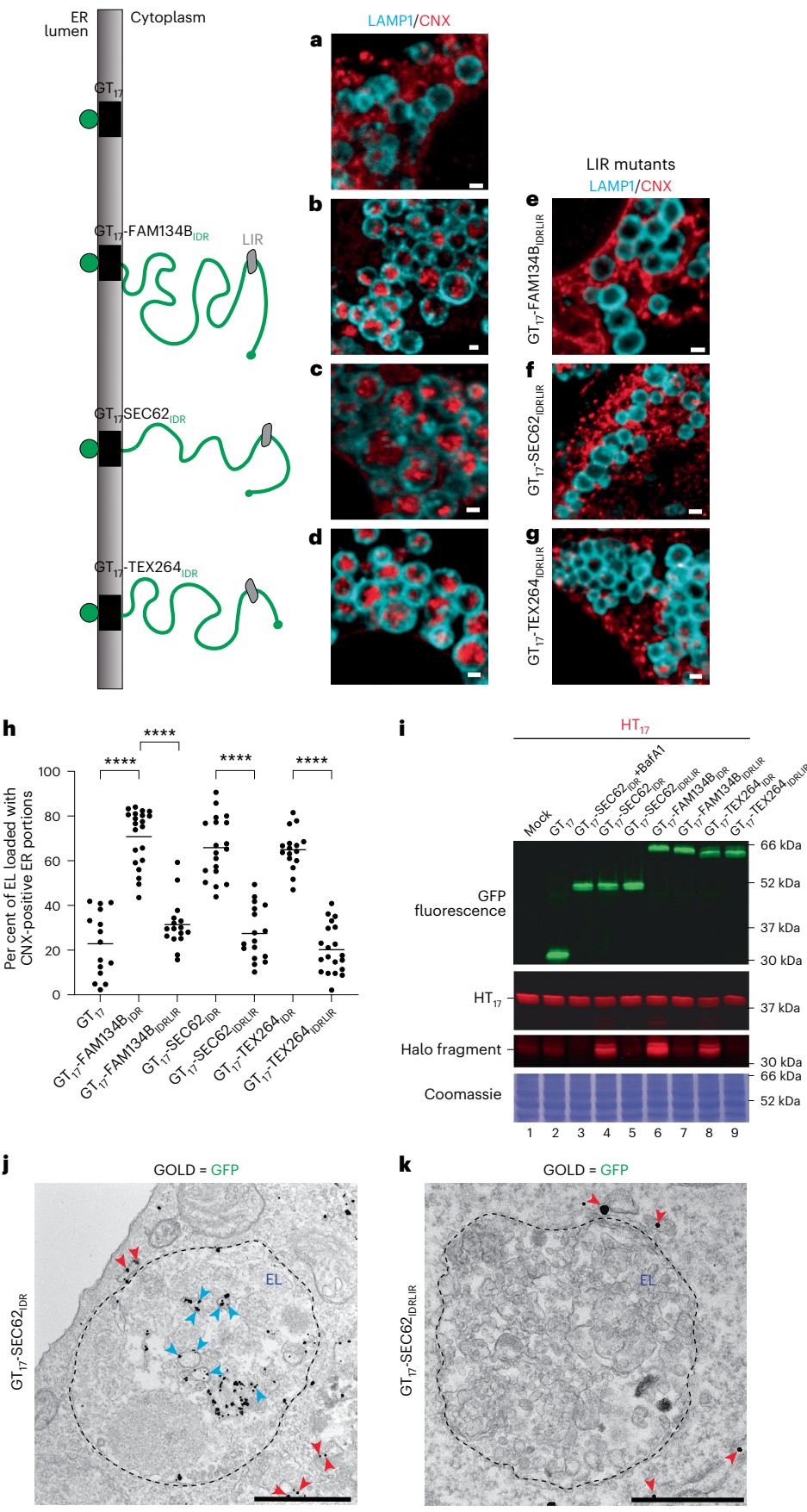

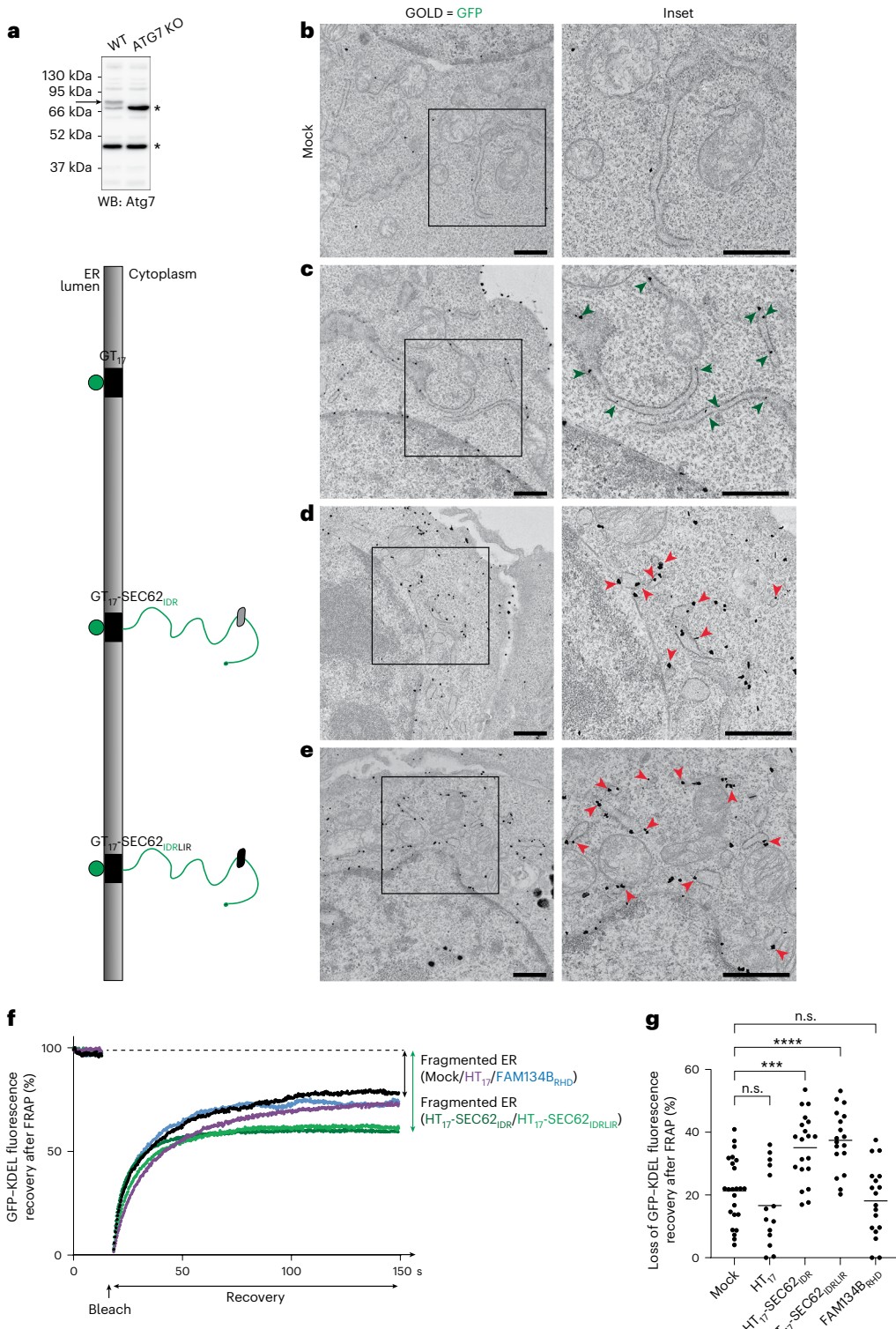

**Fig. 4 | ER fragmentation in ATG7-KO MEF expressing the IDR module of SEC62.**
**a**, Left: a western blot (WB) demonstrating the KO of ATG7. The arrow indicates the band corresponding to ATG7. The stars represent the unspecific bands. WT, wild type. **b**, IEM micrograph showing the ER morphology in mock-transfected ATG7-KO MEF. $N = 1$ biological replicate. **c**, Same as **b** in a cell expressing $GT_{17}$. The gold-labelled ER sheets are indicated with green arrowheads. **d**, An IEM micrograph showing fragmented ER in a cell expressing $GT_{17}$-SEC62$_{IDR}$. The red arrowheads indicate cytoplasmic ER fragments. **e**, Same as **d** in a cell expressing $GT_{17}$-SEC62$_{IDRLIR}$. Scale bars, 1 μm. The inset represents a magnification of 2×. **f**, Representative FRAP curves of ATG7-KO MEF expressing ER-lumen marker GFP–KDEL. In mock-transfected cells (black) the recovery of GFP–KDEL

fluorescence is efficient. Expression of $HT_{17}$ (violet) and FAM134B$_{RHD}$ does not notably affect the proportion of ER that recovers its fluorescence, indicating largely interconnected ER network. Expression of $HT_{17}$-SEC62$_{IDR}$ and $HT_{17}$-SEC62$_{IDRLIR}$ reduces the proportion of ER that recovers its fluorescence, indicating fragmented ER. **g**, A quantification of the fraction of GFP–KDEL fluorescence that is permanently lost after FRAP in **f**, $N = 4$ biological replicates for mock, $N = 3$ for $HT_{17}$-SEC62$_{IDR}$, $HT_{17}$-SEC62$_{IDRLIR}$ and FAM134B$_{RHD}$, $N = 2$ for $HT_{17}$; $n = 24, 14, 20, 18$ and 18 cells. An ordinary one-way ANOVA with a Dunnett's multiple comparisons test was performed, $F = 11.74$. ****$P < 0.0001$, ***$P = 0.0003$, $P$(mock versus $HT_{17}$) = 0.6066, $P$(mock versus FAM134B$_{RHD}$) = 0.8390; n.s., not significant. The mean bar is shown.

IDR modules carrying a mutation in the LIR domain that prevents LC3 engagement (Fig. 5d,f,h and Extended Data Fig. 3a–j, for the same experiment replicated in HEK293 cells and for the control of LC3 binding). The cells were exposed to BafA1 to preserve the endogenous TOMM20-positive mitochondrial fragments delivered within endolysosomal degradative compartments[63]. In mock-transfected cells and in cells expressing the Mito-GFP at the OMM (Fig. 5a,b), a low percentage of the LAMP1-positive EL accumulates the endogenous mitochondrial TOMM20 protein marker (quantifications with LysoQuant in Fig. 5j), reporting on the constitutive lysosomal turnover of mitochondrial portions. Mitochondrial transplantation of the SEC62$_{IDR}$, FAM134B$_{IDR}$ or TEX264$_{IDR}$ modules, enhances the delivery of the mitochondrial fragments within degradative LAMP1-positive EL (Fig. 5c,e,g,j). Mutations of the IDR's LIR motifs that inhibit engagement of LC3 proteins (Fig. 5d,f,h,j) or transplantation of the ER-phagy receptors IDR modules at the OMM in MEF lacking ATG7, which are defective in LC3 lipidation (Extended Data Fig. 4) hamper the delivery of TOMM20-positive mitochondrial portions within the degradative LAMP1 compartment.

Next, the HaloTag assay in cells expressing the mitophagy reporter HMAVS$_{TM}$ (HaloTag tethered to the mitochondrial OMM with the 31-residue-long transmembrane domain of the mitochondrial antiviral-signalling protein (MAVS)) reports on the absence of Halo fragment in mock-transfected cells and in cells expressing Mito-GFP at the OMM (Fig. 5i, lanes 1 and 2), indicating low detectable constitutive mitophagy. Expression of ER-phagy IDR modules tethered to the OMM (Fig. 5c,e,g) induces the formation of Halo fragments (Fig. 5i, lanes 4, 6 and 8), which requires hydrolytic lysosomal activity (that is, exposure of cells to BafA1 inhibits the generation of the Halo fragment, lane 3), indicating the delivery of mitochondrial fragments to the hydrolytically active degradative compartments. Importantly, this delivery is abrogated in cells expressing LC3-binding-deficient IDR modules (Fig. 5i, lanes 5, 7 and 9), once again indicating that mitophagy requires LC3 engagement by the ER-phagy receptors' IDR modules that have been transplanted at the limiting membrane of mitochondria.

In MEF, we also monitored the lysosomal delivery of the mitochondrial fragments generated by the expression of the Mito-GSEC62$_{IDR}$ polypeptide by IEM (Fig. 5k). The Mito-GSEC62$_{IDR}$ polypeptide immunolabelled with gold particles is seen at the OMM of mitochondria in the process of being engulfed by an EL (Fig. 5k, left) or of mitochondria within the lumen of the EL upon inactivation of hydrolytic enzymes with BafA1 (Fig. 5k, right), thus confirming the results of the CLSM analyses and of the HaloTag assay (Fig. 5c,i,j). The IEM also reveals that the mitochondrial portions captured by the LAMP1-positive EL maintain the characteristic morphology characterized by a double membrane and luminal cristae. The IEM analyses of cells that display at the OMM the Mito-GSEC62$_{IDRLIR}$ polypeptide (Fig. 5d) that fragments mitochondria ('ER-phagy receptors IDRs-driven mitochondrial fragmentation' section) but cannot engage LC3 (Extended Data Fig. 3j) reveal that these mitochondrial portions fail to be delivered within the EL and remain in the cytoplasm (Fig. 5l).

## ER-phagy receptors IDRs-driven mitochondrial fragmentation

The accumulation of mitochondrial portions within LAMP1-positive EL implies that expression of IDR modules at the OMM triggers mitochondrial fragmentation. As previously shown for the ER fragmentation induced by the expression at the ER membrane of the SEC62$_{IDR}$ IDR module, to directly monitor the mitochondrial fragmentation triggered upon transplantation of the same IDR modules at the OMM, the experiments shown in Fig. 5b–h were repeated in MEF lacking ATG7 (Fig. 6a–g), where the absence of LC3 lipidation[82] inhibits the delivery of mitochondria portions within the LAMP1-positive degradative organelles (Extended Data Fig. 4), thus preserving them in the cytoplasm. The morphology of mitochondria stained with a TOMM20-reactive antibody that decorates the OMM was first compared by CLSM in cells visible in the same coverslips that do express (Fig. 6a–g, green cells) or do not express (Fig. 6a–g, red cells) the GFP chimeras. The expression at the OMM of Mito-GFP has minor consequences on the network of mitochondria (Fig. 6a, insets 1 and 2, comparing the mitochondria in non-transfected cells with the mitochondria in the cell expressing the Mito-GFP). In cells expressing the cytoplasmic ER-phagy receptors' IDR modules attached to Mito-GFP, the mitochondria appear smaller (Fig. 6b–g, inset 2) compared with non-transfected cells in the same coverslips (inset 1) and to cells expressing Mito-GFP. Importantly, as previously shown for the IDR modules exposed at the ER membrane, the analyses of the light microscopy (Fig. 6b–g) and of the IEM micrographs (Fig. 6h–k and quantification in Fig. 6l) confirm that the LC3-binding function of the IDR modules is dispensable for organelle fragmentation. These results were also replicated in HEK293 cells, in which mitochondria labelled with IDR modules at the OMM (Extended Data Fig. 5b–g, inset 2, and quantification in Extended Data Fig. 5h) are smaller compared with the mitochondria in cells expressing Mito-GFP (Extended Data Fig. 5a, inset 2) or non-transfected cells (Extended Data Fig. 5a–g, inset 1, and quantification in Extended Data Fig. 5h), indicating the induction of mitochondrial fragmentation by ER-phagy IDR modules transplanted at the OMM.

## IDRs-induced mitochondrial fragmentation is driven by DRP1

The fission of mitochondria that precedes delivery of the fragmented organelles to endolysosomal degradative compartments via mitophagy pathways as induced by various cues including nutrient deprivation[84], hypoxia[85] or oxidative stress[86] is controlled by the DRP1 GTPase[87,88], although DRP1-independent mitophagy has also been observed[89]. To assess whether DRP1 controls the mitochondrial fission as induced by the transplantation of ER-phagy receptor's IDR modules at the OMM, Mito-GFP, Mito-GSEC62$_{IDR}$, and Mito-GSEC62$_{IDRLIR}$ polypeptides were expressed at the OMM of wild-type MEF (Fig. 7b–d) and in MEF lacking DRP1 (DRO1 knockout (KO))[90] (Fig. 7a,e–g). DRP1-KO cells have enlarged mitochondria (Fig. 7e–g, inset 1, DRP1-KO, and Fig. 7b–d, inset 1, wild-type MEF) confirming the functional KO of DRP1 leading to a general dysfunction of mitochondrial fission[90]. Expression of the Mito-GFP polypeptide does not affect the mitochondrial size

**Fig. 5 | Monitoring lysosomal delivery of fragmented mitochondria in wild-type MEF expressing the ER-phagy receptors IDR modules. a**, Representative CLSM images of delivery of TOMM20-positive mitochondrial portions (red) to LAMP1-positive EL (cyan) in mock-transfected MEF treated with 50 nM BafA1 and 100 nM TMR. Scale bar, 10 µm. The inset shows a 4× magnification of the merge image. **b**, Same as **a** in cells expressing Mito-GFP (green, GFP). **c**, Same as **b** in cells expressing Mito-GSEC62$_{IDR}$. **d**, Same as **c** in cells expressing LC3-binding-deficient Mito-GSEC62$_{IDRLIR}$. **e,f**, Same as **c** and **d** in cells expressing Mito-GFAM134B$_{IDR}$ (**e**) and Mito-GFAM134B$_{IDRLIR}$ (**f**). **g,h**, Same as **c** and **d** in cells expressing Mito-GTEX264$_{IDR}$ (**g**) and Mito-GTEX264$_{IDRLIR}$ (**h**). **i**, Native fluorescence control of GFP expression in cells expressing constructs from **a–h** (GFP, top). The HaloTag assay shows enhanced delivery of mitophagy reporter HMAVS$_{TM}$ to hydrolytically active EL upon expression of Mito-GSEC62$_{IDR}$ (lane 4), Mito-GFAM134B$_{IDR}$ (lane 6) and Mito-GTEX264$_{IDR}$ (lane 8), as demonstrated by the increase in Halo fragment

(high contrast). $N = 1$ biological replicate. Bottom: the Coomassie loading control. **j**, LysoQuant quantification of the percentage of cellular EL that are degrading TOMM20-positive mitochondria fragments in **a–h** ($N = 3$ biological replicates, $n = 31, 19, 21, 29, 37, 26, 28$ and $29$ for **a–h**, respectively). An ordinary one-way ANOVA with Turkey's multiple comparisons test is shown, $F = 168.8$. The mean bar is shown. Adjusted $P$ value: ****$P < 0.0001$; n.s., not significant ($P = 0.9582$). **k**, Transmission IEM of MEF expressing the Mito-GSEC62$_{IDR}$ polypeptide. Right: the micrograph shows an EL containing a mitochondrial fragment (red 'M'). The clearance of 'M' is prevented by cell exposure to BafA1. Left: the micrograph shows an EL caught in the act of engulfing a mitochondrial fragment (red 'M'). The arrowheads show gold particles used to immunolabel the Mito-GSEC62$_{IDR}$ polypeptide at the OMM. Scale bars, 500 nm. $N = 1$ biological replicate. **l**, Same as **k** in MEF, where the mitochondria display the Mito-GSEC62$_{IDRLIR}$ polypeptide at the OMM.

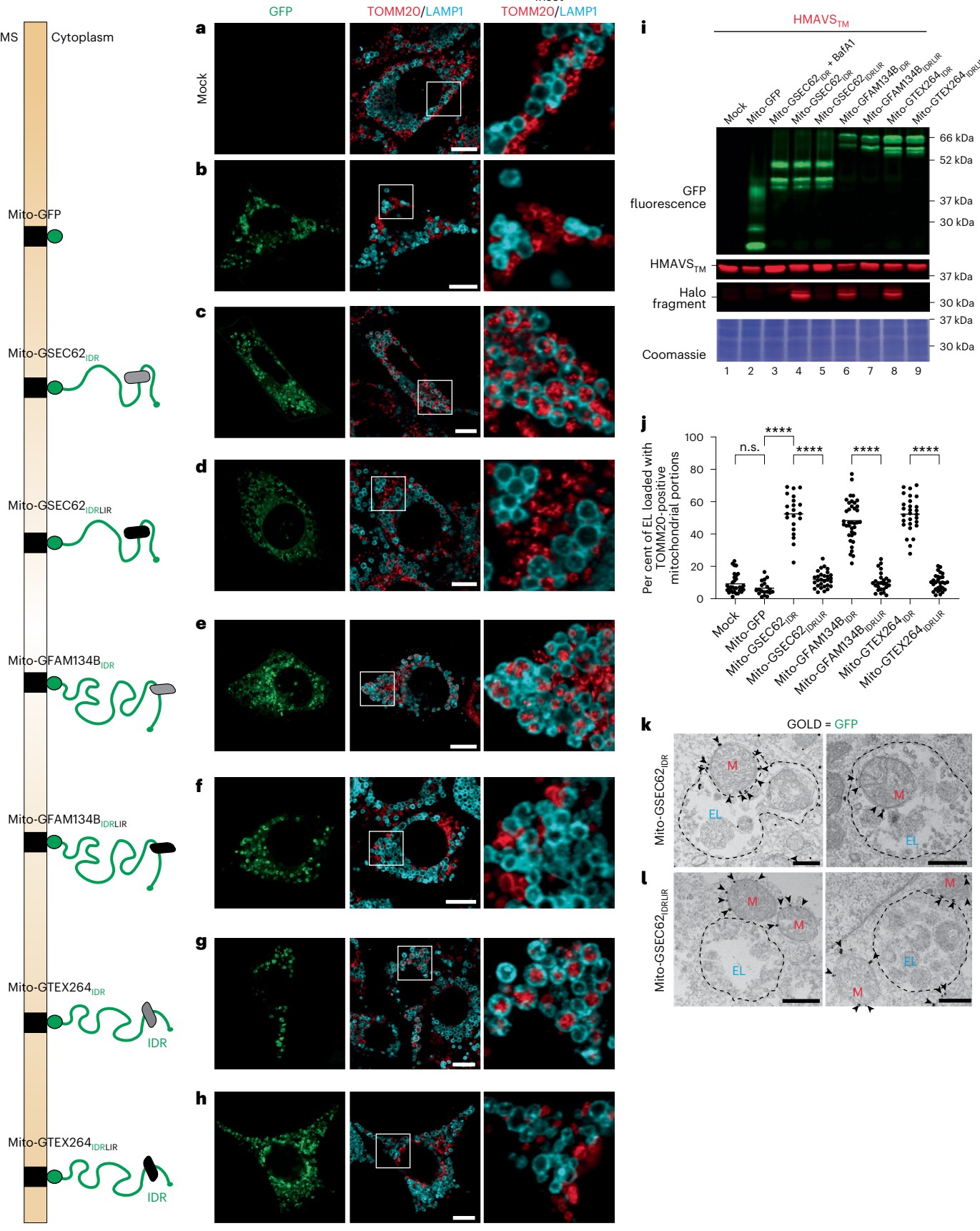

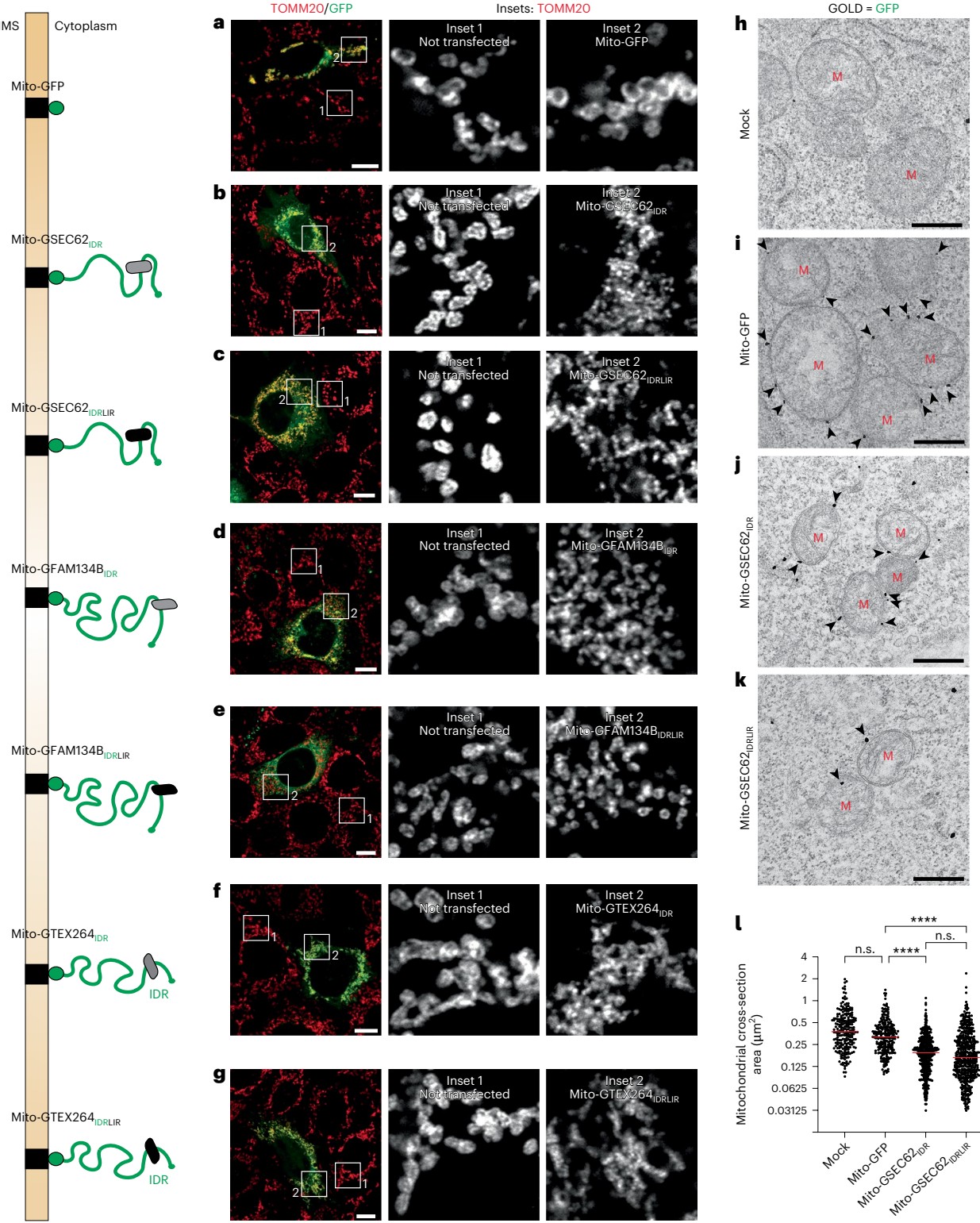

**Fig. 6 | Monitoring mitochondria fragmentation in AT7G7-KO MEF expressing the mitochondria-targeted ER-phagy receptors IDR modules.** **a**, The representative CLSM micrograph shows the MEF, whose mitochondria are labelled with an antibody for TOMM20 (red). Left: the green cell has been transfected and expresses Mito-GFP at the OMM (the transfected constructs are schematically shown). Inset 1 shows the mitochondrial structure in a non-transfected cell. Inset 2 shows the mitochondrial structure in the transfected cell expressing Mito-GFP. Scale bar, 10 μm; inset magnification, 6×. **b**, Same as **a** for a representative cell that expresses Mito-GSEC62_IDR (that is, the Mito-GFP displaying the SEC62 IDR module at the cytoplasmic site of the OMM). **c**, Same as **b** for a cell that expresses Mito-GSEC62_IDRLIR, which does not engage LC3.

**d**, Same as **b** for Mito-GFAM134B_IDR. **e**, Same as **c** for Mito-GFAM134B_IDRLIR. **f,g**, Same as **b** and **c** for Mito-GTEX264_IDR (**f**) and Mito-GTEX264_IDRLIR (**g**). **h**, RT-TEM showing mitochondria in mock-transfected cells. **i**, Same as **h** in a cell expressing Mito-GFP (the arrowheads are pointing at the gold particles marking GFP) at the OMM. **j**, Same as **i** in a cell expressing Mito-GSEC62_IDR. **k**, Same as **j** in a cell expressing Mito-GSEC62_IDRLIR. **l**, A quantification of mitochondrial cross-section area from TEM images (n = 231, 246, 449 and 493 mitochondria for **h**–**k**, respectively). A Kruskal–Wallis test with a Dunn's multiple comparisons test was performed, Kruskal–Wallis statistic of 293.6. The median bar (red) is shown. Adjusted P value: ****P < 0.0001; n.s., not significant (P = 0.0957 for mock versus Mito-GFP and P = 0.9999 for Mito-GSEC62_IDR versus Mito-GSEC62_IDRLIR).

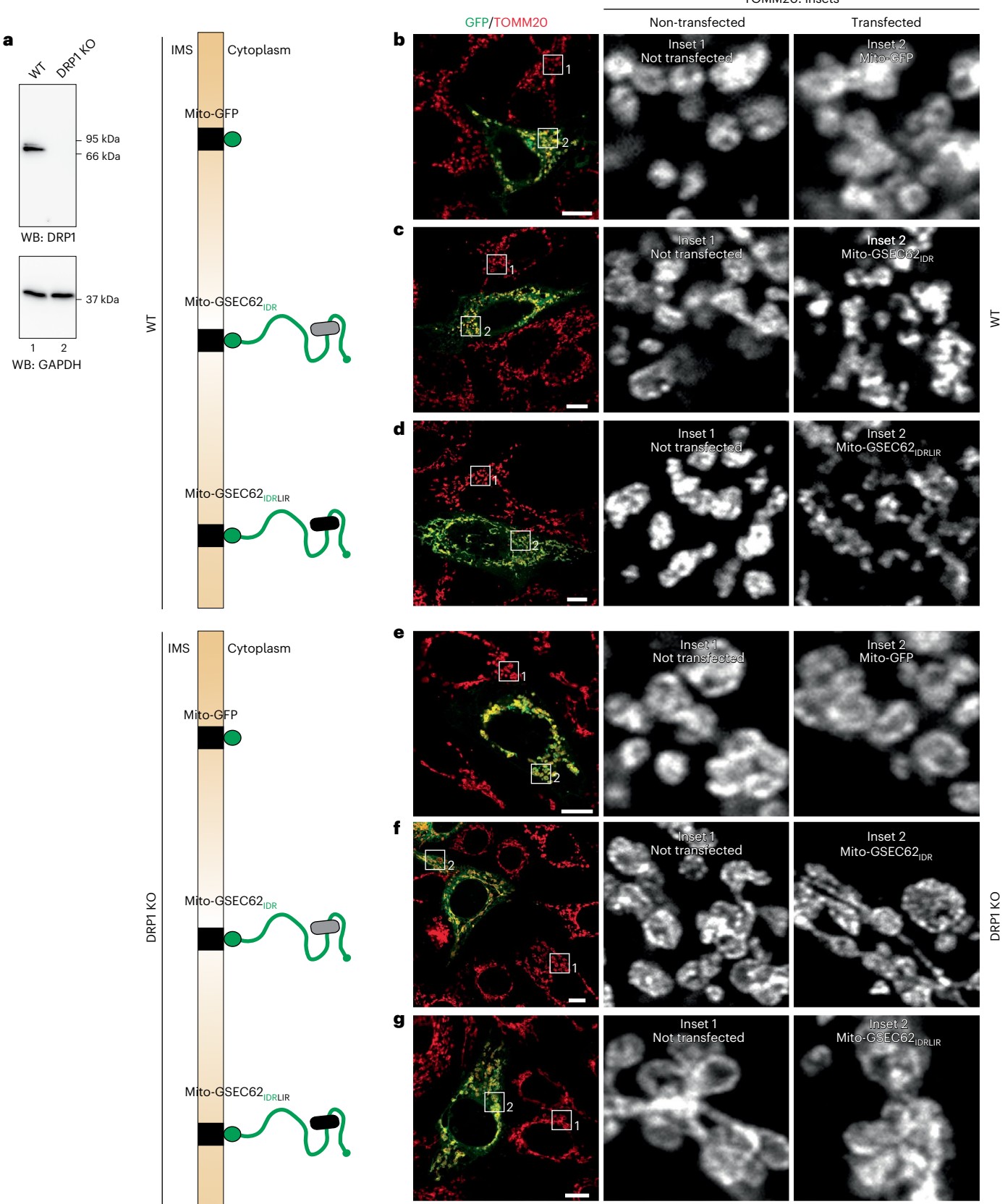

**Fig. 7 | IDR modules fail to induce mitochondrial fragmentation in DRP1-KO cells. a**, The western blot (WB) demonstrates the DRP1 KO (lane 2, top). GAPDH, glyceraldehyde 3-phosphate dehydrogenase. **b**, Monitoring mitochondria fragmentation in wild-type (WT) MEF expressing Mito-GFP at the OMM (green cell) and in non-transfected cells, whose mitochondria are labelled with an antibody for TOMM20 (red). Inset 1 shows the mitochondria in a non-transfected cell. Inset 2 shows the mitochondria in a cell expressing Mito-GFP. $N = 2$ biological replicates. **c**, Same as **b** for a representative cell that expresses Mito-GSEC62$_{IDR}$. **d**, Same as **c** for a representative cell that expresses Mito-GSEC62$_{IDRLIR}$, which does not engage LC3. Scale bar, 10 μm; inset magnification, 10×. **e**–**g**, Same as **b**–**d** in DRP1-KO MEF expressing Mito-GFP (**e**), Mito-GSEC62$_{IDR}$ (**f**) and Mito-GSEC62$_{IDRLIR}$ (**g**).

and morphology in either wild-type or DRP1-KO MEF (Fig. 7b,e, a comparison of inset 2 with the corresponding inset 1). The expression of Mito-GSEC62$_{IDR}$ and Mito-GSEC62$_{IDRLIR}$ polypeptides induces mitochondrial fragmentation in wild-type MEF (Fig. 7c,d, respectively) as shown above for MEF (Fig. 6) and HEK293 cells (Extended Data Fig. 5) but not in DRP1-KO cells (Fig. 7f,g, a comparison of inset 2 with the corresponding inset 1). Thus, DRP1 is necessary for the mitochondrial fragmentation induced by transplantation of ER-phagy receptors IDR modules at the OMM.

Next, we confirmed the involvement of DRP1 in the mitophagy induced by the transplantation of the FAM134B$_{IDR}$, SEC62$_{IDR}$ or TEX-264$_{IDR}$ IDR modules at the OMM by monitoring the lysosomal accumulation of TOMM20-positive mitochondrial portions in cells treated with BafA1. As shown in Fig. 5, in mock-transfected wild-type MEF (Extended Data Fig. 6a) and in MEF expressing the Mito-GFP chimera (Extended Data Fig. 6b) mitochondria are virtually absent from the lumen of LAMP1-positive EL, reporting on low levels of constitutive mitophagy. Transplantation of the LC3-binding-competent chimeras at the OMM of wild-type MEF enhances mitophagy (Extended Data Fig. 6c,e,g and LysoQuant quantification in Extended Data Fig. 6q, black dots), which is substantially inhibited upon mutations of the LIR that prevent LC3 engagement (Extended Data Fig. 6d,f,h and LysoQuant quantification in Extended Data Fig. 6q). In the DRP1-KO MEF, the delivery of TOMM20-positive mitochondrial portions within LAMP1-positive EL is substantially inhibited (Extended Data Fig. 6a–h versus Extended Data Fig. 6i–p and Extended Data Fig. 6q, quantifications, and Extended Data Fig. 7a versus Extended Data Fig. 7b and Extended Data Fig. 7l, quantification). This shows that DRP1 contributes to the fission of mitochondria that display the ER-phagy receptors IDR modules at their OMM, before the delivery of formed mitochondrial fragments within degradative EL.

### IDRs-induced mitophagy requires the GTPase activity of DRP1

Finally, to confirm that the GTPase activity of DRP1 controls the mitophagy programme elicited by exposure of ER-phagy receptors IDR modules at the OMM, the active (mCherry–DRP1), or the inactive forms of the GTPase (mCherry–DRP1$_{K38A}$)[91] were back-transfected in the DRP1-KO MEF. The analyses show that the back-transfection of the active form of DRP1 partially restores the capacity of the SEC62, FAM134B and TEX264 IDR modules competent for LC3 engagement to trigger mitophagy (Extended Data Fig. 7c,f,i,l, respectively) but the back-transfection of the inactive GTPase DRP1$_{K38A}$ does not (Extended Data Fig. 7d,g,j,l). As expected, reconstitution of the DRP1 activity does not activate mitophagy, if the IDR modules cannot engage LC3 (Extended Data Fig. 7e,h,k,l). Notably, the fissionase DRP1 is not involved in ER-phagy programmes induced by ER-phagy IDR modules (Extended Data Fig. 8).

### Expression of the FUNDC1 IDR at the OMM triggers mitophagy

The conservation amongst organellophagy receptors of membrane-tethered IDR modules with net cumulative negative charge led us to postulate a functional conservation that would allow heterologous transplantation to trigger autophagic regulation of a given organelle on demand. So far, we demonstrated that the IDR modules of ER-phagy receptors trigger mitophagy when transplanted at the OMM. Are IDR modules of mitophagy receptors (for example, of FUNDC1 (refs. 13,14)) competent for ER-phagy induction when transplanted at the ER membrane?

First, we assessed dispensability of the FUNDC1 tri-spanning membrane-tethering part in mitophagy induction. As shown above for the ER-phagy receptors, also for FUNDC1 the substitution of the original membrane anchor with a dissimilar transmembrane domain (the membrane anchor of the mitochondrial protein MAVS) does not affect the capacity of the polypeptide to control generation and delivery of TOMM20-positive mitochondrial portions within LAMP1-positive EL

(Fig. 8a,b,d). The inactivation of the LC3-engaging function of the LIR embedded in the FUNDC1$_{IDR}$-GMAVS module upon mutation of the -YEVL- tetrapeptide to -AAAA-, substantially inhibits mitophagy induction (FUNDC1$_{IDRLIR}$-GMAVS) (Fig. 8c,d).

### Expression of the FUNDC1 IDR at the ER triggers ER-phagy

Next, the IDR module of the mitophagy receptor FUNDC1 was targeted and anchored at the ER membrane with a conventional 17-residue membrane anchor (the T17 used above) (Fig. 8f). The ER-targeted FUNDC1$_{IDR}$-GT$_{17}$ engages lipidated LC3B-II as shown by co-immunoprecipitation (Fig. 8i, lane 6, compared with the LC3-II engagement by the membrane-anchored IDR module of the ER-phagy receptor SEC62, lane 3). LC3 engagement is abolished upon mutation of the -YEVL- tetrapeptide of FUNDC1 to -AAAA- (Fig. 8i, lane 7). The expression of FUNDC1$_{IDR}$-GT$_{17}$ substantially enhances the delivery of CNX-positive ER fragments within the LAMP1-positive EL (GT$_{17}$) (Fig. 8f,h) compared with the expression of ER-targeted GFP (GT$_{17}$) (Fig. 8e,h). The inactivation of the LC3-engaging function of the FUNDC1 IDR module transplanted at the ER membrane substantially inhibits the delivery of ER portions within LAMP1-positive EL (Fig. 8g,h).

### Conservation of features, net cumulative negative charge

To close the loop and go back to the functional conservation of cytoplasmic IDR modules among organellophagy receptors (Fig. 1), we analysed the capacity of the IDR of REEP1 (refs. 92,93) to fragment mitochondria, when targeted at the OMM. REEP1 is member of a conserved family of ER remodelling proteins that generate high ER membrane curvature. It displays a C-terminal IDR module of 51 residues, a length that is in the range of sizes observed for the IDR modules of organellophagy receptors (Fig. 1d). However, the IDR module of REEP1 is characterized by a net cumulative positive charge of 4.03 (Fig. 1e). In contrast to the IDR modules of ER-phagy and mitophagy receptors tested in this work, the transplantation of REEP1$_{IDR}$ at the OMM does not trigger mitochondria fragmentation (Extended Data Fig. 9a,b), even though this module is expressed at much higher level (Extended Data Fig. 9c, lane 11) than the corresponding modules derived from the ER-phagy receptors (Extended Data Fig. 9c, lane 8) that induce mitochondria fragmentation. Notably, the addition in the REEP1 IDR module of a nonapeptide containing the extended LIR sequence of SEC62 (-GNDFEMITK-) results in poor engagement of LC3, both when the REEP1 IDR module is expressed at the ER membrane (GT$_{17}$-REEP1$_{IDRFEMI}$) (Extended Data Fig. 9c, lane 5) or at the mitochondrial membrane (Mito-GREEP1$_{IDRFEMI}$, lane 10). Not surprisingly, therefore, the expression of the IDR module of REEP1 at the OMM or at the ER membrane does not induce mitochondrial (Extended Data Fig. 9d–h) nor ER delivery (Extended Data Fig. 9f–i) to the degradative LAMP1 compartment.

### Conservation of features, LIR distance from the membrane

As shown above, the engagement of LC3 proteins is dispensable for the organelle fragmentation function of the IDR modules of organellophagy receptors. However, it is required to execute the delivery of organelle portions within the degradative compartments. To monitor how the distance from the membrane of the LIR impacts on organellophagy, we made use of the ER-exposed IDR module of SEC62. The full-length IDR module (GT$_{17}$-GSEC62$_{IDR148}$ in Extended Data Fig. 10a) has a total length of 148 residues (including a 4-residue-long linker between the IDR and the T$_{17}$ transmembrane domain) and has 111 residues between the end of the membrane anchor and the LIR. This distance was progressively reduced to 55 (GT$_{17}$-GSEC62$_{IDR92}$), 28 (GT$_{17}$-SEC62$_{IDR65}$) and 10 residues (GT$_{17}$-SEC62IDR$_{47}$) (Extended Data Fig. 10b–d). Analyses by CLSM show that the induction of ER-phagy is maintained by the GT$_{17}$-GSEC62$_{IDR92}$ and the GT$_{17}$-SEC62$_{IDR65}$ modules (Extended Data Fig. 10b,c,f, quantification of ER delivery to the degradative LAMP1-positive EL). The delivery of CNX-positive ER portions within EL drops substantially on ER expression of the GT$_{17}$-SEC62$_{IDR47}$

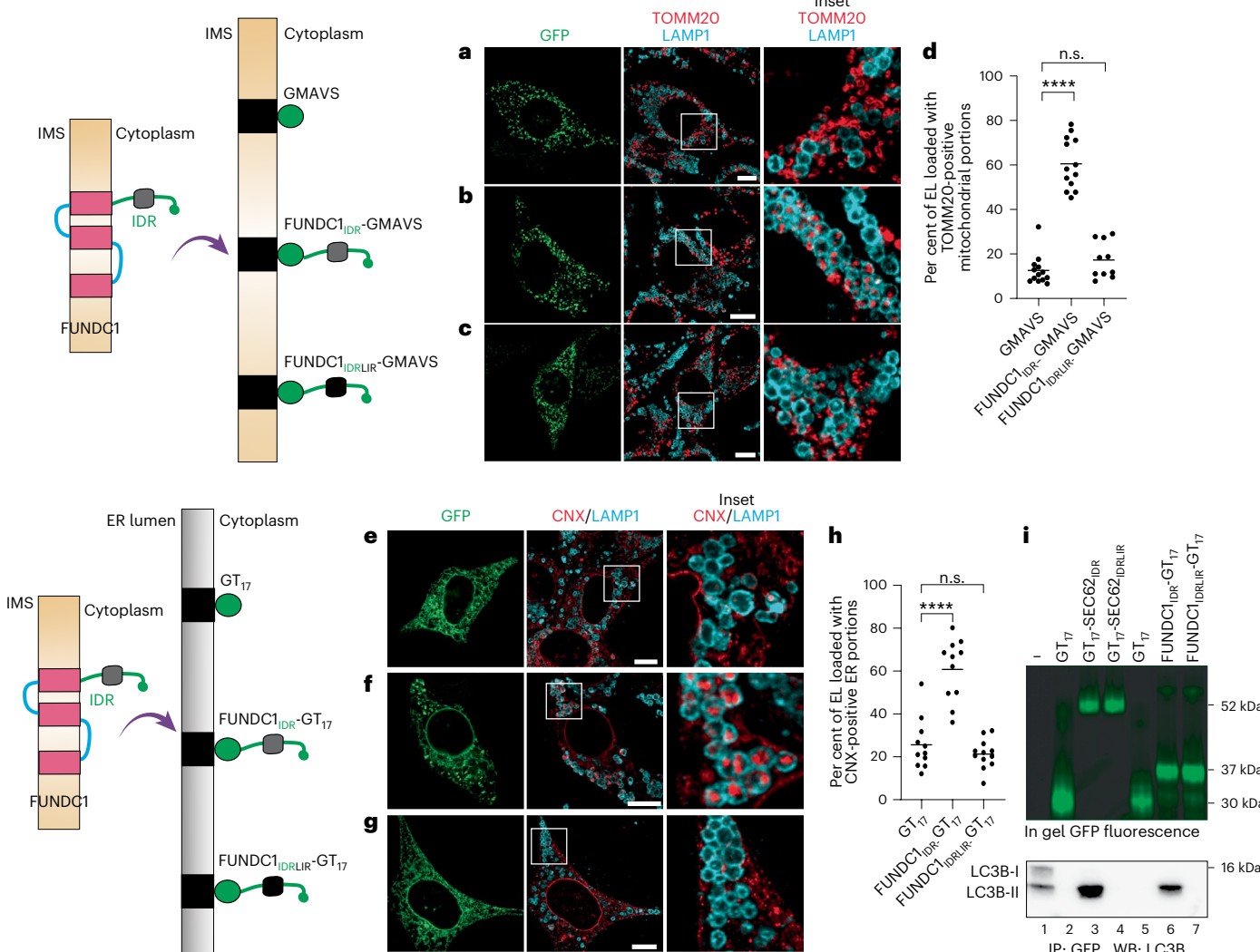

**Fig. 8 | Transplantation of the IDR module of the mitophagy receptor FUNDC1 at the mitochondrial membrane triggers mitophagy and at the ER membrane triggers ER-phagy. a**, A representative green MEF expressing OMM-anchored GFP (GMAVS, the constructs used for these experiments are tethered at the OMM with the transmembrane domain of the mitochondrial protein MAVS). Wild-type MEF were exposed to BafA1 to preserve mitochondria fragments (labelled with TOMM20, red) when delivered within LAMP1-positive EL (cyan). Scale bar, 10 μm; inset magnification, 4×. **b**, Same as **a** for a green cell expressing FUNDC1$_{IDR}$-GMAVS (left inset: the LAMP1-positive EL contain TOMM20-positive mitochondrial fragments) and a non-transfected black cell (right inset: whose LAMP1-positive compartment does not contain mitochondrial fragments). **c**, Same as **b** for cells expressing FUNDC1$_{IDRLIR}$-GMAV deficient in LC3 binding. **d**, LysoQuant quantification of the cells in **a**–**c** (N = 2 biological replicates;

n = 13, 13 and 10 cells for **a**–**c**, respectively). An ordinary one-way ANOVA with a Dunnett's multiple comparisons test was performed, F = 106.5. The mean bar is shown. Adjusted P value: ****P < 0.0001; n.s., not significant (P = 0.4462). **e**–**g**, Same as **a**–**c** for ER-targeted GFP (GT$_{17}$) (**e**), FUNDC1$_{IDR}$-GT$_{17}$ (**f**) and FUNDC1$_{IDRLIR}$-GT$_{17}$ (**g**) constructs. **h**, LysoQuant quantification, N = 2 biological replicates; n = 11, 11 and 12 cells for **e**–**g**, respectively. An ordinary one-way ANOVA with Dunnett's multiple comparisons test was performed, F = 40.35. The mean bar is shown. Adjusted P value: ****P < 0.0001; n.s., not significant (P = 0.5726). **i**, In gel GFP fluorescence (top) and the corresponding immunoprecipitation (IP)–western blot (WB) (bottom) showing the engagement of LC3B by ER-targeted IDRs of SEC62 and FUNDC1 expressed in wild-type MEF. N = 1 biological replicate for lanes 5–7 and N = 3 for lanes 1–4.

module that fails to engage LC3 proteins (Extended Data Fig. 10g), possibly because the short distance of the LIR from the lipid bilayer, 10 residues, impedes the access of LC3 proteins.

### Conservation of features, length of the IDR modules

All the IDR modules examined in these experiments conserve the net negative charge (−19.95 for GT$_{17}$-GSEC62$_{IDR148}$ and for GT$_{17}$-GSEC62$_{IDR92}$, −16.00 for GT$_{17}$-SEC62$_{IDR65}$ and −14.06 for GT$_{17}$-SEC62$_{IDR47}$) and would have a length consistent with the length of cytoplasmic IDR modules displayed by organellophagy receptors (Fig. 1d, the shortest IDR module tested in our experiments is the GT$_{17}$-SEC62$_{IDR47}$, with a total length of 47 amino acids that, despite the incapacity of engaging LC3 proteins, could be competent to induce organelle fragmentation).

To assess how the length of the IDR modules affects their capacity to induce organelle fragmentation, we progressively reduced the length of the SEC62 IDR modules exposed on a sfGFP moiety at the limiting membrane of mitochondria (Extended Data Fig. 10h–m). Our tests show that only the Mito-GFP23 IDR has lost the capacity to fragment the organelle, when exposed at the OMM (Extended Data Fig. 10h,m). This IDR maintains the net cumulative negative charge at physiological pH (−1.03). However, it is shorter—about half the length—of the shortest organellophagy receptors known so far (SAMM50 and FUNDC1) (Fig. 1d).

## Discussion

Delivery of organelle portions to degradative compartments requires the engagement of LC3 proteins by membrane-bound organellophagy

receptors. An open question in the field is how the organelle portions to be removed from the cell are physically separated from the bulk of the organelle that must be preserved. Seminal studies on ER-phagy receptors of the FAM134 family (FAM134A, FAM134B and FAM134C) established that their transcriptional induction, phosphorylation, ubiquitylation and homo-/hetero-oligomerization control ER fragmentation and lysosomal clearance[7,24,25,27,38,53,55,94,95]. FAM134 proteins are anchored at the ER membrane via a RHD, a membrane-tethering module found in ER shaping proteins of the reticulon and REEP families that generates high curvature and constrictions of the ER membrane[30,70–79]. The capacity of the FAM134B-RHD to remodel the ER membrane has been highlighted by molecular dynamics simulations and in vitro liposome remodelling and has been proposed to promote the ER fragmentation required for lysosomal clearance of select ER portions[7,23–28]. During revision of our manuscript, the same group reported that molecular dynamics simulations revealed that the IDR module of FAM134B might enhance the capacity of the RHD of FAM134B to fragment the ER[96]. Notably, the vast majority of organellophagy receptors are anchored at the membrane of their homing organelle (the ER, the mitochondria or the Golgi complex), via conventional multi- or single-spanning transmembrane domains that do not share the membrane remodelling function of RHD modules (Fig. 1). Thus, a model assuming that the membrane-tethering modules of organellophagy receptors determine or give essential contribution to organelle fragmentation cannot be generalized. As shown in Fig. 1, the cytoplasmic IDR modules with net cumulative negative charge are a shared trait of autophagy receptors at the surface of organelles that rely on fragmentation for homoeostatic control. Our work highlights the functional conservation of the cytoplasmic IDR modules of membrane-tethered organellophagy receptors in promoting the organelle fragmentation, which is a conditio sine qua non for the execution of the autophagic programmes that regulate size, activity and homoeostasis of the organelles in our cells. We report that (1) mammalian ER-phagy and mitophagy receptors expose at the cytoplasmic face of their homing organelle a functionally conserved IDR module with diverse primary sequence, a length between 47 and 250 residues, a net cumulative negative charge at physiologic pH and a LIR separated by at least 24 residues from the membrane of the organelle (mammalian Golgiphagy receptors and yeast ER-phagy and mitophagy receptors, whose function has not been analysed here, also expose these modules); (2) the expression of full-length ER-phagy receptors recapitulates ER fragmentation and lysosomal delivery independent of the ER remodelling capacity of their membrane-anchoring modules and bypassing the requirement for external triggering signals; (3) the membrane-anchoring modules of the three ER-phagy receptors analysed in this study (FAM134B, SEC62 and TEX264) determine the suborganelle distribution of the receptors but are dispensable for ER fragmentation and execution of the ER-phagy programme. Their substitution with an unrelated transmembrane domain has no impact on the induction of ER fragmentation and ER-phagy; (4) likewise, the membrane-anchoring module of the mitophagy receptor FUNDC1 can be replaced by an unrelated sequence that targets the IDR module at the OMM, with no consequences on activation of mitochondrial fragmentation and mitophagy; (5) tethering the IDR modules of ER-phagy receptors at the OMM induces mitochondrial fragmentation driven by the DRP1 fissionase and mitophagy; (6) tethering the IDR module of the mitophagy receptor FUNDC1 at the ER membrane triggers ER fragmentation and ER-phagy; (7) the engagement and lipidation of LC3 proteins is dispensable for organelle fragmentation but required for delivery of the organelle portions within LAMP1-positive degradative organelles; (8) finally, the non-compliance with features conserved amongst organellophagy receptors (that is, change from net cumulative negative to positive charge, reduction in size or in the distance of the LIR from the membrane) negates the capacity of the cytoplasmically exposed IDR modules to fragment the organelles and to deliver organelle portions to degradative compartments.

Negatively charged IDRs are common in the proteomes of nucleated cells, where they are mainly found in long aspartic and glutamic acid repeats[97]. Organellophagy receptors IDR modules seem to be characterized by more uniform negative charge distribution along their sequence, which can be further amplified by post-translational modifications such as phosphorylation that has been reported as an activation signal for Metazoan's ER-phagy receptors[53–55,95,98,99] and receptors controlling autophagic turnover of other organelles[100–103] (reviewed in ref. [104] for Fungi). Another peculiarity of IDR modules of organellophagy receptors is obviously the presence of LIR motifs that, upon activation, engage components of the autophagy machinery. It should not be considered a coincidence that the LIR located closer to a lipid bilayer is the one of FUNDC1, placed 24 residues apart. In our tests, the progressive shortening of the distance of the LIR from the membrane clearly generates spatial constraints for the engagement of LC3 proteins. Calculating an elongated structure and an estimated average length for a single amino acid of 0.38 nm (ref. [62]), the $GT_{17}$-$SEC62_{IDR65}$ module places the LIR at a distance of 28 residues from the membrane (that is, an estimated 10.64 nm that still allows LC3 engagement). By contrast, for the $GT_{17}$-$SEC62_{IDR47}$ module the LIR is placed at 10 residues (that is, 3.8 nm) from the lipid bilayer, a distance too short to accommodate an LC3 molecule bound to the IDR. This was confirmed in cellula, where the shortened IDR modules progressively lost the capacity to engage LC3 proteins and to deliver ER portions within degradative compartments. Clearly, the shortening of the IDR module also eliminates domains outside the LIR that contribute to the stable association with lipidated LC3.

For IDR-driven mitochondrial fragmentation our data reveal the intervention of the fissionase and GTPase DRP1, whose involvement in the fragmentation of mitochondria that precedes mitophagy has been previously reported[13,84–88,90]. The identity and/or the involvement of a GTPase that contributes with the ER-phagy receptors in the fragmentation of the ER, if any, remains to be established. The combination of a membrane-tethering module that specifies and pre-remodels the organelle and the suborganellar portion to be removed from cells[8–11,81] and of a functionally conserved IDR module that facilitates organelle fragmentation and engages cytosolic autophagy gene products to deliver organelle portions to the degradative compartments builds an elegant and modulable sensor to adapt organelle size and function to cellular needs.

The capacity to control the integrity of organelles and the lysosomal clearance of select portions of organelles may find applications in the treatment of diseases associated with dysfunctional regulation of organelle size and activity. For example, the enhanced expression of ER-phagy receptors and the resulting hyperactivation of constitutive ER-phagy render tumour cells more resistant to cellular stresses such as those induced upon therapeutic interventions[105–109]. Giant and megamitochondria have been described by pathophysiologists as early hallmarks of human disorders including liver diseases[110,111], cardiomyopathies[112] and alcohol-induced heart disease[113,114] as well as in pathophysiological conditions such as ageing[115–117], which leads to organelle dysfunction. Other organelles may also be affected by gigantism: pathological ER expansion due to chronic ER stress is a hallmark of metabolic and of ER storage disorders[118,119], while giant peroxisomes are present in hepatic tissues of patients affected with rhizomelic chondrodysplasia punctata and acyl-CoA oxidase deficiency[120].

Our work leads the way to develop organelle-targeting chimeras (ORGATACs), where an organelle-targeting signal is fused to a short IDR sequence that conserves the functional features of organellophagy IDR modules. Organelle-targeting chimeras may favour organelle fragmentation (in the absence of engagement of the autophagic machinery) or organelle fragmentation and turnover (upon engagement of LC3 proteins). This is expected to alleviate the symptoms of diseases characterized by organelle dysfunction also related to ageing, intraorganelle accumulation of harmful macromolecules including

misfolded proteins, and to promote de novo biogenesis of functional replacements. All in all, our study offers a method to control integrity and activity of intracellular organelles by surface activation of IDR modules with net negative charges. It highlights a remarkable example of functional conservation on sequence divergency that would support a model, where IDRs characterized by different protein primary structure may retain biophysical features that are important for their function[121] and paves the way to study the evolution of IDR modules to control organellar homoeostasis across species.

## Online content

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

## Methods

### Cell culture, transient transfection and inhibitors

MEF and human embryonic kidney 293 (HEK293) cells were cultured in DMEM supplemented with 10% foetal calf serum (FCS, Dubco) at 37 °C in a 5% $CO_2$ atmosphere. Transient transfections were carried out using jetPRIME transfection reagent (PolyPlus) according to the manufacturer's instructions. Wild-type and ATG7-KO MEF are a kind gift from M. Komatsu. Wild-type and DRP1-KO MEF are a kind gift from Mike Ryan and Susanna Manley. BafA1 (Calbiochem) was used at 50 nM for 15 h if not otherwise specified. HEK293 were purchased from ATCC (CRL-1573). ATG7-KO were checked for absence of ATG7 and for lack of LC3 lipidation (for example, refs. 8,82,122). DRP1-KO MEFs were checked for absence of DRP1 and enlarged mitochondria. Wild-type MEF cells resulted positive for DRP1 and ATG7, displayed classical fibroblast morphology and were positive for LAMP1 staining with mouse-specific LAMP1 antibody. HEK293 were obtained from ATCC, and their identity was confirmed by morphology.

### Expression plasmids and antibodies

ER-targeted IDR constructs of ER-phagy receptors were subcloned in a pcDNA3.1(+) backbone with an N-terminal sfGFP or HALO tag, an inactive bacterial hydrolase that covalently binds cell permeable ligands coupled to fluorescent probes[58]. The ER targeting of ER-phagy receptors' IDR portions was achieved by incorporating an N-terminal prolactin signal sequence, followed by sfGFP, -AGG-TPSETLITTVESNSSW- linker derived from the mouse Cyb5a protein and a $-T_{17}$ transmembrane anchor (-WTNWVIPAISALVVALM-) linker with the -YRGS- cytosolic flanking sequence. Constructs containing transmembrane domain modules were fused with a C-terminal cytosolic GFP or HALO moiety preceded by -AAAGT- or -AAASGAGS- linkers, respectively. ER-targeting of FUNDC1 IDR–GFP and GFP control constructs was achieved by adding the $T_{17}$ sequence at the C-terminal (-WTNWVIPAISALVVALM-) preceded by the sfGFP-GGS-PSETLITTVESNSSW- linker. Mitochondrial targeting of ER-phagy IDRs was achieved by fusing the first 33 amino acids of the human TOMM20 protein to the N-terminus, followed by a -GPVAT- linker, sfGFP, and the IDRs of ER-phagy receptors. Mitochondrial targeting of the FUNDC1 IDR–GFP and GFP control constructs was achieved by adding the transmembrane sequence of MAVS (-RPSPGALWLQVAVTGVLVVTLLVVLYRRRLH-) at the C-terminus, preceded by AAAA-GA-sfGFP and -GGS-PSETLITTVESNSSW- linker. The extended LIR sequence of SEC62 (-GNDFEMITK-) was generated by replacing the QP residues in the REEP1 IDR sequence -GQPK- with -NDFEMIT- to create $REEP1_{IDRFEMI}$. The sequences of IDRs and the corresponding linkers, cumulative charges and lengths are summarized in Supplementary Table 1. The antibodies used in this study, along with their respective dilutions, are listed in Supplementary Table 2.

### Cell lysis and western blot

Following the respective treatments, MEF were washed with ice-cold phosphate-buffered saline (PBS) containing 20 mM $N$-ethylmaleimide (NEM). The cells were then lysed using either 2% CHAPS (in HEPES-buffered saline (HBS), pH 6.8) or RIPA buffer (1% Triton X-100, 0.1% SDS, 0.5% sodium deoxycholate in HBS, pH 7.4), both supplemented with 20 mM NEM and protease inhibitors (200 mM phenylmethylsulfonyl fluoride, 16.5 mM chymostatin, 23.4 mM leupeptin, 16.6 mM antipain, 14.6 mM pepstatin) for 20 min on ice. The postnuclear supernatant (PNS) was obtained by centrifugation at 10,600$g$ for 10 min at 4 °C. The PNS was denatured by adding 100 mM dithiothreitol (DTT) and heated for 5 min at 95 °C before being subjected to SDS–polyacrylamide gel electrophoresis (PAGE). For in gel GFP-fluorescent assays, the PNS supplemented with DTT was kept for 5 min at room temperature (RT). GFP-fluorescent SDS–PAGE gels were scanned with on the Amersham Typhoon scanner (Cytiva) with 488 nm laser and 525BP20 filter. HaloTag(TMR)-fluorescent SDS–PAGE gels were scanned with the same instrument with 532 nm laser and 570BP20

filter. The protein bands were then stained with Coomassie or transferred onto polyvinylidene difluoride membranes using the Trans-Blot Turbo Transfer System (Bio-Rad). The membranes were blocked with 8% (w/v) non-fat dry milk (Bio-Rad) in Tris-buffered saline containing 1% Tween 20 (TBS-T, Sigma-Aldrich) and incubated overnight with primary antibodies diluted in TBS-T. Afterward, horseradish peroxidase (HRP)-conjugated protein A diluted in TBS-T was applied for 45 min. The protein bands were visualized using the WesternBright Quantum detection system (Advansta), and the signals were captured with the FusionFX chemiluminescence imaging system (VILBER/Witec). The quantification of western blot bands was performed using Fusion FX Edge 18.12 and ImageJ 2.16.0/1.54p software[123].

### Protein cross-linking with DSP

After treatment, the cells were washed with PBS and incubated with 1 mM dithiobis-succinimidyl propionate (DSP) (Thermo Fisher Scientific) (prepared from a 100× stock in dimethylsulfoxide) in PBS for 30 min at RT. The reaction was quenched by adding 1 M Tris (pH 7.8) to a final concentration of 20 mM, followed by a 15-min incubation at RT. The cells were then washed with PBS, treated with 20 mM NEM and lysed using RIPA buffer (1% Triton X-100, 0.1% SDS, 20 mM NEM, 0.5% sodium deoxycholate in HBS, pH 7.4) for 20 min on ice. The PNS were collected by centrifugation at 10,600$g$ for 10 min and used for immunoprecipitation of IDR chimera–LC3 complexes.

### Affinity purification-liquid chromatography/mass spectrometry

The HEK293 cells were transfected with GFP-tagged TMD/RHD, SEC62/SEC62LIR or $GT_{17}$-$SEC62_{IDR/IDRLIR}$ chimeras. Fifteen-hour post-transfection, cells were treated with 100 nM BafA1 for 6 h, crosslinked with DSP as described above, and the supernatant was collected. The protein complexes were isolated with 100 µl GFP–TRAP (Chromotek) following manufacturer's protocol. The samples were digested on beads following a modified version of the iST method (named miST method). A total of 25 µl of miST lysis buffer (1% sodium deoxycholate, 100 mM Tris pH 8.6, 10 mM DTT), were added to the beads. After mixing and dilution 1:1 (v:v) with $H_2O$, the samples were heated 5 min at 75 °C. The reduced disulfides were alkylated by adding 13 µl of 160 mM chloroacetamide (33 mM final) and incubating for 45 min at 25 °C in the dark. After digestion with 0.5 µg of trypsin/LysC mix (Promega #V5073) for 2 h at 25 °C, the sample supernatants were transferred in new tubes. To remove sodium deoxycholate, two sample volumes of isopropanol containing 1% TFA were added to the digests, and the samples were desalted on a strong cation exchange plate (Oasis MCX; Waters) by centrifugation. After washing with isopropanol/1% TFA, peptides were eluted in 200 µl of 40% MeCN, 59% water and 1% (v/v) ammonia and dried by centrifugal evaporation.

Tryptic peptide mixtures were injected on a Vanquish Neo nanoHPLC system interfaced via a nanospray Flex source to a high resolution Orbitrap Exploris 480 mass spectrometer (Thermo Fisher). The peptides were loaded onto a trapping microcolumn PepMap100 C18 (5 mm × 1.0 mm ID, 5 µm, Thermo Fisher) before separation on a C18 custom packed column (75 µm ID × 45 cm, 1.8 µm particles, Reprosil Pur, Dr. Maisch), using a gradient from 2% to 80% acetonitrile in 0.1% formic acid for peptide separation at a flow rate of 250 nl min$^{-1}$ (total time 130 min). Full MS survey scans were performed at 120,000 resolution. A data-dependent acquisition method controlled by Xcalibur software (Tune 2.9, Thermo Fisher Scientific) was used that optimized the number of precursors selected ('top speed') of charge 2+ to 5+ while maintaining a fixed scan cycle of 2 s. The peptides were fragmented by higher energy collision dissociation with a normalized energy of 30% at 15,000 resolution. The window for precursor isolation was of 1.6 $m/z$ units around the precursor and selected fragments were excluded for 60 s from further analysis. The intensity values were obtained following MaxQuant (V. 2.1.4.0)[124] analysis. Only the proteins

with at least two detected peptides were considered for analyses. The results were normalized by median abundance, the missing values were imputed from a normal distribution of 1.8 standard deviation down shift and with a width of 0.3 of each sample, and unpaired two-tailed tests were used to calculate the *P*value. Data analyses and graph plotting were done with the Perseus platform (version 2.0.5)[125] and MATLAB (The MathWorks, R2023b Update 5 23.2.0.24591. The raw data are deposited in the Pride database (project accession PXD060519).

## CLSM

MEFs were seeded onto alcian blue-coated (Sigma) glass coverslips (VWR) and transiently transfected using jetPRIME reagent following the manufacturer's protocol. For LysoQuant analysis, 8 h post transfection, the cells were treated with 50 nM BafA1 for 15 h. Moreover, the cells transfected with HaloTag-fusion proteins were incubated with 100 nM TMR HaloTag ligand (Promega). A total of 24 h post transfection, the cells were fixed at RT for 20 min using 3.7% formaldehyde in PBS. To permeabilize the membranes, coverslips were incubated for 20 min in a permeabilization solution containing 10% goat serum, 10 mM HEPES, 15 mM glycine and 0.05% saponin. After permeabilization, the cells were incubated with primary antibodies diluted in permeabilization solution (as specified in Supplementary Table 2) for 120 min. Following three washes in permeabilization solution, the cells were then incubated with Alexa Fluor-conjugated secondary antibodies diluted 1:300 in permeabilization solution for 45 min. The cells were rinsed three times with permeabilization solution and water, then mounted onto glass microscope slides (Epredia) using a drop of Mount Liquid anti-fade (Abberior). Confocal images were captured using Leica TCS SP5, STELLARIS 5 and STELLARIS 8 microscopes equipped with a Leica HCX PL APO lambda blue 63×/1.40 oil objective and a pinhole set at 1 a.u. Image acquisition was performed with Leica LAS X software, utilizing diode at 405 nm or laser beams at 489, 499, 552, 561, 587 and 653 nm wavelengths for excitation. Fluorescence emissions were collected at the following ranges: 430–490 nm (Alexa Fluor 405), 494–557 nm (GFP), 504–587 nm (Alexa Fluor 488), 557–663 nm (TMR), 592–644 nm (Alexa Fluor 568) and 658–750 nm (Alexa Fluor 646). Image analysis and quantification were performed using LysoQuant and ImageJ 2.16.0/1.54p software[64,123]. Image post-processing was performed with Adobe Photoshop (v26.2.0).

## RT-TEM and electron tomography

MEFs were plated on gridded glass-bottom dishes (MatTek Corporation) and transiently transfected using jetPRIME reagent following the manufacturer's protocol. A total of 24 h after transfection, the cells were prefixed with a 4% formaldehyde EM-grade solution supplemented with 0.1% glutaraldehyde. For morphological ER analyses, the coordinates of the cells on the finder grid were determined using wide-field microscopy on the Leica STELLARIS 8 microscope with Leica LAS X 4.5.0.025531 software with Leica PL APO 10× 0.40 air objective with pinhole 2 a.u., and the transfected and non-transfected cells were identified based on GFP fluorescence (493–556 nm range). The cells were then fixed with a solution containing 2.5% glutaraldehyde in 0.1 M sodium cacodylate buffer (pH 7.4). After several washes in sodium cacodylate buffer, the cells were post-fixed in 1% osmium tetroxide (OsO4), 1.5% potassium ferricyanide (K4(Fe(CN)6)) in 0.1 M sodium cacodylate buffer for 1 h on ice, washed with distilled water and stained en bloc with 0.5% uranyl acetate in distilled water overnight at 4 °C in the dark. The samples were then rinsed in distilled water, dehydrated with increasing concentrations of ethanol and embedded in Epoxy resin (Sigma-Aldrich). After curing for 48 h at 60 °C, ultrathin sections (70–90 nm thickness) were collected using an ultramicrotome (UC7, Leica microsystem), stained with uranyl acetate and Sato's lead solutions and observed under a Transmission Electron Microscope Talos L120C (FEI, Thermo Fisher Scientific) operating at 120 kV. The images of transfected and non-transfected cells were acquired with a Ceta CCD

camera (FEI, Thermo Fisher Scientific). For three-dimensional serial section electron tomography, serial sections (130–150 nm thickness) were collected on formvar-carbon coated slot grids, tilted images series (+60°) were acquired with a Talos L120C TEM using Tomography 5 software (FEI, Thermo Fisher Scientific). The tilted images were aligned, and tomograms reconstructed using IMOD software (v. 4.11.24) (organelle segmentation was performed in Microscopy Image Browser (MIB, v. 2.84 (ref. 126)) and visualized in IMOD v4.11.24 (ref. 127)).

## Immunogold electron microscopy

MEFs were seeded onto glass-bottom dishes (MatTek Corporation) and transiently transfected with GFP-tagged chimeras using jetPRIME reagent, following the manufacturer's instructions. A total of 8 h after transfection, the cells were either treated with 50 nM BafA1 for 15 h and then fixed for 2 h in periodate-lysine-paraformaldehyde[128] (FUJIFILM Wako) at RT or fixed 25 h post-transfection without treatment. After the fixative was removed by washing with 1× PBS, the cells were incubated with 50 mM glycine and blocked for 30 min in a blocking solution (0.2% bovine serum albumin, 5% goat serum, 50 mM NH₄Cl, 0.1% saponin, 20 mM phosphate buffer, 150 mM NaCl) at RT. The cells were stained with primary rabbit anti-GFP antibody (Abcam) and gold-labelled secondary antibodies (Nanoprobes) in the blocking buffer at RT. Afterwards, the cells were refixed for 30 min in 1% glutaraldehyde, and the nanogold particles were enhanced using a gold enhancement solution (Nanoprobes) as per the manufacturer's protocol. The cells were post-fixed with osmium tetroxide (OsO₄) and processed for electron microscopy. The samples were then rinsed with distilled water, dehydrated through a graded ethanol series and embedded in epoxy resin (Sigma-Aldrich). After curing for 48 h at 60 °C, ultrathin sections (70–90 nm) were cut using an ultramicrotome (UC7, Leica Microsystems), stained with uranyl acetate and Sato's lead solutions. The images were captured using a Ceta CCD camera (FEI, Thermo Fisher Scientific) and Velox 3.6.0 software (FEI, Thermo Fisher Scientific) on a Talos L120C TEM (FEI, Thermo Fisher Scientific) operating at 120 kV. For live FRAP analyses, ATG7-KO MEF expressing GFP–KDEL and HaloTag fusion proteins were imaged for 50 frames before bleach, then the GFP signal was bleached by 20 pulses of 405 diode at 100% intensity and imaged post-bleaching for 500 frames with the frame rate of 0.261 s. The bleaching and intensity measurements were done over a peripheral ER area of 29 µm². The intensity values were then normalized to prebleach measurements, and the maximum normalized postbleach value was used to assess ER fragmentation.

## Statistical analyses, data acquisition and blinding

The statistical comparisons and graphical plots were performed using GraphPad Prism 10 (GraphPad Software, version 10.1.2 for Windows). A data analysis for proteomics data was performed in Perseus v2.0.5. Tests employed for statistical analysis are reported in figure legends. Data distributions (individual points) are always shown. Whenever data distribution was not assumed to be normal, an appropriate statistical test was used to take it into account. An adjusted *P* value <0.05 (for one-way and two-way analysis of variance (ANOVA) with multiple comparison tests) was considered statistically significant. The ANOVA test results were corrected for multiple comparisons. All *t*-tests performed are two-sided. All experimental replicates represent independent biological replicates. Figure legends specify the number of cells, EL or mitochondria analysed (*n*) and the number of independent biological replicates of the experiment (*N*). No statistical methods were used to predetermine sample sizes, but our sample sizes are similar to those reported in previous publications[57,64,65,129,130]. No randomization was performed as this is not common for western blot or microscopy analysis. The covariates were controlled with the corresponding negative controls.

Unbiased imaging data collection/analyses were performed by at least two different scientists, one of which was blind to the identity of the sample. Morphological analyses of mitochondrial size were

performed by the scientists blinded to the identity of the sample. The fluorescent image quantifications of ER and mitochondria delivery to ELs were performed using LysoQuant, an unbiased and automated deep learning tool. Acquisition of MS data was done by scientist blinded to the nature of the sample and the scope of the experiment. For western blot and immunoprecipitation analyses, conclusions were drawn based on qualitative presence/absence of band signal; therefore, blinding is not relevant. For electron microscopy, acquisition of samples was performed by a scientist blinded to the nature of the sample. No data were excluded from the analyses.

## Reporting summary

Further information on research design is available in the Nature Portfolio Reporting Summary linked to this article.

## Data availability

The MS proteomics data supporting the data in Extended Data Fig. 1f,g can be accessed from the ProteomeXchange Consortium via the PRIDE partner repository under accession code PXD060519. Uncropped blots and gels are included in image source data. No restrictions on data availability apply. All materials used in the analysis are available, without restriction upon reasonable request to reproduce or extend the analyses. Source data are provided with this paper.

## Code availability

No custom code was developed or used in this study.

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

## Acknowledgements

We thank the members of Molinari's laboratory for discussions and critical reading of the manuscript and T. Calì for suggestions on mitochondrial targeting. We thank S. Manley, M. Ryan, T. Saitoh and M. Komatsu for gifts of cell lines. Funding was provided by the Swiss National Science Foundation (grant nos 310030_214903 and 320030-227541 to M.M.).

## Author contributions

Conceptualization: M.M., M.R. and C.G. Methodology: M.R., C.G., A.R. and M.M. Investigation: M.R. and C.G. Visualization: M.R., C.G., A.R. and M.M. Funding acquisition: M.M. Project administration: M.M. Supervision: M.M. Writing—original draft: M.M. Writing—review and editing: M.M., M.R. and C.G.

## Competing interests

The authors declare no competing interests.

## Additional information

**Extended data** is available for this paper at https://doi.org/10.1038/s41556-025-01728-4.

**Correspondence and requests for materials** should be addressed to Maurizio Molinari.

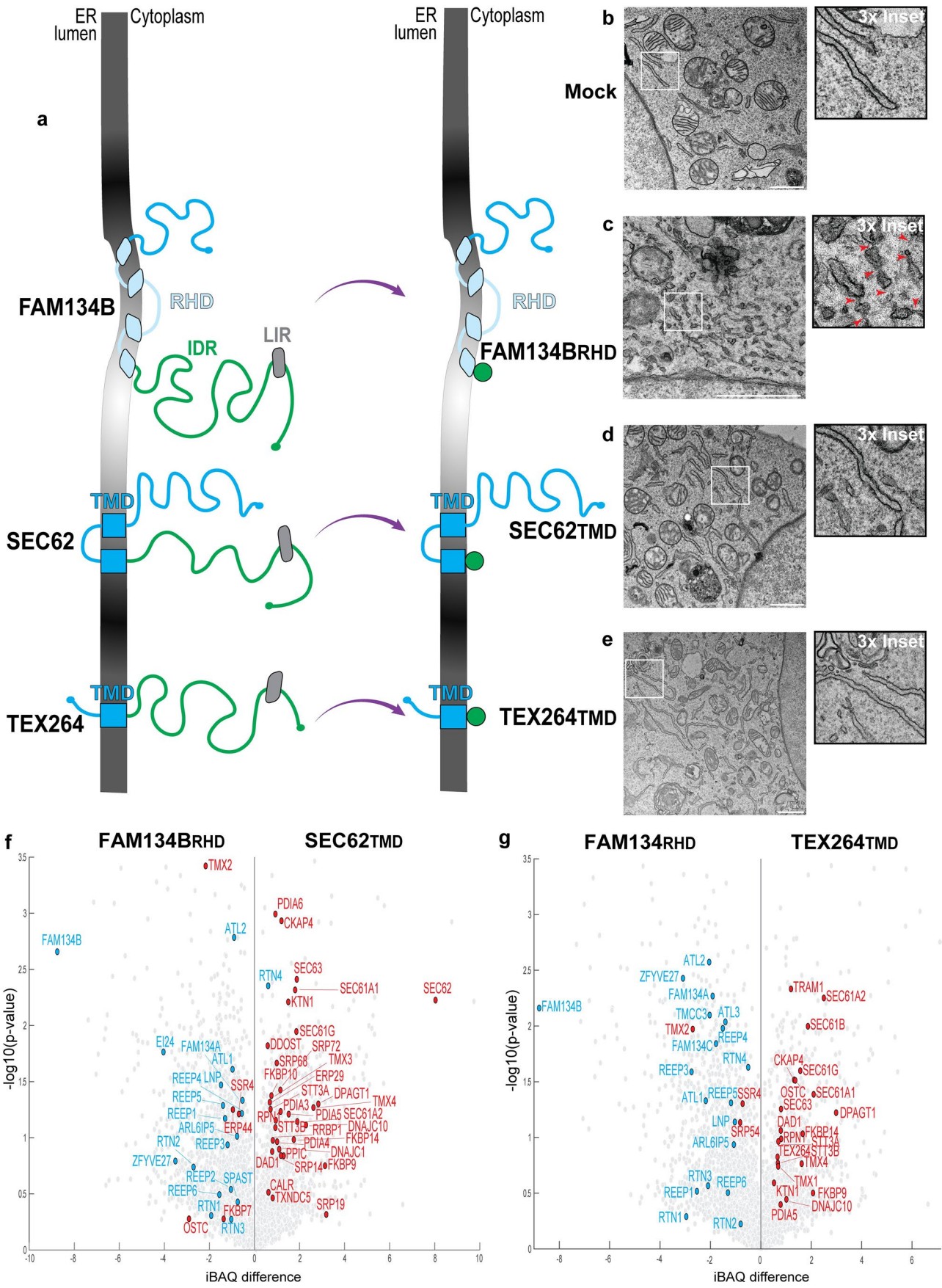

**Extended Data Fig. 1 | See next page for caption.**

**Extended Data Fig. 1 | Membrane remodeling and sub-compartmental localization functions of membrane-tethering modules.** (**a**) ER-phagy receptors FAM134B, SEC62, and TEX264. TMD: transmembrane domain; RHD: reticulon homology domain; LIR: LC3-interacting region; IDR: intrinsically disordered region. (**b**) RT-TEM showing ER ultrastructure in mock-transfected MEF (also in Supplementary Video 1). Scale bar: 1 μm. N = 1 biological replicate. (**c**) Same as (**b**) showing constricted ER tubuli in MEF expressing FAM134B$_{RHD}$ (Supplementary Video 2). (**d**) Same as (**c**) for SEC62$_{TMD}$-transfected cell, showing intact ER morphology (Supplementary Video 3). (**e**) Same as (**d**) for TEX264$_{TMD}$ (Supplementary Video 4). (**f**) Vulcano plot, endogenous ER proteins among FAM134B$_{RHD}$ or SEC62$_{TMD}$ interactors in HEK293 cells, N = 2 biological replicates. Two-tailed t-test, with permutation-based FDR not considered. -Log10(Adjusted p-value) is plotted on the y axis. (**g**) Same as (**f**) for FAM134B$_{RHD}$ and TEX264$_{TMD}$.

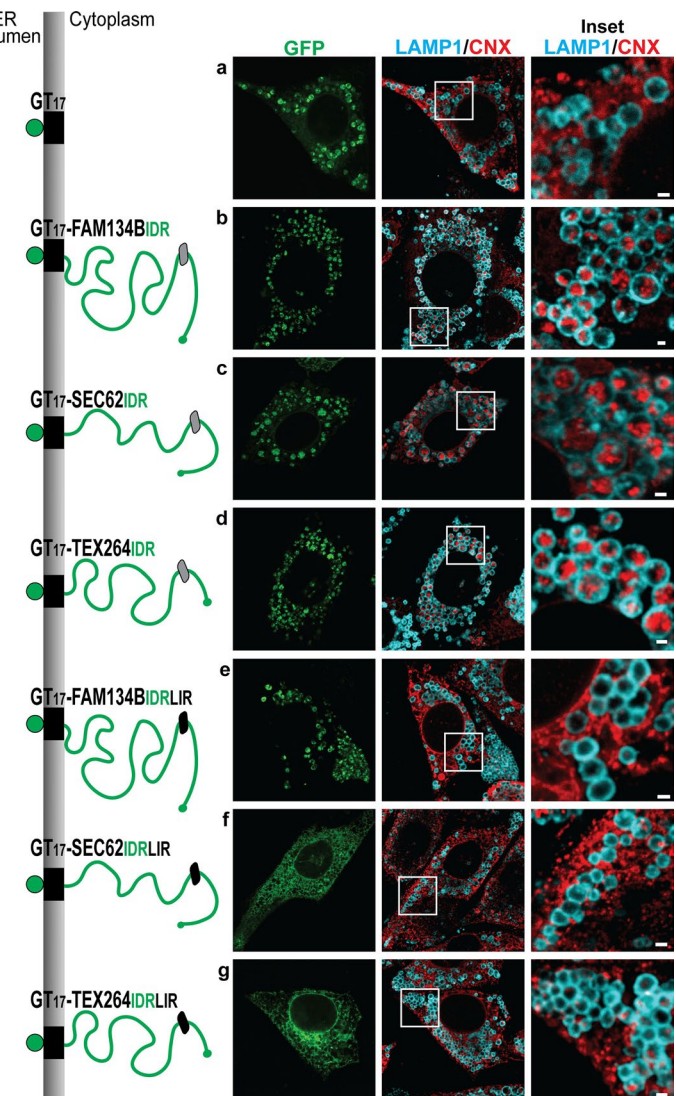

**Extended Data Fig. 2 | Membrane-associated IDRs trigger ER-phagy. (a-g)** Schematics and full cell CLSM images of insets shown in Fig. 3a–g. Scale bar 1 µm.

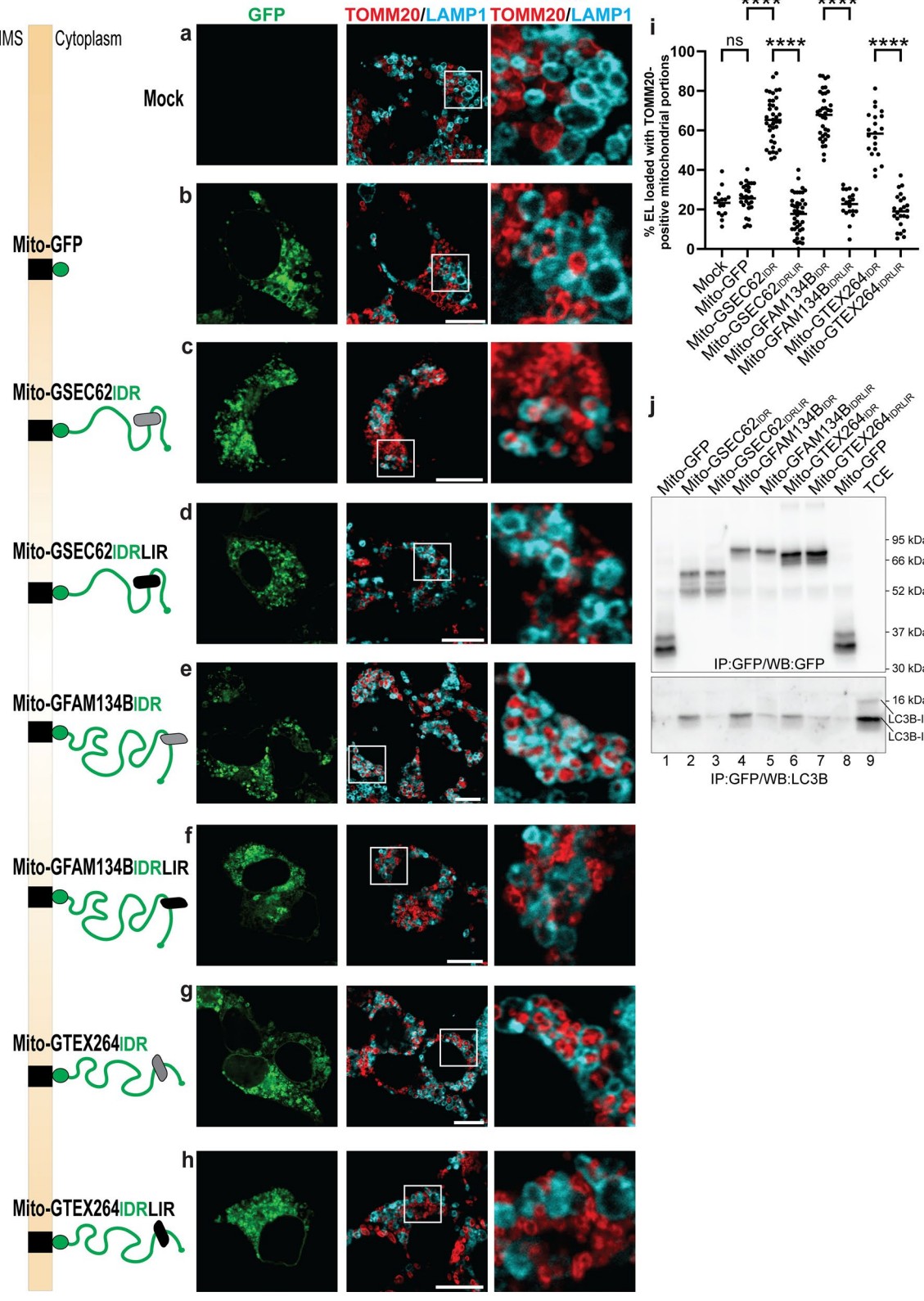

**Extended Data Fig. 3 | See next page for caption.**

**Extended Data Fig. 3 | Monitoring lysosomal delivery of mitochondrial fragments in HEK293 cells expressing the mitochondria-targeted ER-phagy receptors IDR modules.** (**a**) CLSM images of mock-transfected cells exposed to BafA1 to stabilize the cargo delivered within degradative endolysosomes and chemically fixed for immunostaining. Mitochondria are labeled with anti-TOMM20, endolysosomes with anti-LAMP1. Scale bar: 10 μm, inset magnification: 4x. (**b**) Same as (**a**) in cells transfected with Mito-GFP chimera. (**c-h**) Same as (**b**) in cells exposed to BafA1 to stabilize the cargo delivered within LAMP1-positive degradative endolysosomes and expressing mitochondria-targeted IDR chimeras. (**i**) LysoQuant quantification of the percentage of cellular endolysosomes that contain TOMM20-positive fragmented mitochondria in panels **a-h** (N = 3 biological replicates for **a-d**, N = 2 for **e-h**, n = 15, 28, 38, 41, 36, 20, 21, and 24 cells, for **a-h**, respectively). Ordinary one-way ANOVA with Turkey's multiple comparisons test, F = 149.6. Mean bar is shown. Adjusted P value: ****p < 0.0001, ns (P = 0.9960), not significant. (**j**) Mitochondria-targeted ER-phagy receptors IDR modules interact with LC3B-II through their LC3-interacting motif. Co-immunoprecipitation of GFP-tagged IDR modules (bait, upper panel) with LC3B (lower panel) from WT MEF cells treated with 50 nM BafA1 for 6 hours. LC3B-II interacts with IDR modules containing active LIR domains (lanes 2, 4, and 6). Lane 9: detergent-soluble total cell extracts (TCE) is shown. N = 3 biological replicates for lanes 1-3 and 9, N = 1 for lanes 4-8.

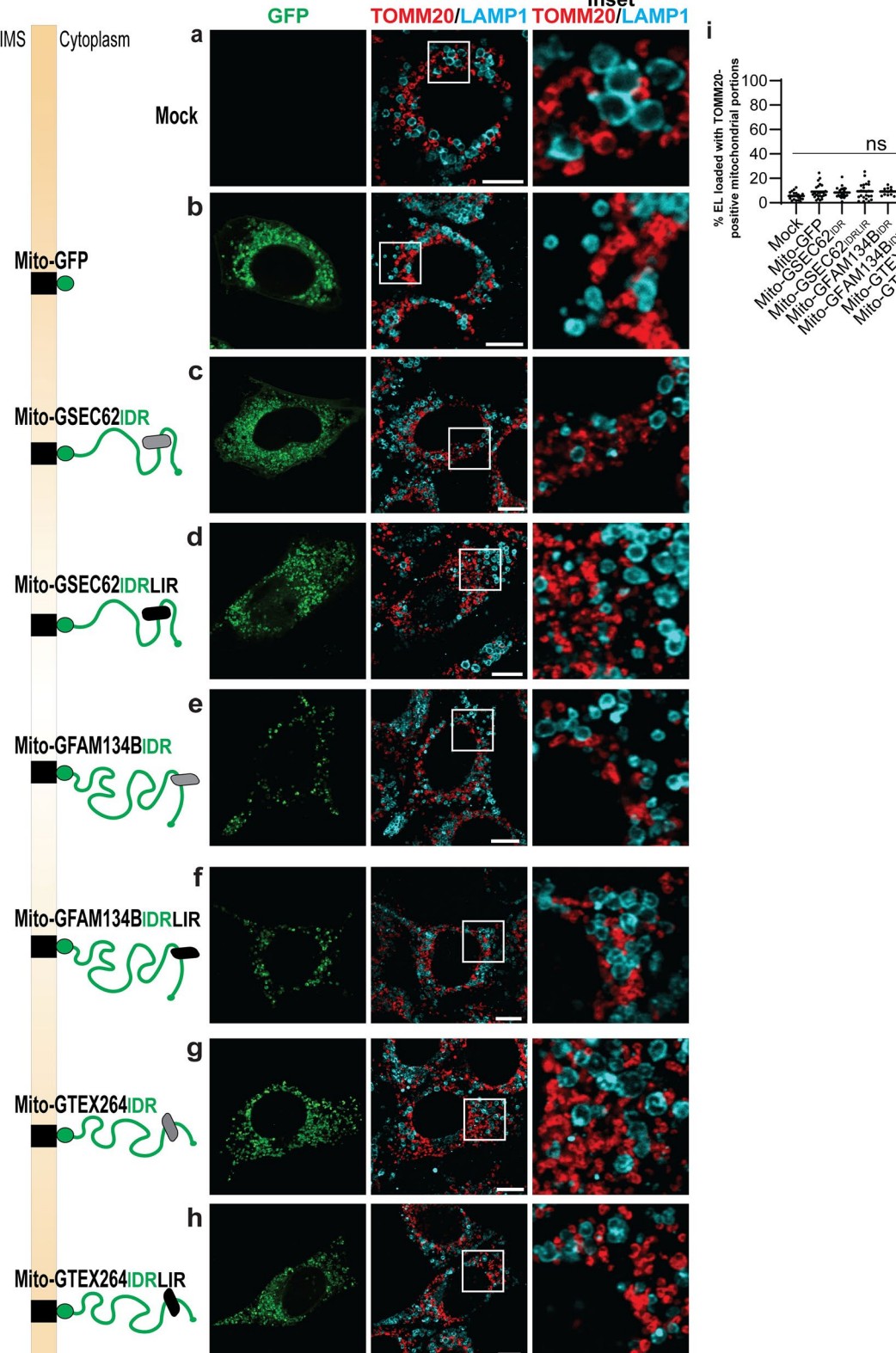

**Extended Data Fig. 4 | Monitoring mitophagy in ATG7 KO MEF expressing the ER-phagy receptors IDR modules.** ATG7KO MEF were exposed to BafA1 to preserve mitochondria (labeled with TOMM20, red) possibly transported within LAMP1-positive degradative endolysosomes (cyan) and imaged in CLSM. The absence of mitochondria in the lumen of the endolysosomes certifies absence of detectable mitophagy. Scale bar: 10 μm, inset magnification: 4x. (**b**) Same as the CLSM image in (**a**) for cells expressing Mito-GFP at the OMM. (**c**) same as (**b**) for cells expressing Mito-GSEC62$_{IDR}$. (**d**) Same as (**c**) for cells expressing Mito-GSEC62$_{IDRLIR}$. (**e**) same as (**c**) for Mito-GFAM134B$_{IDR}$. (**f**) Same as (**d**) for Mito-GFAM134B$_{IDRLIR}$. (**g-h**) Same as (**c-d**) for Mito-GTEX264$_{IDR}$ and Mito-GTEX264$_{IDRLIR}$. (**i**) LysoQuant quantification of the percentage of cellular endolysosomes that are degrading TOMM20-positive mitochondria fragments in panels **A-H** (N = 3 biological replicates for **a-d**, N = 1 for **e-h**, n = 31, 19, 21, 29, 37, 26, 28, and 29 cells for **a-h**, respectively). Ordinary one-way ANOVA with Turkey's multiple comparisons test, F = 1.4. Mean bar is shown. Adjusted P value: ns (P = 0.2070), not significant.

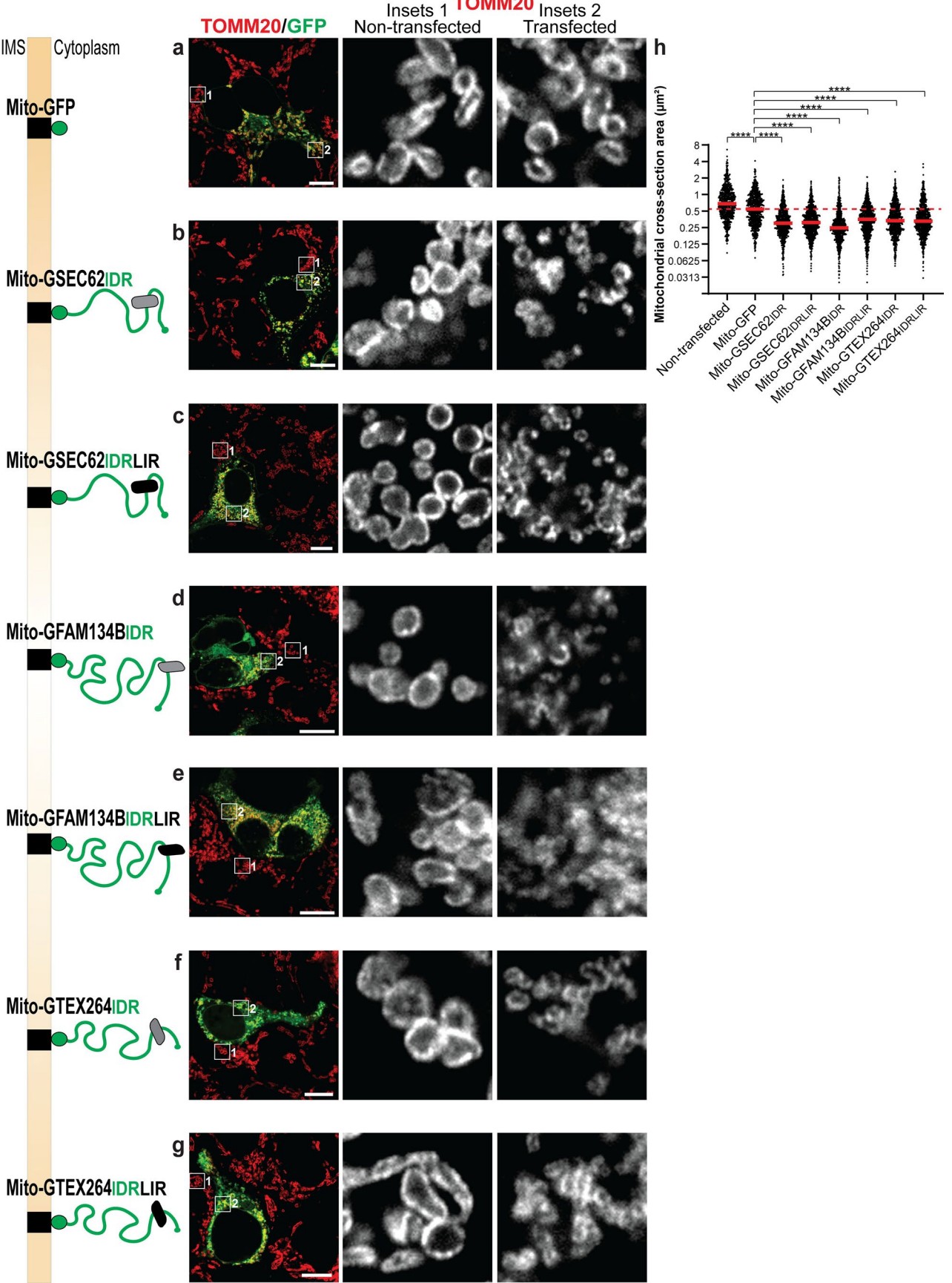

**Extended Data Fig. 5 | See next page for caption.**

**Extended Data Fig. 5 | Transplant of ER-phagy receptors IDR modules onto OMM triggers mitochondria fragmentation in HEK293 cells. (a)** The representative CLSM micrograph shows one cell expressing sfGFP at the OMM (Mito-GFP, green in the panel on the left), and non-transfected cells, whose mitochondria are labeled with TOMM20 (red). Inset 1: mitochondrial structure in a non-transfected cell; inset 2: mitochondrial structure in a cell expressing Mito-GFP. Scale bar: 10 μm, inset magnification: 10x. **(b)** same as **(a)** for a representative cell that expresses Mito-GSEC62$_{IDR}$. **(c)** Same as **(b)** for a representative cell that expresses Mito-GSEC62$_{IDRLIR}$, which does not engage LC3.

**(d)** same as **(b)** for Mito-GFAM134B$_{IDR}$. **(e)** Same as **(c)** for Mito-GFAM134B$_{IDRLIR}$. **(f-g)** Same as **(b-c)** for Mito-GTEX264$_{IDR}$ and Mito-GTEX264$_{IDRLIR}$. **(h)** Quantification of mitochondrial cross-section area of HEK293 cells (n = 708, 787, 766, 1007, 911, 1128, 924, and 994 mitochondria for non-transfected cells and transfected cells in **a-g**, respectively), measured in CLSM. Kruskal-Wallis test with Dunn's multiple comparisons test, Kruskal-Wallis statistic=1152. Median bar (red) is shown, y axis is Log2. N = 3 biological replicates. Adjusted P value: ****P < 0.0001.

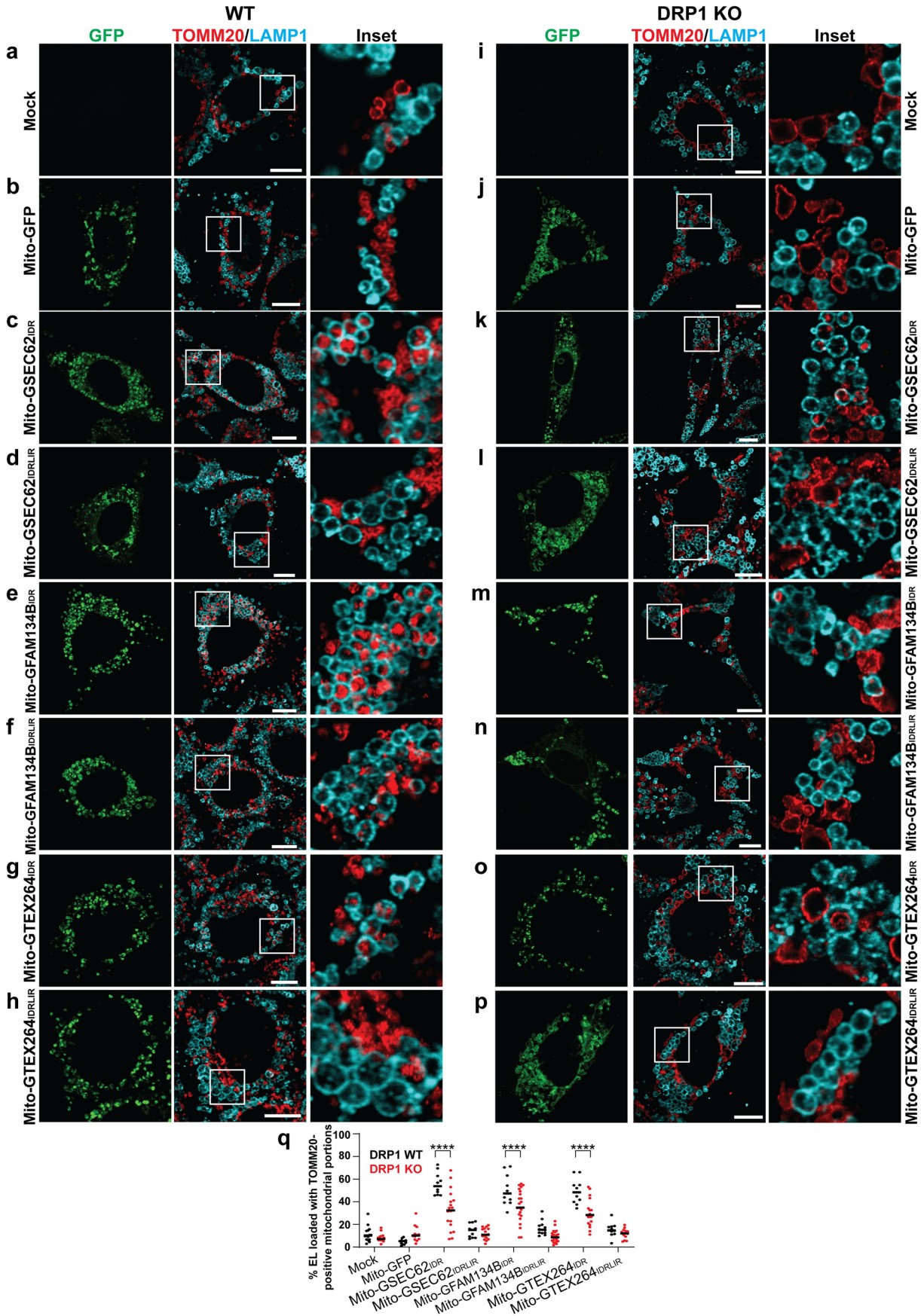

**Extended Data Fig. 6 | See next page for caption.**

**Extended Data Fig. 6 | Monitoring mitophagy in DRP1 KO MEF expressing the ER-phagy receptors IDR modules.** (**a**) WT MEF were exposed to BafA1 to preserve mitochondria (labeled with TOMM20, red) possibly transported within LAMP1-positive degradative endolysosomes (cyan). The absence of mitochondria in the lumen of the endolysosomes certifies absence of detectable mitophagy. Scale bar: 10 μm, inset magnification: 4x. (**b**) Same as (**a**) for cells expressing Mito-GFP at the OMM. (**c**) same as (**b**) for cells expressing Mito-GSEC62$_{IDR}$. (**d**) Same as (**c**) for cells expressing Mito-GSEC62$_{IDRLIR}$. (**e**) same as (**c**) for Mito-GFAM134B$_{IDR}$. (**f**) Same as (**d**) for Mito-GFAM134B$_{IDRLIR}$. (**g**) Same as (**c**) for Mito-GTEX264$_{IDR}$. (**h**) Same as (**d**) for Mito-GTEX264$_{IDRLIR}$. (**i-p**) Same as (**a-h**) for DRP1 KO MEF cells, showing reduced delivery of TOMM20 to endolysosomes. (**q**) LysoQuant quantification of the percentage of cellular endolysosomes that are degrading TOMM20-positive mitochondria fragments in panels **a-h** (black dots) and **i-p** (red dots). (N = 2 biological replicates, n = 15, 11, 10, 11, 11, 12, 10, and 11 cells for panels (**a-h**), n = 18, 12, 19, 16, 23, 21, 19, and 13 cells for (**i-p**), respectively). Two-way ANOVA with Turkey's multiple comparisons test. Mean bar is shown. Adjusted P value: **** $p < 0.0001$.

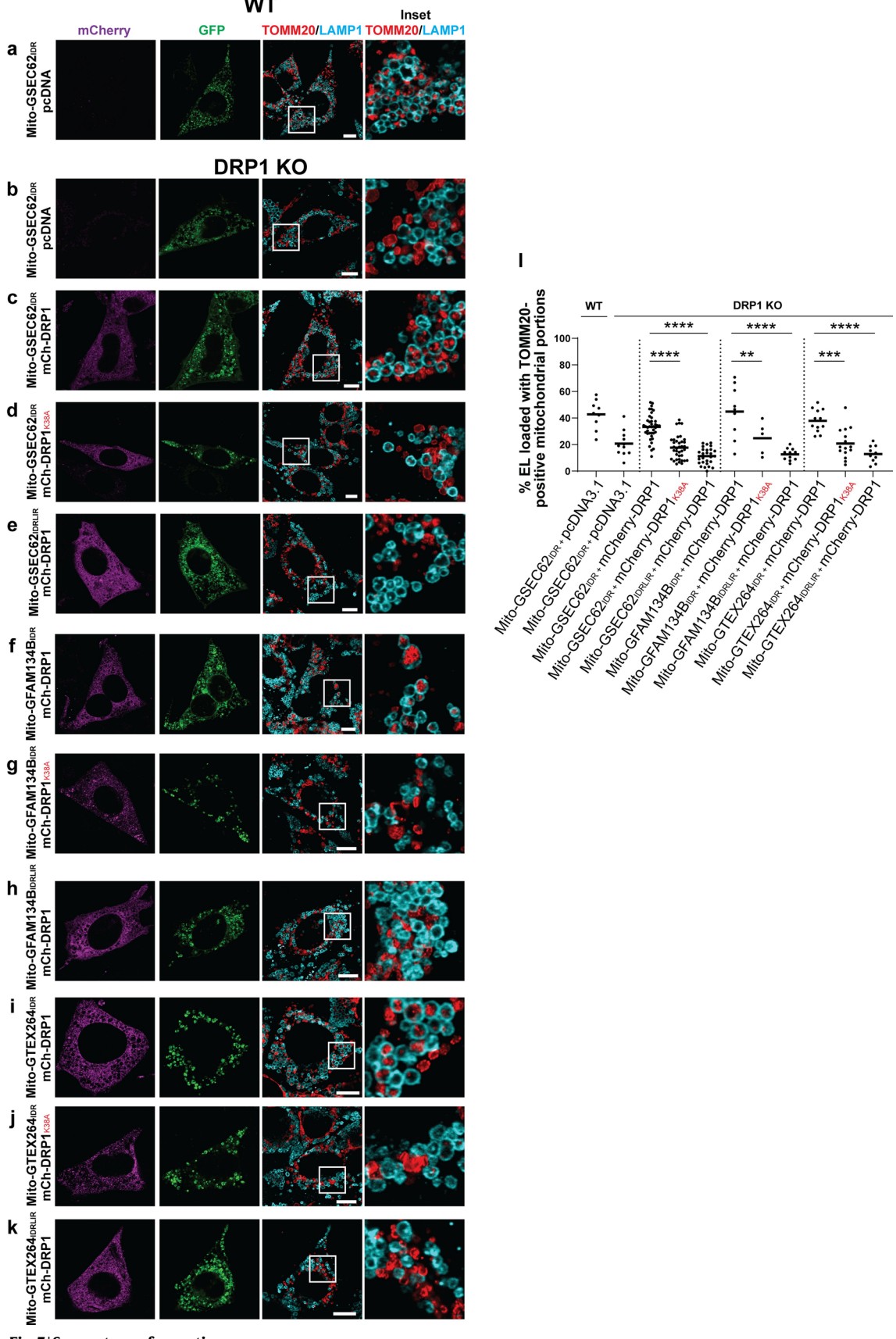

**Extended Data Fig. 7 | See next page for caption.**

**Extended Data Fig. 7 | Reconstitution of DRP1 KO cells with GTPase-active WT DRP1 rescues IDR-induced mitophagy.** (**a**) WT MEF were exposed to BafA1 to preserve mitochondria (labeled with TOMM20, red) possibly transported within LAMP1-positive degradative endolysosomes (cyan). Mitochondria were delivered into the lumen of the endolysosomes in the absence of ectopically expressed DRP1. Scale bar: 10 μm, inset magnification: 4x. (**b**) DRP1 KO MEF cells transfected with Mito-GSEC62$_{IDR}$ show reduced delivery of mitochondria into the lumen of LAMP1-positive endolysosomes, demonstrating reduced mitophagy. (**c**) Co-transfection of mCherry-DRP1 construct rescues mitophagy induced by Mito-GSEC62$_{IDR}$ as shown by the increased prevalence of TOMM20-positive mitochondria inside LAMP1-positive endolysosomes. (**d**) GTPase-inactive mutant DRP1 K38A fails to rescue mitophagy inhibition in DRP1 KO MEF cells. (**e**) Same as (**c**) for Mito-GSEC62$_{IDRLIR}$, showing no induction of mitophagy. (**f-h**) Same as (**c-e**) for Mito-GFAM134B$_{IDR}$ chimeras. (**i-k**) Same as (**c-e**) for Mito-GTEX264$_{IDR}$ chimeras. (**l**) LysoQuant quantification of the percentage of cellular endolysosomes that are degrading TOMM20-positive mitochondria fragments in panels (**a-k**) (N = 3 biological replicates for panels **c-e**, N = 2 for **a-b** and **f-k**, n = 9, 11, 40, 37, 28, 9, 5, 11, 12, 15, and 10 cells for panels **a-k**, respectively). Ordinary one-way ANOVA with Turkey's multiple comparisons test. F = 22.48. Mean bar is shown. Adjusted P value: **** p < 0.0001, ***p = 0.0003, **p = 0082.

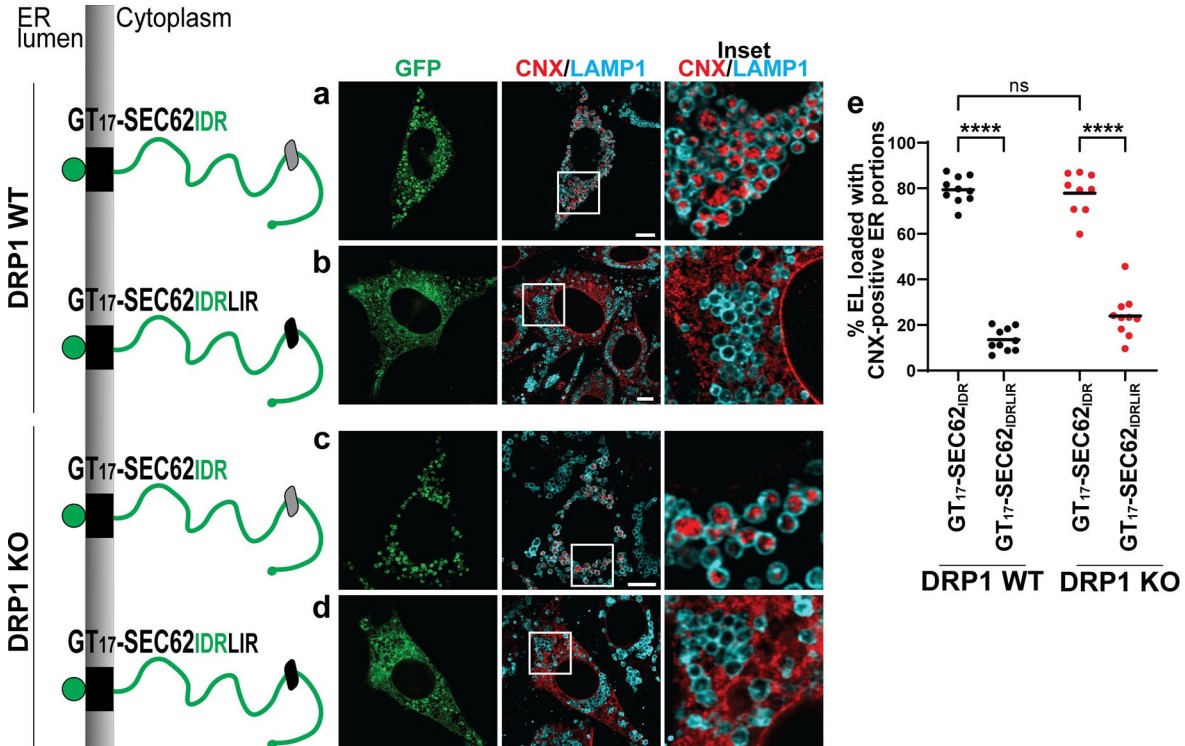

**Extended Data Fig. 8 | Knock outs of DRP1 has no effect on the ER-phagy induced by the IDR chimeras.** (**a-b**) WT and (**c-d**) DRP1 KO MEF were exposed to BafA1 to preserve ER portions (labeled with CNX, red) possibly transported within LAMP1-positive degradative endolysosomes (cyan). Scale bar: 10 μm, inset magnification: 4x. (**a**) WT MEF cells transfected with GT$_{17}$-SEC62$_{IDR}$ show efficient delivery of ER portions into the lumen of LAMP1-positive endolysosomes (**b**) Same for cells expressing GT$_{17}$-SEC62$_{IDRLIR}$, demonstrating no induction of

ER-phagy. (**c-d**) Same as (**b-c**) for DRP1 KO MEFs expressing SEC62 IDR chimeras. ER portions were efficiently delivered into the lumen of the endolysosomes in the absence of endogenous DRP1. (**e**) Lysoquant quantification of the percentage of cellular endolysosomes that are degrading CNX-positive ER fragments in panels (**a-d**) (N = 2 biological replicates, n = 10, 10, 10, and 9 cells). Two-way ANOVA with Turkey's multiple comparisons test. Adjusted ****P < 0.0001, ns (P = 0.9733), not significant. Mean bar is shown.

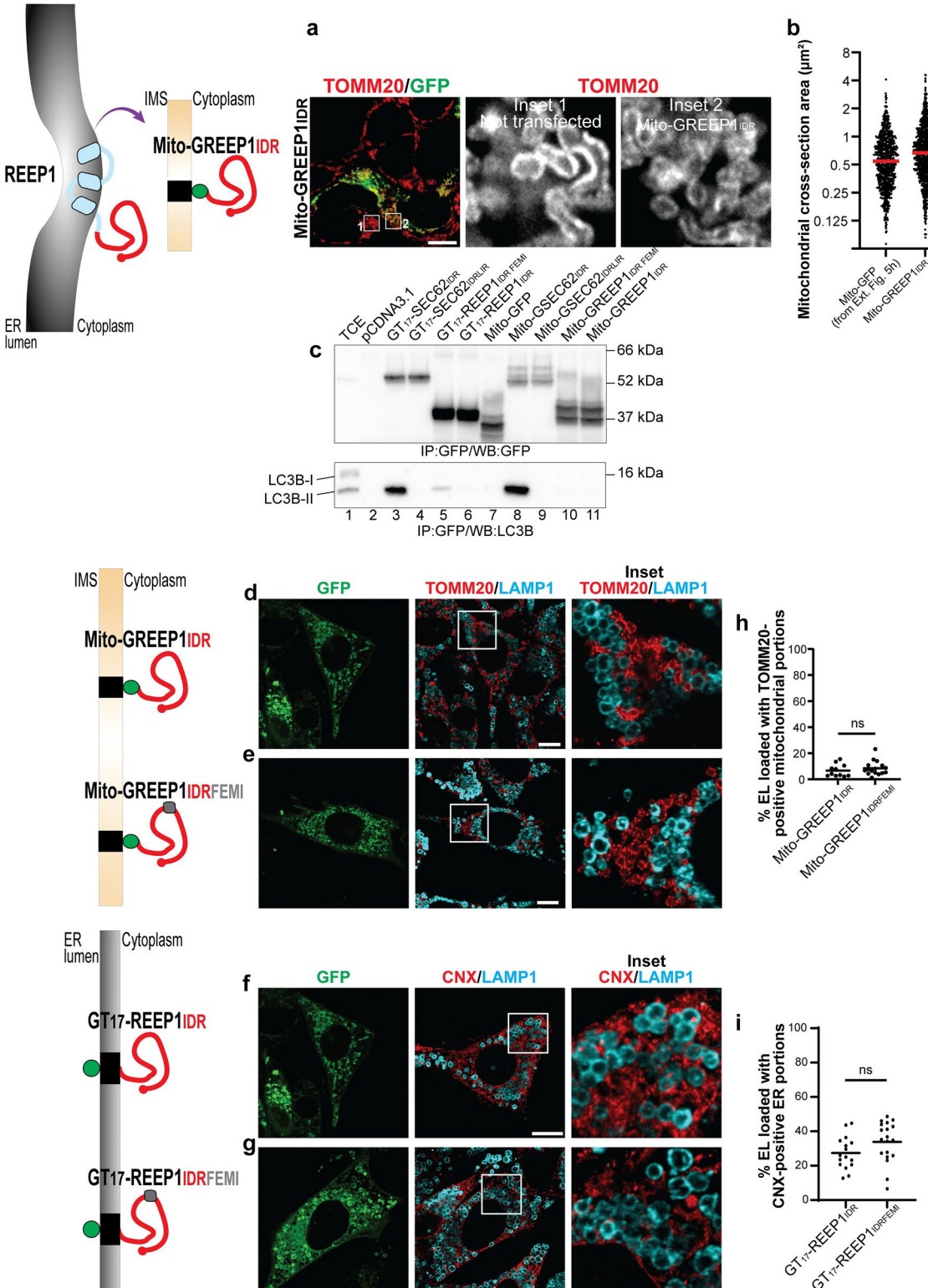

**Extended Data Fig. 9 | See next page for caption.**

**Extended Data Fig. 9 | Non-autophagy IDR does not induce organelle fragmentation nor organellophagy.** (**a**) Schematic of Mito-GREEP1$_{IDR}$ construct targeting the OMM with the transmembrane domain of the mitochondrial protein TOMM20 (green in the panel on the left). The representative CLSM micrograph shows a HEK293 cell expressing Mito-GREEP1$_{IDR}$ at the OMM (green) and non-transfected cells, whose mitochondria are labeled with TOMM20 (red). Inset 1: mitochondrial structure in a non-transfected cell; inset 2: mitochondrial structure in a cell expressing Mito-GREEP1$_{IDR}$. Scale bar: 10 μm, inset magnification: 10x. (**b**) Quantification of mitochondrial cross-section area in (**a**) (n = 787 mitochondria), compared with Mito-GFP-expressing cells (Extended Data Fig. 5a, Extended Data Fig. 5h, n = 1120 mitochondria) showing that the Mito-GREEP1$_{IDR}$ construct does not induce mitochondrial fragmentation. (**c**) IP-WB showing bait level (upper panel) and engagement of LC3B (lower panel) by ER-targeted (lanes 3-6) or mitochondria-targeted (lanes 8-11) IDRs of SEC62 and REEP1 expressed in wild type MEF. N = 3 biological replicates for lanes 1-4 and 7-9, N = 1 for lanes 5-6 and 10–11. (**d**) MEF cells expressing the Mito-GREEP1$_{IDR}$ were exposed to BafA1 to preserve mitochondria fragments (labeled with TOMM20, red) possibly transported within LAMP1-positive degradative endolysosomes (cyan). The representative CLSM micrograph shows a cell expressing Mito-GREEP1$_{IDR}$ at the OMM (green). Scale bar: 10 μm, inset magnification: 4x. The absence of mitochondrial portions within LAMP1-positive ELs indicates the absence of mitophagy. (**e**) Same as (**d**) for cells expressing Mito-GREEP1$_{IDRFEMI}$ construct equipped with the LIR of SEC62. (**h**) LysoQuant quantification of (**d-e**) (N = 1 biological replicate, n = 11 and 16 cells for panels **d** and **e**, respectively). Unpaired two-tailed t test. Mean bar is shown. P-value = 0.3691, ns, not significant (p = 0.3691). (**f-i**) Same as (**d-h**) for GT$_{17}$-REEP1$_{IDR}$ and GT$_{17}$-REEP1$_{IDRFEMI}$, showing no delivery of the ER marker CNX (red). N = 1 biological replicate, n = 16 and 19 cells for panels **f** and **g**, respectively. Unpaired two-tailed t test. Mean bar is shown. P-value = 0.0863, ns, not significant.

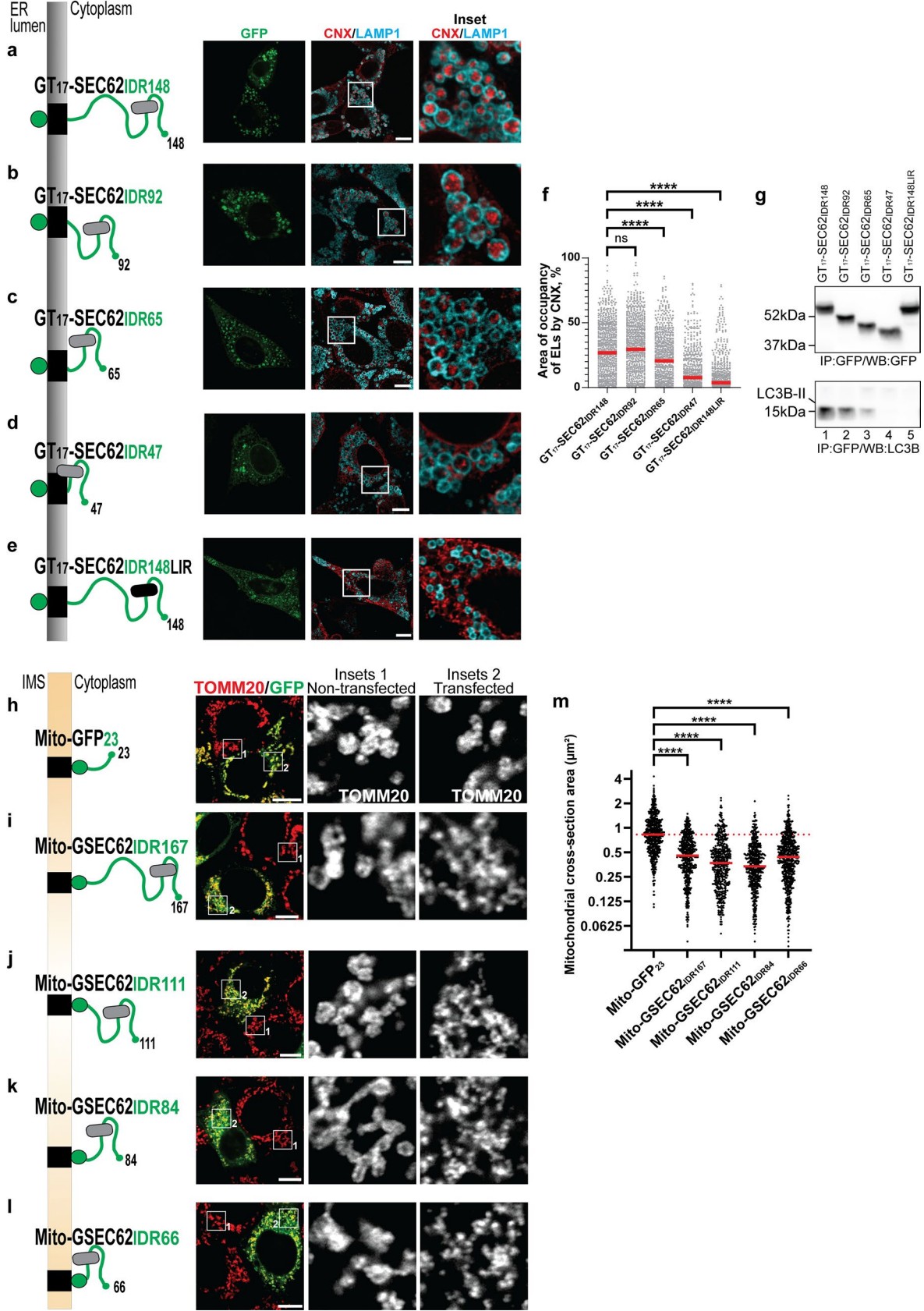

Extended Data Fig. 10 | See next page for caption.

**Extended Data Fig. 10 | ER-phagy and Mitochondrial fragmentation induced by IDR chimeras of different lengths. (a)** WT MEF were exposed to BafA1 to preserve ER portions (labeled with CNX, red) possibly transported within LAMP1-positive degradative endolysosomes (cyan). Scale bar: 10 μm, inset magnification: 4x. (**a**) WT MEF cells transfected with $GT_{17}$-$SEC62_{IDR148}$ (GFP, green) show efficient delivery of ER portions into the lumen of LAMP1-positive endolysosomes (**b-d**) Same for cells expressing truncated $GT_{17}$-$SEC62_{IDR}$ chimeras. (**e**) Same as (**a**) for cells expressing $GT_{17}$-$SEC62_{IDR148LIR}$, demonstrating no induction of ER-phagy. (**f**) Lysoquant quantification of the area of LAMP1-positive EL occupied by CNX in MEF from (**a-e**) (each dot represents a single EL, n = 811, 831, 739, 629, and 615 endolysosomes analyzed in cells from (**a-e**) respectively, N = 2 biological replicates). Kruskal-Wallis ANOVA test with Dunn's multiple comparison test, Kruskal-Wallis statistic=521.3. Adjusted ****P < 0.0001, ns: not significant (P = 0.8932). Median bar is shown. (**g**) Association of endogenous LC3B-II with $GT_{17}$-$SEC62_{IDR}$ chimeras in MEF treated with 50 nM BafA1. Immunocomplexes were revealed with anti-GFP (upper panel) or anti-LC3B antibodies (lower panel). N = 3 biological replicates. (**h**) The representative CLSM micrographs showing one ATG7 KO MEF cell expressing Mito-GFP construct exposing a 23 residues long linker at the OMM (Mito-$GFP_{23}$, green in the panel on the left) that does not induce mitochondrial fragmentation, and non-transfected cells, whose mitochondria are labeled with TOMM20 (red). Inset 1: mitochondrial structure in a non-transfected cell; Inset 2: mitochondrial structure in a cell expressing Mito-$GFP_{23}$. Scale bar: 10 μm, inset magnification: 6x. (**i-l**) Same as (**a**) for truncated Mito-$GSEC62_{IDR}$ chimeras that show mitochondria fragments in transfected cells. (**m**) Quantification of mitochondrial cross-section area of ATG7 KO cells (n = 505, 478, 453, 505, and 623 mitochondria for cells in **g-k**, respectively), measured in CLSM. Kruskal-Wallis test with Dunn's multiple comparisons test, Kruskal-Wallis statistic=502. 4. Median bar (red) is shown, y axis is Log2. N = 1 biological replicate. Adjusted P value: ****P < 0.0001.

# Reporting Summary

## Statistics

For all statistical analyses, confirm that the following items are present in the figure legend, table legend, main text, or Methods section.

| n/a | Confirmed | |
|---|---|---|
| ☐ | ☒ | The exact sample size (*n*) for each experimental group/condition, given as a discrete number and unit of measurement |
| ☐ | ☒ | A statement on whether measurements were taken from distinct samples or whether the same sample was measured repeatedly |
| ☐ | ☒ | The statistical test(s) used AND whether they are one- or two-sided *Only common tests should be described solely by name; describe more complex techniques in the Methods section.* |
| ☒ | ☐ | A description of all covariates tested |
| ☐ | ☒ | A description of any assumptions or corrections, such as tests of normality and adjustment for multiple comparisons |
| ☐ | ☒ | A full description of the statistical parameters including central tendency (e.g. means) or other basic estimates (e.g. regression coefficient) AND variation (e.g. standard deviation) or associated estimates of uncertainty (e.g. confidence intervals) |
| ☐ | ☒ | For null hypothesis testing, the test statistic (e.g. *F*, *t*, *r*) with confidence intervals, effect sizes, degrees of freedom and *P* value noted *Give P values as exact values whenever suitable.* |
| ☒ | ☐ | For Bayesian analysis, information on the choice of priors and Markov chain Monte Carlo settings |
| ☒ | ☐ | For hierarchical and complex designs, identification of the appropriate level for tests and full reporting of outcomes |
| ☒ | ☐ | Estimates of effect sizes (e.g. Cohen's *d*, Pearson's *r*), indicating how they were calculated |

*Our web collection on statistics for biologists contains articles on many of the points above.*

## Software and code

Policy information about availability of computer code

| Data collection | Immunoflurescence data: Leica TCS SP5 confocal system and Leica Stellaris SP8 microscope with Leica LAS X 4.5.0.025531<br>Western blotting data: FusionFX7 VILBER (Witec) with Fusion FX7 Edge 18.12 software<br>Fluorescent gels: Amersham Typhoon scanner (Cytiva)<br>Immunogold electron microscopy: Transmission Electron Microscope Talos L120C (FEI, Thermo Fisher Scientific) with a Ceta CCD camera (FEI, Thermo Fisher Scientific), Velox 3.6.0 software (FEI, Thermo Fisher Scientific).<br>RT-Electron Tomography : Leica Stellaris SP8 Leica LAS X 4.5.0.025531 software, Transmission Electron Microscope Talos L120C (FEI, Thermo Fisher Scientific) with Tomography 5 software (FEI, Thermo Fisher Scientific).<br>LC/MS: Vanquish Neo nanoHPLC system interfaced via a nanospray Flex source to a high resolution Orbitrap Exploris 480 mass spectrometer (Thermo Fisher Scientific) and Xcalibur software (Tune 2.9, Thermo Fisher Scientific) |
|---|---|
| Data analysis | MaxQuant 2.1.4.0<br>ImageJ 2.16.0/1.54f for windows64<br>LysoQuant plugin for ImageJ (an unbiased and automated deep learning tool for fluorescent image quantification, which is freely available (https://www.irb.usi.ch/lysoquant/))<br>IMOD 4.11.24<br>Microscopy Image Browser (MIB) 2.84<br>Graphing and statistic analysis were performed using GraphPad PRISM 10 (10.1.2 for Windows)<br>Photoshop 26.2.0 |

For manuscripts utilizing custom algorithms or software that are central to the research but not yet described in published literature, software must be made available to editors and reviewers. We strongly encourage code deposition in a community repository (e.g. GitHub). See the Nature Portfolio guidelines for submitting code & software for further information.

Matlab R2023b Update 5 23.2.0.24591.
Data analysis for proteomics data was performed in Perseus v2.0.5

For manuscripts utilizing custom algorithms or software that are central to the research but not yet described in published literature, software must be made available to editors and reviewers. We strongly encourage code deposition in a community repository (e.g. GitHub). See the Nature Portfolio guidelines for submitting code & software for further information.

## Data

Policy information about availability of data

All manuscripts must include a data availability statement. This statement should provide the following information, where applicable:
- Accession codes, unique identifiers, or web links for publicly available datasets
- A description of any restrictions on data availability
- For clinical datasets or third party data, please ensure that the statement adheres to our policy

The MS proteomics data supporting the data in Ext. Fig. 1f and Ext. Fig. 1g can be accessed from the ProteomeXchange Consortium via the PRIDE partner repository under accession code PXD060519. Numerical data used to create graphs is included as Source data tables. Uncropped blots and gels are included in Image source data. No restrictions on data availability apply.

## Research involving human participants, their data, or biological material

Policy information about studies with human participants or human data. See also policy information about sex, gender (identity/presentation), and sexual orientation and race, ethnicity and racism.

| | |
|---|---|
| Reporting on sex and gender | N/A |
| Reporting on race, ethnicity, or other socially relevant groupings | N/A |
| Population characteristics | N/A |
| Recruitment | N/A |
| Ethics oversight | N/A |

Note that full information on the approval of the study protocol must also be provided in the manuscript.

# Field-specific reporting

Please select the one below that is the best fit for your research. If you are not sure, read the appropriate sections before making your selection.

☒ Life sciences ☐ Behavioural & social sciences ☐ Ecological, evolutionary & environmental sciences

For a reference copy of the document with all sections, see nature.com/documents/nr-reporting-summary-flat.pdf

# Life sciences study design

All studies must disclose on these points even when the disclosure is negative.

| | |
|---|---|
| Sample size | No statistical methods were used to pre-determine sample sizes but our sample sizes are similar to those reported in previous publications (DOIs: 10.15252/embj.2020107240; 10.15252/embj.201899259; 10.1091/mbc.E20-04-0269; 10.1038/s41467-023-39172-3; 10.1091/mbc.E21-10-0526). |
| Data exclusions | No data were excluded from the analyses. |
| Replication | The number of cells analyzed and the number of independent experiments is specified for each experiment in the figure legends. |
| Randomization | No randomization was performed as this is not common for western blot or microscopy analysis. Covariates were controlled with the corresponding negative controls. |
| Blinding | Unbiased imaging data collection/analyses were performed by at least two different scientists, one of which was blind to the identity of the sample. Morphological analyses of mitochondrial size were performed by the scientists blinded to the identity of the sample. The fluorescent image quantifications of ER and mitochondria delivery to endolysosomes were performed using LysoQuant, an unbiased and automated deep learning tool. Acquisition of MS data was done by scientist blinded to the nature of the sample and the scope of the experiment. For WB and IP analyses, conclusions were drawn based on qualitative presence/absence of band signal, therefore, blinding is not relevant. For electron microscopy, acquisition of samples was performed by a scientist blinded to the nature of the sample. |

# Reporting for specific materials, systems and methods

We require information from authors about some types of materials, experimental systems and methods used in many studies. Here, indicate whether each material, system or method listed is relevant to your study. If you are not sure if a list item applies to your research, read the appropriate section before selecting a response.

## Materials & experimental systems

| n/a | Involved in the study |
|-----|-----------------------|
| ☐ | ☒ Antibodies |
| ☐ | ☒ Eukaryotic cell lines |
| ☒ | ☐ Palaeontology and archaeology |
| ☒ | ☐ Animals and other organisms |
| ☒ | ☐ Clinical data |
| ☒ | ☐ Dual use research of concern |
| ☒ | ☐ Plants |

## Methods

| n/a | Involved in the study |
|-----|-----------------------|
| ☒ | ☐ ChIP-seq |
| ☒ | ☐ Flow cytometry |
| ☒ | ☐ MRI-based neuroimaging |

## Antibodies

| | |
|---|---|
| Antibodies used | Rat anti-LAMP1 DSHB 1D4B 1:50 (CLSM)<br>Rabbit anti-TOMM20 Abcam ab186734 1:100 (CLSM)<br>Rabbit anti-LC3B Sigma L7543 1:1000 (IB)<br>Mouse anti-LAMP1 DSHB H4A3 1:100 (CLSM)<br>Rabbit Anti-ATG7 Sigma A2856 1:600 (IB)<br>Rabbit Anti-DRP1 Abcam ab184247 1:1000 (IB)<br>Rabbit anti-GFP Abcam ab290 1:50 (IEM) 1:1500 (IB)<br>Mouse anti-GAPDH Millipore MAB374 clone C 1:30000 (IB)<br>Protein A HRP-conjugated Invitrogen 101023 1:20000 (IB)<br>Goat anti-rabbit AlexaFluor488-conjugated Thermo Fisher Scientific A-21206 1:300 (CLSM)<br>Goat anti-rabbit AlexaFluor568-conjugated Thermo Fisher Scientific A-11036 1:300 (CLSM)<br>Goat anti-rat AlexaFluor647-conjugated Thermo Fisher Scientific A-21247 1:300 (CLSM)<br>Goat anti-rabbit AlexaFluor405-conjugated Thermo Fisher Scientific A-31556 1:150 (CLSM)<br>Goat anti-rabbit gold-labelled Nanoprobes 2004 1:100 (IEM)<br>Rabbit anti-CNX Kind gift from A. Helenius Not applicable 1:100 (CLSM) |
| Validation | Anti-ATG7 and anti-DRP1 antibodies were validated with western blots using KO cell lines. Anti-LC3B antibody in WB by observing 2 characteristic bands corresponding to the lipidated and non-lipidated form at the corresponding MWs. Primary Anti-GFP antibody was validated in overexpression experiments using sfGFP-tagged proteins. Anti-TOMM20 antibody was validated by CLSM by observing the signal colocalizing with mitochondria (MitoTracker). Anti-CNX, anti-TOMM20 and anti-LAMP1 antibodies were validated by CLSM by colocalization with the Endoplamsic Reticulum, mitochondria (MitoTracker) and endolysosomes, respectively. Anti-GAPDH antibody was validated by WB with the only band visible corresponding to the expected MW. |

## Eukaryotic cell lines

Policy information about cell lines and Sex and Gender in Research

| | |
|---|---|
| Cell line source(s) | MEF WT and Atg7KO are kind gift from M Kumatsu, HEK293 were purchased from ATCC (CRL-1573). WT and DRP1KO MEFs are kind gift from S. Manley and M. Ryan |
| Authentication | ATG7KO were checked for absence of ATG7 and for lack of LC3 lipidation (e.g., Komatsu, M. et al. JCB 2005; Fumagalli, F. et al. Nature Cell Biol 2016; Fasana, E. et al EMBO Rep. 2024). DRP1KO MEFs were checked for abscence of DRP1 and enlarged mitochondria. WT MEF cells resulted positive for DRP1 and ATG7, displayed classical fibroblast morphology and were positive for LAMP1 staining with mouse-specific LAMP1 antibody. HEK293 were obtained from ATCC, and their identity was confirmed by morphology. |
| Mycoplasma contamination | The cell lines were tested regularly to be negative for mycoplasma contamination. |
| Commonly misidentified lines<br>(See ICLAC register) | Commonly misidentified lines were NOT used in this study. |

## Plants

Seed stocks

N/A

Novel plant genotypes

N/A

Authentication

N/A

