## [Peer Review File · Nature Cell Biology]

The intrinsically disordered regions of organellophagy receptors are interchangeable and control organelle fragmentation, ER-phagy and mitophagy flux

Corresponding Author: Professor Maurizio Molinari

Version 0:

Decision Letter:

Dear Mauri,

Thank you again for submitting your manuscript to the journal and please accept our sincere apologies for the long delay before we could share our decision with you, as one reviewer had requested more time.

Your manuscript "Intrinsically disordered modules drive organelle fragmentation and autophagy", has now been seen by 3 referees, who are experts in ER-phagy (Referee #1); autophagy (Referee #2); and biomolecular condensate biology (Referee #3), and whose comments are pasted below. In light of their advice, we regret that we cannot offer to publish the study in Nature Cell Biology.

As you will see, although the reviewers found this work interesting, they raised serious concerns that question the strength of the data and of the novel conclusions that can be drawn at this stage. In particular, they shared concerns about the generality of the mechanism for other forms of organellophagy, asked for more information about the features of the IDRs that are functionally involved, asked for more evidence in support of the mechanism, and were concerned about the use of over-expression. We discussed their concerns editorially and regrettably find that they are too significant to move forward with the manuscript at the journal.

Although we cannot publish your paper, it may be appropriate for another journal in the Nature Portfolio. If you wish to explore the journals and transfer your manuscript please use our manuscript transfer portal. You will not have to re-supply manuscript metadata and files, unless you wish to make modifications. For more information, please see our [manuscript transfer FAQ](http://www.nature.com/authors/author_resources/transfer_manuscripts.html?WT.mc_id=EMI_NPG_1511_AUTHORTRANSF&WT.ec_id=AUTHOR) page.

We are very sorry that we could not be more positive on this occasion, but we thank you for the opportunity to consider this work. We also hope you find the reviews useful as you determine the next steps for the manuscript.

With kind regards,
Melina

Melina Casadio, PhD
Senior Editor, Nature Cell Biology
ORCID ID: <https://orcid.org/0000-0003-2389-2243>

Reviewers' comments:

Reviewer #1 (Remarks to the Author):

Rudinsky et al present an elegant dataset that probes the mechanism of action of ER-phagy receptors in fragmenting ER, prior to inclusion into autophagolysosomes. Their data suggest that the intrinsically-disordered regions of these receptors, the function of which was previously unknown, drive fragmentation (whereas transmembrane domains mainly drive localisation to particular subregions of this organelle, e.g. tubules versus sheets). LC3-binding sites embedded within the intrinsically-disordered regions are required only at a late step for inclusion into autophagosomes. Finally, the authors suggest that this mechanism is generalisable to other selective autophagy pathways and their cognate receptors. Thus, overall, these data suggest a new paradigm for how clearance of ER (and potentially other membrane bound organelles) is

targeted for clearance. This is a novel and important finding as it provides a unifying mechanism for ER-phagy (and indeed, potentially, selective autophagy of other membrane bound organelles). The manuscript is well-written and referenced. However, the scope of the present study is slightly limited in regard of the latter-mentioned generalisability of this phenomenon to other organelles. Furthermore, some further controls/refinements may be required in a limited number of places to strengthen the data already presented (although these should not present great difficulty).

MAJOR POINTS

The authors indicate that additional organelles, other than the ER, may depend upon IDR-containing receptors for membrane fragmentation during initial stages of autophagy. This is established using a chimeric approach using model targeting domains and studying action on mitochondria. Ideally, it should be established whether bona fide receptor(s) for organelles other than the ER utilise this mode of action.

Figure 3 - Are IDRs that are not from ER-phagy receptors capable of driving ER-phagy? Are there any minimal/maximal lengths or other restrictions or criteria for the success of an IDR in driving membrane fragmentation? This information would be useful.

ADDITIONAL CONTROLS

Figure 1 - should ideally have some quantification of fragmentation.

Figure 1 – localisation of mutants versus full-length, wild-type should be addressed using the same tagging strategy (this could be done with colocalization by standard immunofluorescence) to properly enable comparison, in the same system, of the targeting specificity of the TM domain alone. This is recommended in case the additional IDR and any other sequence nuances the targeting (the authors should not rely solely on previous descriptions of the behaviour of the wild-type).

Figure 3 – luminal tagging is performed to study the minimal role of IDR(s) in targeting to autophagolysosomes, by attaching IDRs to a bulky GFP tag. The latter may have unappreciated multimerization or scaffolding effects, particularly considering the ER-phagy receptors studied have few or no luminal regions. The GT fusion construct alone, with no IDR, should also be studied as a negative control to consolidate the interpretation of the effects of the IDR (I appreciate that IDR-LIR mutants are studied but do not think these are a sufficient control).

Figure 3 – it is mentioned that IDRs attached to a generic ER anchor drive lysosomal targeting of both ER sheets and tubules (Supp Figures 1 and 2). This is used as further evidence that the TM domains of receptors mediate specific sub-organellar targeting. Perhaps better evidence would be swapping the IDR of receptors targeting sheets or tubules while retaining the same TM domain.

MINOR POINTS

Fig 1 – Volcano plots are illegible

Line 61 – What is meant by “tout court”?

Reviewer #2 (Remarks to the Author):

During organellophagy such as ER-phagy and mitophagy, it is necessary to fragment the organelles to the size smaller than that of autophagosomes for sequestration. Previously, RHD domains of some ER-phagy receptors were proposed to mediate this process; however, many ER-phagy/mitophagy receptors lack such domains and the mechanisms that cause organelle fragmentation during organellophagy remained largely unknown. In this manuscript, the authors studied the regions in ER-phagy receptors responsible for ER fragmentation and ER-phagy by overexpression and found that IDR, but not RHD, of ER-phagy receptors is sufficient for fragmentation of ER and ER-phagy. The authors then showed that LC3-binding LIR in the IDRs is necessary for ER-phagy, but not for ER fragmentation. Moreover, the authors showed that the transplantation of ER-phagy receptors IDRs at the mitochondrial membrane induced mitochondrial fragmentation and mitophagy. Based on these data, the authors concluded that membrane-exposed IDRs promote organelle fragmentation and organellophagy.

The mechanism of organellophagy is one of the hottest topics in the field of autophagy and the proposed concept of IDR-mediated fragmentation of organelles during organellophagy is novel and attractive. The data are straightforward and consistent, and the logic of the manuscript is easy to follow. On the other hand, this manuscript has serious shortcomings; it just looks at phenotypes when overexpressed and it is not clear how much IDR contributes to fragmentation of ER and mitochondria at physiological expression levels. It is essential to provide data on the role of IDRs at endogenous expression levels.

Major comments

1) All the provided data are obtained by overexpression. It is known that overexpression of proteins in mitochondria cause their fragmentation by stress. The obtained data shown here may simply indicate that overexpression of any IDRs on the surface of ER and mitochondria causes their fragmentation and may not indicate a physiological function for the IDR of the ER-phagy receptors. It is essential to study the role of IDRs at the physiological expression level.

2) It is necessary to perform control experiments. Use some IDRs derived from non-autophagic proteins (with length and charge similar to ER-phagy IDRs) for the same experiments and study their effect on ER-phagy, ER fragmentation, and mitochondrial fragmentation.

3) ER-phagy activity was only examined by fluorescence microscopy. It is essential that another additional assay be performed to robustly verify ER-phagy activity (e.g., HaloTag assay to monitor degradation of ER components (PMID 35938926)).

Reviewer #3 (Remarks to the Author):

This study focuses on the role of intrinsically-disordered regions (IDRs) within ER-phagy receptor proteins and the authors find that these regions control membrane fragmentation. The authors determine this through complementary light microscopy and electron microscopy approaches with a set of three ER-phagy receptor proteins in MEF cells. These three proteins include FAM134, SEC62, and TEX264. Through several interesting experiments using bafilomycin-inhibited or ATG7 knockout MEF cells and chimeric constructs, the authors demonstrate that the IDR segments of these ER-phagy receptor proteins drive membrane fragmentation. The LC3 interacting region (LIR) within these IDRs drive localization of ER or mitochondria delivery to lysosomes. The RHD (membrane-tethering domains) of these receptors determine sub-compartment distribution, as shown with chemical crosslinking and LC/MS proteomics analysis. Importantly, using engineered mitochondria-directing constructs, the authors also find that the IDRs of ER-phagy receptors can be used to program mitochondria membranes for mitochondria fragmentation. The insight of this story reveals a new function of IDRs in controlling membrane fragmentation; this is further buttressed by another preprint from a different group on a similar topic (preprint is mentioned by the authors in the discussion). The authors also employ additional IDR mutants containing the LC3 binding segment to determine the length of IDR required for coupled fragmentation-endolysosomal delivery. In general, the data appear to be well-presented, but the story is incomplete. Several questions abound regarding the IDRs as little is presented on these regions, yet they are the focus of this story. Little is presented regarding the mechanism by which these IDRs function to fragment membranes. Additionally, the study requires deposition of certain data.

To explain the origins of membrane fragmentation by the IDR segments of these proteins (FAM134, SEC62, and TEX264), work is required to compare these IDR sequences biophysically. What is unique about these amino acid sequences that drive fragmentation? The authors mention possible hypotheses within the text, and these should be tested. The authors should use other IDR segments (from other proteins) of similar biophysical properties (and/or length) and see whether they are able to perform similar fragmentation function. Are there specific mutations that are disease-linked to some of these IDR segments in ER-phagy receptor proteins? If so, the authors should try a mutation or two to make these results very impactful for the community.

Strangely, Figure S5 is presented within the discussion but this should be part of the results. The type of work presented in Figure S5 is important to better understand the function of the IDR region.

Are there any concerns with overexpressing these constructs into cells? The authors do not provide overall cell images (for example in Figure 3a,b,d,e,g,h) to highlight differences in cells with and without transfected constructs. Furthermore, one of these constructs should be introduced at endogenous expression levels to observe whether membrane fragmentation occurs.

The AP-LC/MS data presented must be deposited in a proteomics database for sharing – I did not see any information regarding this throughout the manuscript.

Akin to what is shown in Fig S5, gels are required to showcase expression levels of transfected constructs as well as to show the size of transfected constructs in cells for data in Figures 1-4.

Are the results presented for MEF cells representative in other cell types? This would be important to support the main claim of the paper. Some data are presented in HEK293 cells (only Figure S4).

Was there a reason for switching between GFP and HaloTag between Figure 3 and Figure S3?

**For Nature Portfolio general information and news for authors, see <http://npg.nature.com/authors>.

Version 1:

Decision Letter:

Dear Professor Molinari,

Thank you for your email asking us to reconsider our decision on your manuscript, "The intrinsically disordered regions of organellophagy receptors control organelle fragmentation, ER-phagy and mitophagy flux". We are always willing to hear the authors' perspective, but we must first prioritize decisions on new submissions. We appreciate your patience while we

considered this appeal.

I have now discussed your manuscript, the referees' comments and your rebuttal, in detail with my colleagues, and we would be willing to reconsider a revised manuscript provided the following issues can be addressed, and that nothing similar is accepted for publication at Nature Cell Biology or published elsewhere in the meantime.

In addition, please pay close attention to our guidelines on statistical and methodological reporting (listed below) as failure to do so may delay the reconsideration of the revised manuscript. In particular please provide:

On resubmission please provide the completed Editorial Policy Checklist (found here https://www.nature.com/documents/nr-editorial-policy-checklist.pdf), and Reporting Summary (found here https://www.nature.com/documents/nr-reporting-summary.pdf). This is essential for reconsideration of the manuscript and these documents will be available to editors and referees in the event of peer review. For more information see below. Please also ensure that the presentation of statistical information in the revised submission complies with Nature Cell Biology's statistical guidelines (see below).

Please use the link below to submit the complete manuscript files and include a point-by-point response to the complete reviewer comments, verbatim as provided in their reports.

Link Redacted

Please let us know how you wish to proceed and when we can expect your revised manuscript.

With kind regards,

Angela Parrish

Angela R Parrish, PhD
Locum Senior Editor
Nature Cell Biology

GUIDELINES FOR EXPERIMENTAL AND STATISTICAL REPORTING

REPORTING REQUIREMENTS – To improve the quality of methods and statistics reporting in our papers we have recently revised the reporting checklist we introduced in 2013. We are now asking all life sciences authors to complete two items: an Editorial Policy Checklist (found here https://www.nature.com/documents/nr-editorial-policy-checklist.pdf) that verifies compliance with all required editorial policies and a reporting summary (found here https://www.nature.com/documents/nr-reporting-summary.pdf) that collects information on experimental design and reagents. These documents are available to referees to aid the evaluation of the manuscript. Please note that these forms are dynamic 'smart pdfs' and must therefore be downloaded and completed in Adobe Reader. We will then flatten them for ease of use by the reviewers. If you would like to reference the guidance text as you complete the template, please access these flattened versions at http://www.nature.com/authors/policies/availability.html.

We strongly recommend the presentation of source data for graphical and statistical analyses as a separate Supplementary

Table, and request that source data for all independent repeats are provided when representative experiments of multiple independent repeats, or averages of two independent experiments are presented. This supplementary table should be in Excel format, with data for different figures provided as different sheets within a single Excel file. It should be labelled and numbered as one of the supplementary tables, titled "Statistics Source Data", and mentioned in all relevant figure legends.

Version 2:

Decision Letter:

*Please delete the link to your author homepage if you wish to forward this email to co-authors.

Dear Professor Molinari,

Your manuscript "The intrinsically disordered regions of organellophagy receptors control organelle fragmentation, ER-phagy and mitophagy flux", has now been seen by 3 referees, who are experts in ER-phagy (referee 1); selective autophagy and condensates (referee 2); and IDRs and condensates (referee 3). As you will see from their comments (attached below) they find this work of interest, but have raised some important points. Although we are also very interested in this study, we believe that their concerns should be addressed before we can consider publication in Nature Cell Biology.

Nature Cell Biology editors discuss the referee reports in detail within the editorial team, including the chief editor, to identify key referee points that should be addressed with priority, and requests that are overruled as being beyond the scope of the current study. To guide the scope of the revisions, I have listed these points below. We are committed to providing a fair and constructive peer-review process, so please feel free to contact me if you would like to discuss any of the referee comments further.

In particular, it would be essential to:

- 1) Please perform the requested panel of IDR controls as requested by Reviewer #2.
- 2) Please cite the recent paper "Intrinsically disordered region amplifies membrane remodeling to augment selective ER-phagy" (PMID: 39453744) and discuss with regard to the current work. Additional experiments as outlined by Reviewer #1 would be welcome but are not essential.
- 3) All other referee concerns pertaining to methodological details, clarifications and textual changes, should also be addressed.
- 4) Please revise the figure layout to meet our guidelines (maximum of 8 main figures and 10 Extended Data figures).
- 5) Finally please pay close attention to our guidelines on statistical and methodological reporting (listed below) as failure to do so may delay the reconsideration of the revised manuscript. In particular please provide:
 - a Supplementary Figure including unprocessed images of all gels/blots in the form of a multi-page pdf file. Please ensure that blots/gels are labeled and the sections presented in the figures are clearly indicated.
 - a Supplementary Table including all numerical source data in Excel format, with data for different figures provided as different sheets within a single Excel file. The file should include source data giving rise to graphical representations and statistical descriptions in the paper and for all instances where the figures present representative experiments of multiple independent repeats, the source data of all repeats should be provided.

We therefore invite you to take these points into account when revising the manuscript. In addition, when preparing the revision please:

- ensure that it conforms to our format instructions and publication policies (see below and <https://www.nature.com/nature/for-authors>).
- provide a point-by-point rebuttal to the full referee reports verbatim, as provided at the end of this letter.
- provide the completed Reporting Summary (found here <https://www.nature.com/documents/nr-reporting-summary.pdf>). This is essential for reconsideration of the manuscript and will be available to editors and referees in the event of peer review. For more information see <http://www.nature.com/authors/policies/availability.html> or contact me.

When submitting the revised version of your manuscript, please pay close attention to our <https://www.nature.com/nature-portfolio/editorial-policies/image-integrity> Digital Image Integrity Guidelines. and to the following points below:

- that unprocessed scans are clearly labelled and match the gels and western blots presented in figures.

- that control panels for gels and western blots are appropriately described as loading on sample processing controls
- all images in the paper are checked for duplication of panels and for splicing of gel lanes.

EXTENDED DATA FIGURES

Nature Cell Biology is committed to improving transparency in authorship. As part of our efforts in this direction, we are now requesting that all authors identified as 'corresponding author' on published papers create and link their Open Researcher and Contributor Identifier (ORCID) with their account on the Manuscript Tracking System (MTS), prior to acceptance. ORCID helps the scientific community achieve unambiguous attribution of all scholarly contributions. You can create and link your ORCID from the home page of the MTS by clicking on 'Modify my Springer Nature account'. For more information please visit www.springernature.com/orcid.

This journal strongly supports public availability of data. Please place the data used in your paper into a public data repository, or alternatively, present the data as Supplementary Information. If data can only be shared on request, please explain why in your Data Availability Statement, and also in the correspondence with your editor. Please note that for some data types, deposition in a public repository is mandatory - more information on our data deposition policies and available repositories appears below.

Link Redacted

We would like to receive the revision within four weeks. If submitted within this time period, reconsideration of the revised manuscript will not be affected by related studies published elsewhere, or accepted for publication in Nature Cell Biology in the meantime. We would be happy to consider a revision even after this timeframe, but in that case we will consider the published literature at the time of resubmission when assessing the file.

We hope that you will find our referees' comments, and editorial guidance helpful. Please do not hesitate to contact me if there is anything you would like to discuss.

Best wishes,

Angela Parrish

Angela R Parrish, PhD
Locum Senior Editor
Nature Cell Biology

Reviewers' Comments:

Reviewer #1 (Remarks to the Author):

Rudinskiy et al have revised and significantly strengthened the initial conclusions of their manuscript. The manuscript is presented in a way that is lucid and the data remain well-presented

However, since the initial submission of the manuscript, a PNAS paper from the Dikic lab has been published - simulating the effect of the IDR in co-operating with other membrane reshaping activities (e.g. RHD domain) in clustering FAM134B for ER fission. There is partial conceptual overlap with the current submission. Irrespective of this, I find it an omission that the published model is not substantially discussed in the current manuscript and it is not made clear precisely where and how the authors' model fits and/or differs.

Figures 2 and 3 of the re-organized paper now contain mostly confirmatory data showing that removing the IDR and LIR

together, or mutating the LIR, prevents ER-phagy upon overexpression of ER-phagy receptors. This is to be expected. The interesting new data is that which uncouples the function of the LIR from other functions of the IDR. We get to this at Figure 4 (which shows that ER fragmentation occurs independently of the LIR but requires the IDR for SEC62). I appreciate that these experiments are difficult. However, they are largely confined to SEC62 and their generalizability is unclear. The above-mentioned PNAS paper proposes FAM134B IDR function acting in concert with the RHD membrane reshaping domain in order drive ER fission. Thus, a comprehensive analysis of FAM134B is required in this new manuscript, at the least (the authors do show, unsurprisingly, that FAM134B RHD alone can't fragment the ER - the LIR has already been proposed to assist with this - but what about, more importantly, the role of the LIR versus the IDR?)

Reviewer #2 (Remarks to the Author):

The authors have significantly improved their manuscript by performing extensive revision experiments. However, all validations remain limited to overexpression systems. Although it is understandable that technical challenges make it difficult to verify these results outside of overexpression systems, as it stands, there remains uncertainty regarding whether the observed effects might be artifacts arising specifically from IDR overexpression. If the authors must rely on overexpression systems, they should include additional IDR controls. While the authors chose the IDR of REEP1 as a control, this choice is inappropriate, as REEP1's IDR is positively charged, whereas all receptor-derived IDRs investigated in this study are negatively charged. Additionally, the length of the REEP1 IDR (51 residues) is comparable to the shortest receptor IDR examined. The authors should test at least three non-receptor-derived IDRs that cover the full range of receptor IDRs in terms of charge distribution and amino acid length.

Reviewer #3 (Remarks to the Author):

I thank the authors for addressing comments in this work and performing additional experiments to further strengthen this already-insightful manuscript.

With the discussion on 'functional conservation on sequence divergency' for intrinsically-disordered regions (IDRs), it may be useful to include a reference to Lucia Chemes' recent work in 2022: "Conformational buffering underlies functional selection in intrinsically disordered protein regions" doi: 10.1038/s41594-022-00811-w. The current work on IDR modules of organellphagy receptors reminded me of how IDRs may not be sequence-conserved but still retain certain conservation of biophysical features that are important for function. Lucia's work reports a similar phenomenon where the "effective" length (not actual length) of disordered linkers is conserved for a tethering function in the E1A family.

A minor comment: on line 52, should be 'has two major consequences'

GUIDELINES FOR SUBMISSION OF NATURE CELL BIOLOGY ARTICLES

ARTICLE FORMAT

ABSTRACT – should not exceed 150 words and should be unreferenced. This paragraph is the most visible part of the paper and should briefly outline the background and rationale for the work, and accurately summarize the main results and conclusions. Key genes, proteins and organisms should be specified to ensure discoverability of the paper in online searches.

TEXT – the main text consists of the Introduction, Results, and Discussion sections and must not exceed 3500 words including the abstract. The Introduction should expand on the background relating to the work. The Results should be divided in subsections with subheadings, and should provide a concise and accurate description of the experimental findings. The Discussion should expand on the findings and their implications. All relevant primary literature should be cited, in particular when discussing the background and specific findings.

REFERENCES – are limited to a total of 70 in the main text and Methods combined. They must be numbered sequentially as they appear in the main text, tables and figure legends and Methods and must follow the precise style of Nature Cell Biology references. References only cited in the Methods should be numbered consecutively following the last reference cited in the main text. References only associated with Supplementary Information (e.g. in supplementary legends) do not count toward the total reference limit and do not need to be cited in numerical continuity with references in the main text. Only published papers can be cited, and each publication cited should be included in the numbered reference list, which should include the manuscript titles. Footnotes are not permitted.

Methods should be written concisely, but should contain all elements necessary to allow interpretation and replication of the results. As a guideline, Methods sections typically do not exceed 3,000 words. The Methods should be divided into subsections listing reagents and techniques. When citing previous methods, accurate references should be provided and any alterations should be noted. Information must be provided about: antibody dilutions, company names, catalogue numbers and clone numbers for monoclonal antibodies; sequences of RNAi and cDNA probes/primers or company names and catalogue numbers if reagents are commercial; cell line names, sources and information on cell line identity and authentication. Animal studies and experiments involving human subjects must be reported in detail, identifying the committees approving the protocols. For studies involving human subjects/samples, a statement must be included confirming that informed consent was obtained. Statistical analyses and information on the reproducibility of experimental results should be provided in a section titled "Statistics and Reproducibility".

All Nature Cell Biology manuscripts submitted on or after March 21 2016, must include a Data availability statement as a separate section after Methods but before references, under the heading "Data Availability". For Springer Nature policies on data availability see <http://www.nature.com/authors/policies/availability.html>; for more information on this particular policy see <http://www.nature.com/authors/policies/data/data-availability-statements-data-citations.pdf>. The Data availability statement should include:

- Accession codes for primary datasets (generated during the study under consideration and designated as "primary accessions") and secondary datasets (published datasets reanalysed during the study under consideration, designated as "referenced accessions"). For primary accessions data should be made public to coincide with publication of the manuscript. A list of data types for which submission to community-endorsed public repositories is mandated (including sequence, structure, microarray, deep sequencing data) can be found here <http://www.nature.com/authors/policies/availability.html#data>.
- Unique identifiers (accession codes, DOIs or other unique persistent identifier) and hyperlinks for datasets deposited in an approved repository, but for which data deposition is not mandated (see here for details <http://www.nature.com/sdata/data-policies/repositories>).
- At a minimum, please include a statement confirming that all relevant data are available from the authors, and/or are included with the manuscript (e.g. as source data or supplementary information), listing which data are included (e.g. by figure panels and data types) and mentioning any restrictions on availability.
- If a dataset has a Digital Object Identifier (DOI) as its unique identifier, we strongly encourage including this in the Reference list and citing the dataset in the Methods.

We recommend that you upload the step-by-step protocols used in this manuscript to [protocols.io](https://www.protocols.io). More details can be found at <https://www.protocols.io/help/publish-articles>.

DISPLAY ITEMS – main display items are limited to 6-8 main figures and/or main tables. For Supplementary Information see below.

FIGURES – Colour figure publication costs \$395 per colour figure. All panels of a multi-panel figure must be logically connected and arranged as they would appear in the final version. Unnecessary figures and figure panels should be avoided (e.g. data presented in small tables could be stated briefly in the text instead).

All imaging data should be accompanied by scale bars, which should be defined in the legend.

Cropped images of gels/blots are acceptable, but need to be accompanied by size markers, and to retain visible background signal within the linear range (i.e. should not be saturated). The boundaries of panels with low background have to be demarked with black lines. Splicing of panels should only be considered if unavoidable, and must be clearly marked on the figure, and noted in the legend with a statement on whether the samples were obtained and processed simultaneously. Quantitative comparisons between samples on different gels/blots are discouraged; if this is unavoidable, it has to be performed for samples derived from the same experiment with gels/blots were processed in parallel, which needs to be stated in the legend.

Regardless of format, all figures must be vector graphic compatible files, not supplied in a flattened raster/bitmap graphics format, but should be fully editable, allowing us to highlight/copy/paste all text and move individual parts of the figures (i.e. arrows, lines, x and y axes, graphs, tick marks, scale bars etc). The only parts of the figure that should be in pixel raster/bitmap format are photographic images or 3D rendered graphics/complex technical illustrations.

Unprocessed scans of all key data generated through electrophoretic separation techniques need to be presented in a supplementary figure that should be labeled and numbered as the final supplementary figure, and should be mentioned in every relevant figure legend. This figure does not count towards the total number of figures and is the only figure that can be displayed over multiple pages, but should be provided as a single file, in PDF or TIFF format. Data in this figure can be displayed in a relatively informal style, but size markers and the figures panels corresponding to the presented data must be indicated.

The total number of Supplementary Figures (not including the "unprocessed scans" Supplementary Figure) should not exceed the number of main display items (figures and/or tables (see our Guide to Authors and March 2012 editorial <http://www.nature.com/ncb/authors/submit/index.html#suppinfo>; <http://www.nature.com/ncb/journal/v14/n3/index.html#ed>). No restrictions apply to Supplementary Tables or Videos, but we advise authors to be selective in including supplemental data.

GUIDELINES FOR EXPERIMENTAL AND STATISTICAL REPORTING

REPORTING REQUIREMENTS – We ask authors to complete a Reporting Summary that collects information on experimental design and reagents. We hope this will aid in your evaluation of the paper. The Reporting Summary can be found here <https://www.nature.com/documents/nr-reporting-summary.pdf>. Please note that these forms are dynamic 'smart pdfs' and must therefore be downloaded and completed in Adobe Reader. We will then flatten them for ease of use. If you would like to reference the guidance text as you complete the template, please access these flattened versions at <http://www.nature.com/authors/policies/availability.html>.

Version 3:

Decision Letter:

Our ref: NCB-A54062C

22nd May 2025

Dear Dr. Molinari,

Thank you for submitting your revised manuscript "The intrinsically disordered regions of organellophagy receptors control organelle fragmentation, ER-phagy and mitophagy flux" (NCB-A54062C). It has now been seen by the original referees and

their comments are below. The reviewers find that the paper has improved in revision, and therefore we'll be happy in principle to publish it in Nature Cell Biology, pending minor revisions to satisfy the referees' final requests and to comply with our editorial and formatting guidelines.

Thank you again for your interest in Nature Cell Biology Please do not hesitate to contact me if you have any questions.

Sincerely,

Angela R Parrish, PhD
Locum Senior Editor
Nature Cell Biology

Reviewer #3 (Remarks to the Author):

In responding to Reviewer 2's request for additional IDR controls ("The authors should test at least three non-receptor-derived IDRs that cover the full range of receptor IDRs in terms of charge distribution and amino acid length"), there is merit and value to be gained. This was also one of Reviewer 3's requests originally: "The authors should use other IDR segments (from other proteins) of similar biophysical properties (and/or length) and see whether they are able to perform similar fragmentation function."

First, it is important to point out that the authors do have additional controls as they mention (e.g., ER membrane-bound Halo, mitochondria membrane-bound short IDR of 23 residues, Ext. Fig. 10 constructs of various IDR lengths for SEC62, and their positively-charged REEP1 protein). I would suggest the authors to compile a table that details the sequences of these constructs, their net charge, and length. If the authors want to use their existing controls in their rebuttal, I think it is important to justify how these controls address the reviewer's concern.

Reviewer 2 asked to examine IDRs from non-autophagic proteins with similar length and charge to ER-phagy IDRs. The authors' experiments where they test various lengths of SEC62 IDR on fragmentation (Ext. Fig. 10) does help address this question as the shortened IDRs act as different proteins. One additional way to address the reviewer's concern is to generate a few scrambled IDR constructs (take a sequence that does drive fragmentation and scramble the order of amino acids). If these IDRs are still able to drive fragmentation, it would strengthen the author's point of 'conservation of function without conservation of sequence'. If not, then it would suggest more nuances to the IDR sequences for follow-up work.

The work is impactful and the scrambled IDR experiment plus additional data table will further strengthen the points the authors are making.

Version 4:

Decision Letter:

Dear Dr Molinari,

I am pleased to inform you that your manuscript, "The intrinsically disordered regions of organellophagy receptors are interchangeable and control organelle fragmentation, ER-phagy and mitophagy flux", has now been accepted for publication in Nature Cell Biology.

Please note that *Nature Cell Biology* is a Transformative Journal (TJ). Authors may publish their research with us through the traditional subscription access route or make their paper immediately open access through payment of an article-processing charge (APC). Authors will not be required to make a final decision about access to their article until it has been accepted. <https://www.springernature.com/gp/open-research/transformative-journals> Find out more about Transformative Journals

Authors may need to take specific actions to achieve [compliance with funder and institutional open access mandates](https://www.springernature.com/gp/open-research/funding/policy-compliance-faqs). If your research is supported by a funder that requires immediate open access (e.g. according to [Plan S principles](https://www.springernature.com/gp/open-research/plan-s-compliance)) then you should select the gold OA route, and we will direct you to the compliant route where possible. For authors selecting the subscription publication route, the journal's standard licensing terms will need to be accepted, including [self-archiving policies](https://www.springernature.com/gp/open-research/policies/journal-policies). Those licensing terms will supersede any other terms that the author or any third party may assert apply to any version of the manuscript.

If you have not already done so, we strongly recommend that you upload the step-by-step protocols used in this manuscript to protocols.io (<https://protocols.io>), an open online resource that allows researchers to share their detailed experimental know-how. All uploaded protocols are made freely available and are assigned DOIs for ease of citation. Protocols and Nature Portfolio journal papers in which they are used can be linked to one another, and this link is clearly and prominently visible in the online versions of both. Authors who performed the specific experiments can act as primary authors for the Protocol as they will be best placed to share the methodology details, but the Corresponding Author of the present research paper should be included as one of the authors. By uploading your Protocols onto protocols.io, you are enabling researchers to more readily reproduce or adapt the methodology you use, as well as increasing the visibility of your protocols and papers. You can also establish a dedicated workspace to collect your lab Protocols. Further information can be found at <https://www.protocols.io/help/publish-articles>.

Nature Cell Biology encourages authors presenting evidence for cell, biological, molecular, and genetic interactions to consider communicating these findings using Biofactoid (<https://biofactoid.org/>). This tool helps users share a searchable representation of interactions (e.g. binding, gene expression, post-translational modification) between genes, gene products, or chemicals. Information added to Biofactoid, with author attribution, is shared on social media and public databases, such

as Pathway Commons, where it can be discovered and analyzed in the context of a large and growing corpus of knowledge.

With kind regards,

Angela R Parrish, PhD
Locum Senior Editor
Nature Cell Biology

** Visit the Springer Nature Editorial and Publishing website at http://editorial-jobs.springernature.com?utm_source=ejp_NCB_email&utm_medium=ejp_NCB_email&utm_campaign=ejp_NCB for more information about our career opportunities. If you have any questions please click [here](mailto:editorial.publishing.jobs@springernature.com).

Start of the Editor's letter

Your manuscript "Intrinsically disordered modules drive organelle fragmentation and autophagy", has now been seen by 3 referees, who are experts in ER-phagy (Referee #1); autophagy (Referee #2); and biomolecular condensate biology (Referee #3), and whose comments are pasted below. In light of their advice, we regret that we cannot offer to publish the study in Nature Cell Biology.

As you will see, although the reviewers found this work interesting, they raised serious concerns that question the strength of the data and of the novel conclusions that can be drawn at this stage. In particular, they shared concerns about:

i) the generality of the mechanism for other forms of organellophagy

In the original submission, we showed that the expression at the mitochondrial membrane of the IDR modules of three different ER-phagy receptors (FAM134B, SEC62, TEX264) triggers mitochondrial fragmentation and mitophagy. We now show that also the opposite is true (i.e., expression at the ER membrane of the IDR module of a mitophagy receptor triggers ER fragmentation and ER-phagy). These results strongly support the functional conservation of the IDR modules of organellophagy receptors;

ii) asked for more information about the features of the IDRs that are functionally involved
We now provide schematics and graphs that summarize the properties of organello-phagy receptors IDR modules with lengths ranging from the 47 and the 250 residues, their net negative cumulative charge at physiologic pH, their LIR placed at least 24 residues from the membrane and highlight their conservation in ER-phagy, mitophagy and Golgiphagy receptors in mammalian, yeast (Fig. 1), and plant cells.

We also include experiments showing the consequences of a progressive shortening of the IDR modules and of the reduction of the distance of the LIR from the membrane. These modifications eventually inhibits the capacity to fragment the homing organelle and to engage lipidated LC3 proteins, when the sizes fall below the length conserved in physiologically active organellophagy receptors.

Moreover, we add a control where the IDR module of REEP1, a membrane remodeling protein of the ER, which has length in the same range as organellophagy receptor's IDR modules but it is characterized by a net cumulative positive charge, fails to fragment the ER when expressed at the ER membrane or to fragment the mitochondria when expressed at the OMM and to deliver organelle portions within endolysosomes upon addition of a LIR;

iii) asked for more evidence in support of the mechanism

Molecular dynamic studies by the group of Dikic/Bhaskara led to propose that the membrane remodeling reticulon homology domain (RHD) module of the FAM134 family of ER-phagy receptors plays a crucial role in ER fragmentation.

What motivated our work is that this model can certainly not be generalized because all other organellophagy receptors are anchored at membranes with conventional transmembrane domains lacking the membrane remodeling function.

In a manuscript published in the October issue of PNAS, the Dikic/Bhaskara group now recognizes, based on new molecular dynamic studies, that the IDR module of FAM134B (associated with the FAM134B's RHD) eventually facilitates membrane budding (Intrinsically disordered region amplifies membrane remodelling to augment selective ER-phagy, PNAS, 2024). Their analyses show that the IDR modules collapse onto the membrane and induce positive membrane curvature. They write that the charge patterns underlying this behavior are conserved across other ER-phagy receptors, reports that IDRs alone are sufficient to sense curvature and that when combined with RHDs, they enhance membrane remodeling leading to faster budding, thereby amplifying RHD remodeling functions. This offers support to the mechanism that we originally proposed. The novelty of our work is that we show *in cellula* that the membrane anchoring modules of ER-phagy and mitophagy receptors are actually dispensable for organelle fragmentation, regardless of their capacity to remodel membranes. Rather, organelle fragmentation is promoted by surface exposure of IDR modules (being them derived from an ER-phagy or a mitophagy receptor). This is important because it explains how organellophagy receptors lacking RHDs (i.e., most ER-phagy receptors and all autophagy receptors at the limiting membrane of other organelles in mammals, yeast and plant cells), do work. In our submission, we also show that the mitochondrial fragmentation promoted by the expression of ER-phagy and mitophagy IDR modules at the OMM requires the intervention of the *fissionase* DRP1 (and of its GTPase activity). Most likely, a *fissionase* also intervenes to complete the ER fragmentation process promoted by the IDR modules of ER-phagy and mitophagy receptors. However, our tests failed to confirm an intervention of Atlastin proteins (Atlastins were likely candidates as their involvement in ER-phagy has been reported in a study by the Corn's lab, JCB 2018).

iii) and were concerned about the use of over-expression

More precisely, referee 2 comments that "On the other hand, this manuscript has serious shortcomings; it just looks at phenotypes when overexpressed and it is not clear how much IDR contributes to fragmentation of ER and mitochondria at physiological expression levels. It is essential to provide data on the role of IDRs at endogenous expression levels. All the provided data are obtained by overexpression. It is known that overexpression of proteins in mitochondria cause their fragmentation by stress. The obtained data shown here may simply indicate that overexpression of any IDRs on the surface of ER and mitochondria causes their fragmentation and may not indicate a physiological function for the IDR of the ER-phagy receptors. It is essential to study the role of IDRs at the physiological expression level."

In the new version of the manuscript, we examine a long list of controls that conclusively show that the phenotypes observed relates to the intrinsic membrane remodeling properties of the IDR modules and not to their overexpression possibly causing a stress-induced organelle fragmentation as observed by referee 2.

In fact, the membrane expression of various control polypeptides does not induce significant ER or mitochondria fragmentation (as checked by morphometric and by FRAP analyses). Amongst the control polypeptides used in our work there are the transmembrane modules of ER-phagy receptors (including the RHD), ER membrane-bound sfGFP, ER membrane-bound Halo, mitochondria membrane-bound sfGFP, mitochondria membrane-bound short IDR of 23 residues (i.e., shorter than the shortest IDR module conserved in organellophagy receptors (the IDR of FUNDC1, 47 residues) that has lost the capacity to fragment the mitochondria. In some cases, these polypeptides were expressed

at higher levels than the membrane tethered IDR modules, yet no organelle fragmentation was induced (these controls are shown in Figs. 3-9, Ext. Figs. 2-6, 10) and variations in mitochondrial size are quantified in Figs. 6l, 9b, Ext. Figs. 5, 10.

Importantly, to directly answer to the concern by referee 2 (The obtained data shown here may simply indicate that overexpression of any IDRs on the surface of ER and mitochondria causes their fragmentation), **we now report that the expression of the IDR module of the REEP1 protein tethered at the outer mitochondrial membrane does not fragment the mitochondria (nor does it fragment the ER, when expressed at the ER membrane).**

On a separate note, the request of “performing the experiments at endogenous/ physiologic expression levels” should take into consideration the fact that organellophagy receptors are “inactive” at steady state. Organellophagy receptors are activated by cues (e.g., nutrient deprivation, accumulation of misfolded proteins and others) that increase their overall (e.g., upon transcriptional induction, de-repression, ...) and/or their local level (via clustering and oligomerization) at organelle subdomains that must be cleared from cells. Some of the cues that trigger these processes (e.g., nutrient restriction) simultaneously activate multiple organellophagy receptors, others (e.g., expression of a misfolded protein in the ER lumen) may activate only ER-phagy receptors.

To assess the role of organellophagy modules *in cellula*, it is crucial to induce organellophagy by-passing the need of activation signals that would activate organelle fragmentation *per se*. Expression of full-length organellophagy receptors has been informative in dissecting their function and regulation (references in the text) and this experimental approach is also used in our work, implemented with the appropriate controls that exclude experimental artifacts as a consequence of protein overexpression.

Reviewers' comments:

Reviewer #1 (Remarks to the Author):

Rudinsky et al present an elegant dataset that probes the mechanism of action of ER-phagy receptors in fragmenting ER, prior to inclusion into autophagolysosomes. Their data suggest that the intrinsically-disordered regions of these receptors, the function of which was previously unknown, drive fragmentation (whereas transmembrane domains mainly drive localisation to particular subregions of this organelle, e.g. tubules versus sheets). LC3-binding sites embedded within the intrinsically-disordered regions are required only at a late step for inclusion into autophagosomes. Finally, the authors suggest that this mechanism is generalisable to other selective autophagy pathways and their cognate receptors. Thus, overall, these data suggest a new paradigm for how clearance of ER (and potentially other membrane bound organelles) is targeted for clearance. This is a novel and important finding as it provides a unifying mechanism for ER-phagy (and indeed, potentially, selective autophagy of other membrane bound organelles). The manuscript is well-written and referenced. However, the scope of the present study is slightly limited in regard of the latter-mentioned generalisability of this phenomenon to other organelles. Furthermore, some further controls/refinements may be required in a limited number of places to strengthen the data already presented (although these should not present great difficulty).

We thank the reviewer for the suggestions, and we address his/her concerns below:

MAJOR POINTS

The authors indicate that additional organelles, other than the ER, may depend upon IDR-containing receptors for membrane fragmentation during initial stages of autophagy. This is established using a chimeric approach using model targeting domains and studying action on mitochondria. Ideally, it should be established whether bona fide receptor(s) for organelles other than the ER utilise this mode of action.

In the original submission, we showed that the expression at the mitochondrial membrane of the IDR modules of three different ER-phagy receptors (FAM134B, SEC62, TEX264) triggers mitochondrial fragmentation and mitophagy (Figs. 5, 6, Ext. Figs. 3, 4, 5). In this submission we add that the mitophagy induced by surface expression of IDR modules of ER-phagy receptors is dependent on the DRP1 GTPase (Fig. 7, Ext. Fig. 6, 7). We now show that also the opposite is true (i.e., expression at the ER membrane of the IDR module of a mitophagy receptor triggers ER fragmentation and ER-phagy, Fig. 8). These results strongly support the functional conservation of the IDR modules of organellophagy receptors;

Figure 3 - Are IDRs that are not from ER-phagy receptors capable of driving ER-phagy? **We show that the net cumulative negative charge conserved amongst organellophagy receptors is important, because expression of IDR modules with similar length, but net cumulative positive charge (in our tests the IDR module of the ER protein REEP1) does not fragment ER when expressed at the ER membrane, nor does it fragment mitochondria when expressed at the mitochondrial membrane, nor delivers organelle portions within degradative endolysosomes (new Fig. 9).**

Are there any minimal/maximal lengths or other restrictions or criteria for the success of an IDR in driving membrane fragmentation? This information would be useful.

The new Fig. 1 shows the conservation of IDR modules between ER-phagy, mitophagy (and Golgiphagy, not studied here) receptors in mammalian, yeast (and plants, not shown in the figure). The organellophagy receptors display a cytoplasmic IDR module ranging from the 47 to the 250 residues, characterized by non conserved sequences with a net negative cumulative charge and a 4-7 residues LC3-interacting region (LIR). We show that the net cumulative negative charge conserved amongst organellophagy receptors is important, because expression of IDR modules with similar length, but net cumulative positive charge (the IDR module of REEP1, Fig. 9) does not fragment organelles, nor delivers them within degradative endolysosomes (new in this submission).

Moreover, we show that shortening the distance of the LIR from the membrane below the shortest distance found in the mitophagy receptor FUNDC1, negatively impacts on the capacity of an IDR module to engage LC3 proteins and trigger lysosomal delivery (Ext. Fig. 9). Also, shortening the IDR modules below the shortest found in organellophagy receptors (the IDR module of FUNDC1, 47 residues) has an impact on the capacity to induce the fragmentation of organelles (an IDR module of 66 residues can still promote mitochondrial fragmentation, a function that is lost when the length of the net negatively charged IDR module is shortened to 23 residues (Ext. Fig. 10)).

Our data also show that the mode of membrane anchoring of the IDR modules (e.g., via a membrane remodeling RHD domain, or via conventional transmembrane domains) does not affect the capacity of the IDR modules to promote ER or mitochondria fragmentation. In fact, the original membrane anchors (Fig. 2 for the ER-phagy receptors, Fig. 8 for the mitophagy receptor), can be replaced by unrelated transmembrane domains to anchor the IDR modules of the ER-phagy receptors at the ER (Figs. 3, 4, Ext. Figs. 2, 8, 9) or at the outer membrane of mitochondria (Figs. 5-7 Ext. Figs. 3-7, 10) and to anchor the IDR module of the mitophagy receptor at the outer membrane of mitochondria or at the ER (Fig. 8).

Altogether, consistent with functional conservation, the IDR modules of ER-phagy and mitophagy receptors are interchangeable, i.e., IDR modules of ER-phagy receptors expressed at the outer mitochondrial membrane (OMM) trigger mitochondrial fragmentation and mitophagy and (new in this submission) IDR modules of mitophagy receptors expressed at the ER membrane trigger ER fragmentation and ER-phagy. We consider this *conservation of function without conservation of sequence*, throughout organelles and species an important paradigm per se, that opens interesting fields of study on the evolution of IDR modules.

ADDITIONAL CONTROLS

Figure 1 - should ideally have some quantification of fragmentation.

Assessment of ER fragmentation is difficult due to the complex morphology of the compartment. In our study, ER fragmentation is now shown in EM micrographs (Figs. 3, 4) and is indirectly quantified by FRAP assay (Fig. 4).

Monitoring the fragmentation of mitochondria is less challenging as it can be monitored by light microscopy (Fig. 6, 7, 9, Ext. Figs. 5, 10), electron microscopy (Figs. 5, 6) and variations in mitochondria size can be measured (Figs. 6, 9, Ext. Figs. 5, 10).

Figure 1 – localisation of mutants versus full-length, wild-type should be addressed using the same tagging strategy (this could be done with colocalization by standard immunofluorescence) to properly enable comparison, in the same system, of the targeting specificity of the TM domain alone. This is recommended in case the additional IDR and any other sequence nuances the targeting (the authors should not rely solely on previous descriptions of the behaviour of the wild-type).

We have renamed this figure as Ext. Fig. 1. This figure confirms data published by other groups and our group, but we still consider it is worth showing. Robust data are available in the literature (references are given in the text, page 6) showing that individual ER-phagy receptors are located, and control lysosomal clearance, of distinct ER subdomains (FAM134B in ER tubuli and (as we also showed by immunoelectron microscopy in Kucinska et al Nature Comm 2023) SEC62 and TEX264 in ER sheets and outer nuclear membrane). We consider of some interest that the distinct sub-organellar localization is also observed upon cellular expression of the membrane anchoring domains of the ER-phagy receptors lacking the cytoplasmic IDR modules. The endogenous interactome of FAM134B_{RHD} reveals its distribution in subdomains hosting proteins of the reticulon and REEP families (as reported for the full-length FAM134B, Ext. Figs. 1f, 1g). SEC62_{TM} and TEX264_{TM} localize in ER sheets also containing subunits of the translocation machinery including the oligosaccharyltransferase and chaperones assisting protein folding (as reported for the full-length receptors, Ext. Figs. 1f, 1g). In the specific case of SEC62, we found particularly revealing that the clients of SEC62-dependent recov-ER-phagy that we identified in Fumagalli et al Nature Cell Biol 2016, Figure 8 (the PDI family proteins PDIA3, 4, 5, 6, Erp29 TMX3, TMX4, TXNDX5, several chaperone and members of the translocation machinery, i.e., the ER proteins identified by MS analyses of isolated endolysosomes upon recovery from ER stress) all show up in the fully different experimental set up shown in the Ext. Figs. 1f, 1g aiming at identifying by chemical cross-linking the proteins located in the ER subdomain hosting the TMD module of SEC62. This shows that, not surprisingly, the sub-organellar distribution of ER-phagy receptors is mainly determined by their mode-of anchoring at the membrane. To study this in more details goes beyond the scope of our manuscript.

Figure 3 – luminal tagging is performed to study the minimal role of IDR(s) in targeting to autophagolysosomes, by attaching IDRs to a bulky GFP tag. The latter may have unappreciated multimerization or scaffolding effects, particularly considering the ER-phagy receptors studied have few or no luminal regions. The GT fusion construct alone, with no IDR, should also be studied as a negative control to consolidate the interpretation of the effects of the IDR (I appreciate that IDR-LIR mutants are studied but do not think these are a sufficient control).

As suggested by the referee, we have now included the control constructs without the IDR module in key experiments (the GT₁₇ polypeptide in Figs. 3, 4, 8, Ext. Fig. 2). Moreover, the superfolder GFP used in our experiments does not multimerize and few experiments have been performed, with consistent results in three different cell lines, using the HaloTag to replace the GFP tag (we selected experiments shown in Figs. 2, 4f, 4g, for inclusion in our submission).

Also, several GFP-tagged constructs used as negative controls do not induce organelle fragmentation and/or lysosomal delivery (e.g., sfGFP-tagged transmembrane modules of

ER-phagy receptors (including the RHD), ER membrane-bound sfGFP, mitochondria membrane-bound sfGFP, sfGFP-tagged mitochondria membrane-bound short IDR of 23 residues (i.e., shorter than the shortest IDR module conserved in organellophagy receptors (the IDR of FUNDC1, 47 residues)). These controls show that these phenotypes are not induced by the tag (or by the overexpression of proteins at the membrane of organelles).

Figure 3 – it is mentioned that IDRs attached to a generic ER anchor drive lysosomal targeting of both ER sheets and tubules (Supp Figures 1 and 2). This is used as further evidence that the TM domains of receptors mediate specific sub-organellar targeting. Perhaps better evidence would be swapping the IDR of receptors targeting sheets or tubules while retaining the same TM domain.

These data are correct, but we decided to replace the old Suppl Fig. 1 and 2 with crucial controls suggested by the 3 referees that are more directly linked to the scope of the work presented here. Our opinion is that the data available in the literature and reporting on the localization of ER-phagy receptors (and other proteins) anchored at the ER membrane via RHD (e.g., reticulons, REEP proteins and ER-phagy receptors of the FAM134 family) and on the localization of other ER-phagy receptors such as TEX264 and SEC62 are solid (see above) and are recapitulated by identification of the endogenous polypeptides extracted from the environment occupied by the three membrane-anchoring modules of FAM134B, SEC62 and TEX264 (Ext. Figs. 1f, 1g). Deeper analyses of this issue would go beyond the scope of our work that focuses on the functional conservation of the ER-phagy and mitophagy IDR modules.

MINOR POINTS

Fig 1 – Volcano plots are illegible. **Thank you, we improved that.**

Line 61 – What is meant by “tout court”? **We have reformulated this sentence in the revised manuscript.**

Reviewer #2 (Remarks to the Author):

During organellophagy such as ER-phagy and mitophagy, it is necessary to fragment the organelles to the size smaller than that of autophagosomes for sequestration. Previously, RHD domains of some ER-phagy receptors were proposed to mediate this process; however, many ER-phagy/mitophagy receptors lack such domains and the mechanisms that cause organelle fragmentation during organellophagy remained largely unknown. In this manuscript, the authors studied the regions in ER-phagy receptors responsible for ER fragmentation and ER-phagy by overexpression and found that IDR, but not RHD, of ER-phagy receptors is sufficient for fragmentation of ER and ER-phagy. The authors then showed that LC3-binding LIR in the IDRs is necessary for ER-phagy, but not for ER fragmentation. Moreover, the authors showed that the transplantation of ER-phagy receptors IDRs at the mitochondrial membrane induced mitochondrial fragmentation and mitophagy. Based on these data, the authors concluded that membrane-exposed IDRs promote organelle fragmentation and organellophagy.

The mechanism of organellophagy is one of the hottest topics in the field of autophagy and the proposed concept of IDR-mediated fragmentation of organelles during organellophagy is novel and attractive. The data are straightforward and consistent, and the logic of the manuscript is easy to follow. On the other hand, this manuscript has serious shortcomings; it just looks at phenotypes when overexpressed and it is not clear how much IDR contributes to fragmentation of ER and mitochondria at physiological expression levels. It is essential to provide data on the role of IDRs at endogenous expression levels.

We thank the reviewer for the comments/suggestions, and we address his/her concerns below:

Major comments

1) All the provided data are obtained by overexpression. It is known that overexpression of proteins in mitochondria cause their fragmentation by stress. The obtained data shown here may simply indicate that overexpression of any IDRs on the surface of ER and mitochondria causes their fragmentation and may not indicate a physiological function for the IDR of the ER-phagy receptors. It is essential to study the role of IDRs at the physiological expression level.

We thank the referee for having raised this important issue. In the new version of the manuscript, we examine a long list of controls that conclusively show that the phenotypes observed relates to the intrinsic membrane remodeling properties of the IDR modules and not to their overexpression possibly causing a stress-induced organelle fragmentation. As an important note, during revision of our manuscript, the Dikic/Bhaskara group reported the results of their latest molecular dynamics simulations eventually proving that the IDR module of FAM134B has membrane remodeling capacity (PNAS 2024 and see below).

In the new submission, we show that the membrane expression of various control polypeptides does not induce significant ER or mitochondria fragmentation (as checked by morphometric and by FRAP analyses). Amongst the control polypeptides used in our work there are the transmembrane modules of ER-phagy receptors (including the RHD module of FAM134B), ER membrane-bound sfGFP, ER membrane-bound Halo, mitochondria membrane-bound sfGFP, mitochondria membrane-bound short IDR of 23 residues (i.e.,

shorter than the shortest IDR module conserved in organellophagy receptors (the IDR of FUNDC1, 47 residues) that has lost the capacity to fragment the mitochondria. In some cases, these polypeptides were expressed at higher levels than the membrane tethered IDR modules, yet no organelle fragmentation was induced (these controls are shown in Figs. 3-9, Ext. Figs. 2-6, 10) and variations in mitochondrial size are quantified in Figs. 6l, 9b, Ext. Figs. 5, 10.

Importantly, to directly answer the concern raised by the referee (The obtained data shown here may simply indicate that overexpression of any IDRs on the surface of ER and mitochondria causes their fragmentation), **we now report that the expression of the IDR module of the REEP1 protein tethered at the outer mitochondrial membrane does not fragment the mitochondria (nor does it fragment the ER, when expressed at the ER membrane).**

On a separate note, the request of “performing the experiments at endogenous/physiologic expression levels” should take into consideration the fact that organellophagy receptors are “inactive” at steady state. Organellophagy receptors are activated by cues (e.g., nutrient deprivation, accumulation of misfolded proteins and others) that increase their overall (e.g., upon transcriptional induction, de-repression, ...) and/or their local level (via clustering and oligomerization) at organelle subdomains that must be cleared from cells. Some of the cues that trigger these processes (e.g., nutrient restriction) simultaneously activate multiple organellophagy receptors, others (e.g., expression of a misfolded protein in the ER lumen) may activate only ER-phagy receptors.

To assess the role of organellophagy modules *in cellula*, it is crucial to induce organellophagy by-passing the need of activation signals that would activate organelle fragmentation *per se*. Expression of full-length organellophagy receptors has been informative in dissecting their function and regulation (references in the text) and this experimental approach is also used in our work, implemented with the appropriate controls that exclude experimental artifacts as a consequence of protein overexpression. We sincerely hope that the referee considers the control experiments performed in this new submission as sufficient to prove that the phenotypes observed are not an artifact of protein overexpression but duly report on the intrinsic and conserved properties of organellophagy IDR modules.

2) It is necessary to perform control experiments. Use some IDRs derived from non-autophagic proteins (with length and charge similar to ER-phagy IDRs) for the same experiments and study their effect on ER-phagy, ER fragmentation, and mitochondrial fragmentation.

In the new submission we added several control experiments and analyzed the impact on organelle fragmentation and organellophagy of the expression of the non-autophagy proteins REEP1 (please also refer to the answer above). The IDR module of REEP1 fails to induce mitochondrial fragmentation and mitophagy or ER-phagy (even if expressed at higher levels than IDR modules of organellophagy receptors, new Figure 9).

3) ER-phagy activity was only examined by fluorescence microscopy. It is essential that another additional assay be performed to robustly verify ER-phagy activity (e.g., HaloTag assay to monitor degradation of ER components (PMID 35938926)).

We thank the referee for this suggestion. In the new version of the manuscript, we measure both the ER-phagy (Figs. 2, 3) and the mitophagy activities (Figs. 5) with the Halo assay cited by the referee that has originally be developed in our lab (Rudinskiy et al., MBoC, 2022; Yim et al., Elife 2022; Rudinskiy et al., PLOS ONE 2023, Rudinskiy et al., Traffic 2024, Mizushima, Curr Op Cell Biol 2025). This biochemical assay was performed with new reporters appositely designed for localization at the ER membrane (HT₁₇) or at the OMM (HMAVS_{TM}) and confirmed the induction of ER-phagy by the expression of full length ER-phagy receptors (Fig. 2n), by the expression of ER-tethered IDR modules (if competent for LC3 engagement, Fig. 3i), and by the expression of IDR modules tethered at the OMM (if competent for LC3 engagement, Fig. 5i). These results fully support the data obtained with the imaging experiments of fluorescence and electron microscopy.

Reviewer #3 (Remarks to the Author):

This study focuses on the role of intrinsically-disordered regions (IDRs) within ER-phagy receptor proteins and the authors find that these regions control membrane fragmentation. The authors determine this through complementary light microscopy and electron microscopy approaches with a set of three ER-phagy receptor proteins in MEF cells. These three proteins include FAM134, SEC62, and TEX264. Through several interesting experiments using bafilomycin-inhibited or ATG7 knockout MEF cells and chimeric constructs, the authors demonstrate that the IDR segments of these ER-phagy receptor proteins drive membrane fragmentation. The LC3 interacting region (LIR) within these IDRs drive localization of ER or mitochondria delivery to lysosomes. The RHD (membrane-tethering domains) of these receptors determine sub-compartment distribution, as shown with chemical crosslinking and LC/MS proteomics analysis. Importantly, using engineered mitochondria-directing constructs, the authors also find that the IDRs of ER-phagy receptors can be used to program mitochondria membranes for mitochondria fragmentation. The insight of this story reveals a new function of IDRs in controlling membrane fragmentation; this is further buttressed by another preprint from a different group on a similar topic (preprint is mentioned by the authors in the discussion). The authors also employ additional IDR mutants containing the LC3 binding segment to determine the length of IDR required for coupled fragmentation-endolysosomal delivery. In general, the data appear to be well-presented, but the story is incomplete. Several questions abound regarding the IDRs as little is presented on these regions, yet they are the focus of this story. Little is presented regarding the mechanism by which these IDRs function to fragment membranes. Additionally, the study requires deposition of certain data.

We thank the reviewer for their insightful comments that we address below:

To explain the origins of membrane fragmentation by the IDR segments of these proteins (FAM134, SEC62, and TEX264), work is required to compare these IDR sequences biophysically. What is unique about these amino acid sequences that drive fragmentation? The authors mention possible hypotheses within the text, and these should be tested. The authors should use other IDR segments (from other proteins) of similar biophysical properties (and/or length) and see whether they are able to perform similar fragmentation function.

Molecular dynamic studies by the group of Dikic/Bhaskara led to propose that the membrane remodeling reticulon homology domain (RHD) module of the FAM134 family of ER-phagy receptors plays a crucial role in ER fragmentation.

What motivated our work is that this model can certainly not be generalized because all other organellophagy receptors are anchored at membranes with conventional transmembrane domains lacking the membrane remodeling function.

In a manuscript published during the revision of our work, in the October issue of PNAS, the Dikic/Bhaskara group now recognizes, based on new molecular dynamic simulations that the IDR module of FAM134B (associated with the FAM134B's RHD) eventually facilitates membrane budding (Intrinsically disordered region amplifies membrane remodelling to augment selective ER-phagy, PNAS 2024). Their analyses show that the IDR modules collapse onto the membrane and induce positive membrane curvature. They

write that the charge patterns underlying this behavior are conserved across other ER-phagy receptors, reports that IDRs alone are sufficient to sense curvature and that when combined with RHDs, they enhance membrane remodeling leading to faster budding, thereby amplifying RHD remodeling functions. This offers (*in silico*) support to the mechanism that we originally proposed.

The novelty of our work is that we show *in cellula* that the membrane anchoring modules of ER-phagy and mitophagy receptors are actually dispensable for organelle fragmentation, regardless of their capacity to remodel membranes. Rather, organelle fragmentation is promoted by surface exposure of IDR modules (being them derived from an ER-phagy or a mitophagy receptor). This is important because it explains how organellophagy receptors lacking RHDs (i.e., most ER-phagy receptors and all autophagy receptors at the limiting membrane of other organelles in mammals, yeast and plant cells), do work. In our submission, we also show that the mitochondrial fragmentation promoted by expression of ER-phagy and mitophagy IDR modules at the OMM requires the intervention of the *fissionase* DRP1 (and of its GTPase activity). We predict that a *fissionase* also intervenes to complete the ER fragmentation process promoted by the IDR modules of ER-phagy and mitophagy receptors. However, our tests failed to confirm an intervention of Atlastin proteins (Atlastins were likely candidates as their involvement in ER-phagy has been reported in a study by the Corn's lab, JCB 2018).

In this new submission, we provide schematics and graphs that summarize the properties of organello-phagy receptors IDR modules with lengths ranging from the 47 and the 250 residues, their net negative cumulative charge at physiologic pH, their LIR placed at least 24 residues from the membrane and highlight their conservation in ER-phagy, mitophagy and Golgiphagy receptors in mammalian, yeast, (and plant cells, not shown in the schematics). We also include experiments showing the consequences of a progressive shortening of the IDR modules and of the reduction of the distance of the LIR from the membrane. These modifications eventually inhibits the capacity to fragment the homing organelle and to engage lipidated LC3 proteins when the sizes fall below the length conserved in physiologically active organellophagy receptors.

Moreover, we add a control where the IDR module of REEP1, a membrane remodeling protein of the ER, which has length in the same range as organellophagy receptor's IDR modules, but it is characterized by a net cumulative positive charge, fails to fragment the ER when expressed at the ER membrane or to fragment the mitochondria when expressed at the OMM and to deliver them within endolysosomes upon addition of a LIR.

Are there specific mutations that are disease-linked to some of these IDR segments in ER-phagy receptor proteins? If so, the authors should try a mutation or two to make these results very impactful for the community.

There is no mutation we are aware of, which is linked to a disease. It must be written that the sequences of the various IDR modules are not conserved (the net cumulative negative charge is conserved and also relevant are a minimal length around the 40 residues and a minimal distance of the LIR of about 24 residues from the membrane, as confirmed in our tests, see above).

We think it is important to express to this referee as well that we consider this *conservation of function without conservation of sequence*, throughout organelles and species (note that also Golgiphagy receptors, not investigated in this study display a

cytoplasmic IDR module with the same features), an important paradigm per se, that opens interesting fields of study on the evolution of IDR modules.

Strangely, Figure S5 is presented within the discussion but this should be part of the results. The type of work presented in Figure S5 is important to better understand the function of the IDR region.

This is well taken. The results of the experiments where we assess the consequences of a progressive shortening of the IDR modules (new, Ext. Fig. 10) and of the reduction of the distance of the LIR from the membrane (Ext. Fig. 9) are now presented in the results section.

Are there any concerns with overexpressing these constructs into cells? The authors do not provide overall cell images (for example in Figure 3a,b,d,e,g,h) to highlight differences in cells with and without transfected constructs. Furthermore, one of these constructs should be introduced at endogenous expression levels to observe whether membrane fragmentation occurs.

Please also see referee 2, comment 1.

We now provide whole cell images and analyze non-transfected and transfected cells in the same coverslips where mitochondrial fragmentation is compared in Insets 1 (non-transfected cells) vs. Insets 2 (transfected cells) (see for example Figs. 6, 7, 9, Ext. Figs. 5, 10). In other figures, mock transfections are shown as negative control (Figs. 2, , 3i, 4f, 4g, 5, 6, 8i, 9c, Ext. Figs. 1b, 3, 4, 6.

In the new version of the manuscript, we examine a long list of controls that conclusively show that the phenotypes observed relates to the intrinsic membrane remodeling properties of the IDR modules and not to their overexpression possibly causing a stress-induced organelle fragmentation.

In fact, the membrane expression of various control polypeptides does not induce significant ER or mitochondria fragmentation (as checked by morphometric and by FRAP analyses). Amongst the control polypeptides used in our work there are the transmembrane modules of ER-phagy receptors (including the RHD), ER membrane-bound sfGFP, ER membrane-bound Halo, ER membrane-bound REEP1' IDR, mitochondria membrane-bound REEP1, mitochondria membrane-bound sfGFP, mitochondria membrane-bound short IDR of 23 residues (i.e., shorter than the shortest IDR module conserved in organellophagy receptors (the IDR of FUNDC1, 47 residues) that has lost the capacity to fragment the mitochondria. In some cases, these polypeptides were expressed at higher levels than the membrane tethered IDR modules, yet no organelle fragmentation was induced (these controls are shown in Figs. 3-9, Ext. Figs. 2-6, 10) and variations in mitochondrial size are quantified in Figs. 6l, 9b, Ext. Figs. 5, 10.

On a separate note, the request of “performing the experiments at endogenous/ physiologic expression levels” should take into consideration the fact that organellophagy receptors are “inactive” at steady state. Organellophagy receptors are activated by cues (e.g., nutrient deprivation, accumulation of misfolded proteins and others) that increase their overall (e.g., upon transcriptional induction, de-repression, ...) and/or their local level (via clustering and oligomerization) at organelle subdomains that must be cleared from cells. Some of the cues that trigger these processes (e.g., nutrient restriction)

simultaneously activate multiple organellophagy receptors, others (e.g., expression of a misfolded protein in the ER lumen) may activate only ER-phagy receptors. To assess the role of organellophagy modules *in cellula*, it is crucial to induce organellophagy by-passing the need of activation signals that would activate organelle fragmentation *per se*. Expression of full-length organellophagy receptors has been informative in dissecting their function and regulation (references in the text) and this experimental approach is also used in our work, implemented with the appropriate controls that exclude experimental artifacts as a consequence of protein overexpression.

The AP-LC/MS data presented must be deposited in a proteomics database for sharing – I did not see any information regarding this throughout the manuscript.

We have now deposited the proteomics data in the open database, the access is described in the Methods section. The data can be accessed here:

<https://www.ebi.ac.uk/pride/login>

Username: reviewer_pxd060519@ebi.ac.uk

Password: jkA8yrbIXHnp

Akin to what is shown in Fig S5, gels are required to showcase expression levels of transfected constructs as well as to show the size of transfected constructs in cells for data in Figures 1-4.

This is now shown in Figs. 2, 3, 5, 8, 9, Ext. Figs. 3, 9.

Are the results presented for MEF cells representative in other cell types? This would be important to support the main claim of the paper. Some data are presented in HEK293 cells (only Figure S4).

The results for IDR-induced fragmentation and organello-phagy are reproducible in different cell types, as shown for HEK293 (Ext. Fig. 5 and Ext. Fig. 3 for mitochondrial fragmentation and mitophagy), in MEF cells (e.g., Ext. Fig. 10 for mitochondrial fragmentation and Fig. 5 and, Ext. Fig. 6 for mitophagy). Similarly, IDR-induced ER-phagy is also reproducible in various cell types (HEK293 in Ext. Fig. 3, and MEF in Figs. 3 and Ext. Fig. 2). MEF cells are primarily used for quantitative microscopy analyses with Lysoquant as they have large endolysosomes that can be quantified with high confidence and in a high-throughput manner. Moreover, we have reproduced our findings that IDR modules trigger delivery of ER and mitochondrial portions to ELs in Hela cells, that we attach here (Ext. Fig. X) but do not include in the manuscript.

Extended Figure X

Extended Figure X: CLSM analyses of the delivery of ER portions to degradative endolysosomes in HeLa cells. **(a)** CLSM images of delivery of the ER marker CNX (red) to LAMP1-positive EL (LAMP1, cyan) in Mock-transfected HeLa cells treated with 50 nm BafA1. Scale bars: 10 μ m **(b)** Same as (a) for cells expressing GT₁₇ **(c)** Same as (b) for GT₁₇-FAM134B_{IDR}. **(d)** Same as (b) for GT₁₇-SEC62_{IDR}. **(e)** Same as (b) for GT₁₇-TEX264_{IDR}. **(f-h)** Same as (c-e) for the corresponding LIR mutants. **(i-p)** Same as (a-h) for the delivery of the mitochondria marker TOMM20 (red) to LAMP1-positive EL (LAMP1, cyan) upon expression of the transplantation of the IDR modules onto OMM.

Was there a reason for switching between GFP and HaloTag between Figure 3 and Figure S3?

We use Halo-tagged and sfGFP-tagged constructs interchangeably, with consistent results. A switch between sfGFP and Halo-tags is also important to confirm that the phenotypes are not caused by the tag.

Reviewers' comments:

Reviewer #1 (Remarks to the Author):

Rudinsky et al present an elegant dataset that probes the mechanism of action of ER-phagy receptors in fragmenting ER, prior to inclusion into autophagolysosomes. Their data suggest that the intrinsically-disordered regions of these receptors, the function of which was previously unknown, drive fragmentation (whereas transmembrane domains mainly drive localisation to particular subregions of this organelle, e.g. tubules versus sheets). LC3-binding sites embedded within the intrinsically-disordered regions are required only at a late step for inclusion into autophagosomes. Finally, the authors suggest that this mechanism is generalisable to other selective autophagy pathways and their cognate receptors. Thus, overall, these data suggest a new paradigm for how clearance of ER (and potentially other membrane bound organelles) is targeted for clearance. This is a novel and important finding as it provides a unifying mechanism for ER-phagy (and indeed, potentially, selective autophagy of other membrane bound organelles). The manuscript is well-written and referenced. However, the scope of the present study is slightly limited in regard of the latter-mentioned generalisability of this phenomenon to other organelles. Furthermore, some further controls/refinements may be required in a limited number of places to strengthen the data already presented (although these should not present great difficulty).

We thank the reviewer for the suggestions, and we address his/her concerns below:

MAJOR POINTS

The authors indicate that additional organelles, other than the ER, may depend upon IDR-containing receptors for membrane fragmentation during initial stages of autophagy. This is established using a chimeric approach using model targeting domains and studying action on mitochondria. Ideally, it should be established whether bona fide receptor(s) for organelles other than the ER utilise this mode of action.

In the original submission, we showed that the expression at the mitochondrial membrane of the IDR modules of three different ER-phagy receptors (FAM134B, SEC62, TEX264) triggers mitochondrial fragmentation and mitophagy (Figs. 5, 6, Ext. Figs. 3, 4, 5). In this submission we add that the mitophagy induced by surface expression of IDR modules of ER-phagy receptors is dependent on the DRP1 GTPase (Fig. 7, Ext. Fig. 6, 7). We now show that also the opposite is true (i.e., expression at the ER membrane of the IDR module of a mitophagy receptor triggers ER fragmentation and ER-phagy, Fig. 8). These results strongly support the functional conservation of the IDR modules of organellophagy receptors;

Figure 3 - Are IDRs that are not from ER-phagy receptors capable of driving ER-phagy? **We show that the net cumulative negative charge conserved amongst organellophagy receptors is important, because expression of IDR modules with similar length, but net cumulative positive charge (in our tests the IDR module of the ER protein REEP1) does not fragment ER when expressed at the ER membrane, nor does it fragment mitochondria when expressed at the mitochondrial membrane, nor delivers organelle portions within degradative endolysosomes (new Fig. 9).**

Are there any minimal/maximal lengths or other restrictions or criteria for the success of an IDR in driving membrane fragmentation? This information would be useful.

The new Fig. 1 shows the conservation of IDR modules between ER-phagy, mitophagy (and Golgiphagy, not studied here) receptors in mammalian, yeast (and plants, not shown in the figure). The organellophagy receptors display a cytoplasmic IDR module ranging from the 47 to the 250 residues, characterized by non conserved sequences with a net negative cumulative charge and a 4-7 residues LC3-interacting region (LIR). We show that the net cumulative negative charge conserved amongst organellophagy receptors is important, because expression of IDR modules with similar length, but net cumulative positive charge (the IDR module of REEP1, Fig. 9) does not fragment organelles, nor delivers them within degradative endolysosomes (new in this submission).

Moreover, we show that shortening the distance of the LIR from the membrane below the shortest distance found in the mitophagy receptor FUNDC1, negatively impacts on the capacity of an IDR module to engage LC3 proteins and trigger lysosomal delivery (Ext. Fig. 9). Also, shortening the IDR modules below the shortest found in organellophagy receptors (the IDR module of FUNDC1, 47 residues) has an impact on the capacity to induce the fragmentation of organelles (an IDR module of 66 residues can still promote mitochondrial fragmentation, a function that is lost when the length of the net negatively charged IDR module is shortened to 23 residues (Ext. Fig. 10)).

Our data also show that the mode of membrane anchoring of the IDR modules (e.g., via a membrane remodeling RHD domain, or via conventional transmembrane domains) does not affect the capacity of the IDR modules to promote ER or mitochondria fragmentation. In fact, the original membrane anchors (Fig. 2 for the ER-phagy receptors, Fig. 8 for the mitophagy receptor), can be replaced by unrelated transmembrane domains to anchor the IDR modules of the ER-phagy receptors at the ER (Figs. 3, 4, Ext. Figs. 2, 8, 9) or at the outer membrane of mitochondria (Figs. 5-7 Ext. Figs. 3-7, 10) and to anchor the IDR module of the mitophagy receptor at the outer membrane of mitochondria or at the ER (Fig. 8).

Altogether, consistent with functional conservation, the IDR modules of ER-phagy and mitophagy receptors are interchangeable, i.e., IDR modules of ER-phagy receptors expressed at the outer mitochondrial membrane (OMM) trigger mitochondrial fragmentation and mitophagy and (new in this submission) IDR modules of mitophagy receptors expressed at the ER membrane trigger ER fragmentation and ER-phagy. We consider this *conservation of function without conservation of sequence*, throughout organelles and species an important paradigm per se, that opens interesting fields of study on the evolution of IDR modules.

ADDITIONAL CONTROLS

Figure 1 - should ideally have some quantification of fragmentation.

Assessment of ER fragmentation is difficult due to the complex morphology of the compartment. In our study, ER fragmentation is now shown in EM micrographs (Figs. 3, 4) and is indirectly quantified by FRAP assay (Fig. 4).

Monitoring the fragmentation of mitochondria is less challenging as it can be monitored by light microscopy (Fig. 6, 7, 9, Ext. Figs. 5, 10), electron microscopy (Figs. 5, 6) and variations in mitochondria size can be measured (Figs. 6, 9, Ext. Figs. 5, 10).

Figure 1 – localisation of mutants versus full-length, wild-type should be addressed using the same tagging strategy (this could be done with colocalization by standard immunofluorescence) to properly enable comparison, in the same system, of the targeting specificity of the TM domain alone. This is recommended in case the additional IDR and any other sequence nuances the targeting (the authors should not rely solely on previous descriptions of the behaviour of the wild-type).

We have renamed this figure as Ext. Fig. 1. This figure confirms data published by other groups and our group, but we still consider it is worth showing. Robust data are available in the literature (references are given in the text, page 6) showing that individual ER-phagy receptors are located, and control lysosomal clearance, of distinct ER subdomains (FAM134B in ER tubuli and (as we also showed by immunoelectron microscopy in Kucinska et al Nature Comm 2023) SEC62 and TEX264 in ER sheets and outer nuclear membrane). We consider of some interest that the distinct sub-organellar localization is also observed upon cellular expression of the membrane anchoring domains of the ER-phagy receptors lacking the cytoplasmic IDR modules. The endogenous interactome of FAM134B_{RHD} reveals its distribution in subdomains hosting proteins of the reticulon and REEP families (as reported for the full-length FAM134B, Ext. Figs. 1f, 1g). SEC62_{TM} and TEX264_{TM} localize in ER sheets also containing subunits of the translocation machinery including the oligosaccharyltransferase and chaperones assisting protein folding (as reported for the full-length receptors, Ext. Figs. 1f, 1g). In the specific case of SEC62, we found particularly revealing that the clients of SEC62-dependent recov-ER-phagy that we identified in Fumagalli et al Nature Cell Biol 2016, Figure 8 (the PDI family proteins PDIA3, 4, 5, 6, Erp29, TMX3, TMX4, TXNDC5, several chaperone and members of the translocation machinery, i.e., the ER proteins identified by MS analyses of isolated endolysosomes upon recovery from ER stress) all show up in the fully different experimental set up shown in the Ext. Figs. 1f, 1g aiming at identifying by chemical cross-linking the proteins located in the ER subdomain hosting the TMD module of SEC62. This shows that, not surprisingly, the sub-organellar distribution of ER-phagy receptors is mainly determined by their mode-of anchoring at the membrane. To study this in more details goes beyond the scope of our manuscript.

Figure 3 – luminal tagging is performed to study the minimal role of IDR(s) in targeting to autophagolysosomes, by attaching IDRs to a bulky GFP tag. The latter may have unappreciated multimerization or scaffolding effects, particularly considering the ER-phagy receptors studied have few or no luminal regions. The GT fusion construct alone, with no IDR, should also be studied as a negative control to consolidate the interpretation of the effects of the IDR (I appreciate that IDR-LIR mutants are studied but do not think these are a sufficient control).

As suggested by the referee, we have now included the control constructs without the IDR module in key experiments (the GT₁₇ polypeptide in Figs. 3, 4, 8, Ext. Fig. 2). Moreover, the superfolder GFP used in our experiments does not multimerize and few experiments have been performed, with consistent results in three different cell lines, using the HaloTag to replace the GFP tag (we selected experiments shown in Figs. 2, 4f, 4g, for inclusion in our submission).

Also, several GFP-tagged constructs used as negative controls do not induce organelle fragmentation and/or lysosomal delivery (e.g., sfGFP-tagged transmembrane modules of

ER-phagy receptors (including the RHD), ER membrane-bound sfGFP, mitochondria membrane-bound sfGFP, sfGFP-tagged mitochondria membrane-bound short IDR of 23 residues (i.e., shorter than the shortest IDR module conserved in organellophagy receptors (the IDR of FUNDC1, 47 residues)). These controls show that these phenotypes are not induced by the tag (or by the overexpression of proteins at the membrane of organelles).

Figure 3 – it is mentioned that IDRs attached to a generic ER anchor drive lysosomal targeting of both ER sheets and tubules (Supp Figures 1 and 2). This is used as further evidence that the TM domains of receptors mediate specific sub-organellar targeting. Perhaps better evidence would be swapping the IDR of receptors targeting sheets or tubules while retaining the same TM domain.

These data are correct, but we decided to replace the old Suppl Fig. 1 and 2 with crucial controls suggested by the 3 referees that are more directly linked to the scope of the work presented here. Our opinion is that the data available in the literature and reporting on the localization of ER-phagy receptors (and other proteins) anchored at the ER membrane via RHD (e.g., reticulons, REEP proteins and ER-phagy receptors of the FAM134 family) and on the localization of other ER-phagy receptors such as TEX264 and SEC62 are solid (see above) and are recapitulated by identification of the endogenous polypeptides extracted from the environment occupied by the three membrane-anchoring modules of FAM134B, SEC62 and TEX264 (Ext. Figs. 1f, 1g). Deeper analyses of this issue would go beyond the scope of our work that focuses on the functional conservation of the ER-phagy and mitophagy IDR modules.

MINOR POINTS

Fig 1 – Volcano plots are illegible. **Thank you, we improved that.**

Line 61 – What is meant by “tout court”? **We have reformulated this sentence in the revised manuscript.**

Reviewer #2 (Remarks to the Author):

During organellophagy such as ER-phagy and mitophagy, it is necessary to fragment the organelles to the size smaller than that of autophagosomes for sequestration. Previously, RHD domains of some ER-phagy receptors were proposed to mediate this process; however, many ER-phagy/mitophagy receptors lack such domains and the mechanisms that cause organelle fragmentation during organellophagy remained largely unknown. In this manuscript, the authors studied the regions in ER-phagy receptors responsible for ER fragmentation and ER-phagy by overexpression and found that IDR, but not RHD, of ER-phagy receptors is sufficient for fragmentation of ER and ER-phagy. The authors then showed that LC3-binding LIR in the IDRs is necessary for ER-phagy, but not for ER fragmentation. Moreover, the authors showed that the transplantation of ER-phagy receptors IDRs at the mitochondrial membrane induced mitochondrial fragmentation and mitophagy. Based on these data, the authors concluded that membrane-exposed IDRs promote organelle fragmentation and organellophagy.

The mechanism of organellophagy is one of the hottest topics in the field of autophagy and the proposed concept of IDR-mediated fragmentation of organelles during organellophagy is novel and attractive. The data are straightforward and consistent, and the logic of the manuscript is easy to follow. On the other hand, this manuscript has serious shortcomings; it just looks at phenotypes when overexpressed and it is not clear how much IDR contributes to fragmentation of ER and mitochondria at physiological expression levels. It is essential to provide data on the role of IDRs at endogenous expression levels.

We thank the reviewer for the comments/suggestions, and we address his/her concerns below:

Major comments

1) All the provided data are obtained by overexpression. It is known that overexpression of proteins in mitochondria cause their fragmentation by stress. The obtained data shown here may simply indicate that overexpression of any IDRs on the surface of ER and mitochondria causes their fragmentation and may not indicate a physiological function for the IDR of the ER-phagy receptors. It is essential to study the role of IDRs at the physiological expression level.

We thank the referee for having raised this important issue. In the new version of the manuscript, we examine a long list of controls that conclusively show that the phenotypes observed relates to the intrinsic membrane remodeling properties of the IDR modules and not to their overexpression possibly causing a stress-induced organelle fragmentation. As an important note, during revision of our manuscript, the Dikic/Bhaskara group reported the results of their latest molecular dynamics simulations eventually proving that the IDR module of FAM134B has membrane remodeling capacity (PNAS 2024 and see below).

In the new submission, we show that the membrane expression of various control polypeptides does not induce significant ER or mitochondria fragmentation (as checked by morphometric and by FRAP analyses). Amongst the control polypeptides used in our work there are the transmembrane modules of ER-phagy receptors (including the RHD module of FAM134B), ER membrane-bound sfGFP, ER membrane-bound Halo, mitochondria membrane-bound sfGFP, mitochondria membrane-bound short IDR of 23 residues (i.e.,

shorter than the shortest IDR module conserved in organellophagy receptors (the IDR of FUNDC1, 47 residues) that has lost the capacity to fragment the mitochondria. In some cases, these polypeptides were expressed at higher levels than the membrane tethered IDR modules, yet no organelle fragmentation was induced (these controls are shown in Figs. 3-9, Ext. Figs. 2-6, 10) and variations in mitochondrial size are quantified in Figs. 6l, 9b, Ext. Figs. 5, 10.

Importantly, to directly answer the concern raised by the referee (The obtained data shown here may simply indicate that overexpression of any IDRs on the surface of ER and mitochondria causes their fragmentation), **we now report that the expression of the IDR module of the REEP1 protein tethered at the outer mitochondrial membrane does not fragment the mitochondria (nor does it fragment the ER, when expressed at the ER membrane).**

On a separate note, the request of “performing the experiments at endogenous/ physiologic expression levels” should take into consideration the fact that organellophagy receptors are “inactive” at steady state. Organellophagy receptors are activated by cues (e.g., nutrient deprivation, accumulation of misfolded proteins and others) that increase their overall (e.g., upon transcriptional induction, de-repression, ...) and/or their local level (via clustering and oligomerization) at organelle subdomains that must be cleared from cells. Some of the cues that trigger these processes (e.g., nutrient restriction) simultaneously activate multiple organellophagy receptors, others (e.g., expression of a misfolded protein in the ER lumen) may activate only ER-phagy receptors.

To assess the role of organellophagy modules *in cellula*, it is crucial to induce organellophagy by-passing the need of activation signals that would activate organelle fragmentation *per se*. Expression of full-length organellophagy receptors has been informative in dissecting their function and regulation (references in the text) and this experimental approach is also used in our work, implemented with the appropriate controls that exclude experimental artifacts as a consequence of protein overexpression. We sincerely hope that the referee considers the control experiments performed in this new submission as sufficient to prove that the phenotypes observed are not an artifact of protein overexpression but duly report on the intrinsic and conserved properties of organellophagy IDR modules.

2) It is necessary to perform control experiments. Use some IDRs derived from non-autophagic proteins (with length and charge similar to ER-phagy IDRs) for the same experiments and study their effect on ER-phagy, ER fragmentation, and mitochondrial fragmentation.

In the new submission we added several control experiments and analyzed the impact on organelle fragmentation and organellophagy of the expression of the non-autophagy proteins REEP1 (please also refer to the answer above). The IDR module of REEP1 fails to induce mitochondrial fragmentation and mitophagy or ER-phagy (even if expressed at higher levels than IDR modules of organellophagy receptors, new Figure 9).

3) ER-phagy activity was only examined by fluorescence microscopy. It is essential that another additional assay be performed to robustly verify ER-phagy activity (e.g., HaloTag assay to monitor degradation of ER components (PMID 35938926)).

We thank the referee for this suggestion. In the new version of the manuscript, we measure both the ER-phagy (Figs. 2, 3) and the mitophagy activities (Figs. 5) with the Halo assay cited by the referee that has originally be developed in our lab (Rudinskiy et al., MBoC, 2022; Yim et al., Elife 2022; Rudinskiy et al., PLOS ONE 2023, Rudinskiy et al., Traffic 2024, Mizushima, Curr Op Cell Biol 2025). This biochemical assay was performed with new reporters appositely designed for localization at the ER membrane (HT₁₇) or at the OMM (HMAVS_{TM}) and confirmed the induction of ER-phagy by the expression of full length ER-phagy receptors (Fig. 2n), by the expression of ER-tethered IDR modules (if competent for LC3 engagement, Fig. 3i), and by the expression of IDR modules tethered at the OMM (if competent for LC3 engagement, Fig. 5i). These results fully support the data obtained with the imaging experiments of fluorescence and electron microscopy.

Reviewer #3 (Remarks to the Author):

This study focuses on the role of intrinsically-disordered regions (IDRs) within ER-phagy receptor proteins and the authors find that these regions control membrane fragmentation. The authors determine this through complementary light microscopy and electron microscopy approaches with a set of three ER-phagy receptor proteins in MEF cells. These three proteins include FAM134, SEC62, and TEX264. Through several interesting experiments using bafilomycin-inhibited or ATG7 knockout MEF cells and chimeric constructs, the authors demonstrate that the IDR segments of these ER-phagy receptor proteins drive membrane fragmentation. The LC3 interacting region (LIR) within these IDRs drive localization of ER or mitochondria delivery to lysosomes. The RHD (membrane-tethering domains) of these receptors determine sub-compartment distribution, as shown with chemical crosslinking and LC/MS proteomics analysis. Importantly, using engineered mitochondria-directing constructs, the authors also find that the IDRs of ER-phagy receptors can be used to program mitochondria membranes for mitochondria fragmentation. The insight of this story reveals a new function of IDRs in controlling membrane fragmentation; this is further buttressed by another preprint from a different group on a similar topic (preprint is mentioned by the authors in the discussion). The authors also employ additional IDR mutants containing the LC3 binding segment to determine the length of IDR required for coupled fragmentation-endolysosomal delivery. In general, the data appear to be well-presented, but the story is incomplete. Several questions abound regarding the IDRs as little is presented on these regions, yet they are the focus of this story. Little is presented regarding the mechanism by which these IDRs function to fragment membranes. Additionally, the study requires deposition of certain data.

We thank the reviewer for their insightful comments that we address below:

To explain the origins of membrane fragmentation by the IDR segments of these proteins (FAM134, SEC62, and TEX264), work is required to compare these IDR sequences biophysically. What is unique about these amino acid sequences that drive fragmentation? The authors mention possible hypotheses within the text, and these should be tested. The authors should use other IDR segments (from other proteins) of similar biophysical properties (and/or length) and see whether they are able to perform similar fragmentation function.

Molecular dynamic studies by the group of Dikic/Bhaskara led to propose that the membrane remodeling reticulon homology domain (RHD) module of the FAM134 family of ER-phagy receptors plays a crucial role in ER fragmentation.

What motivated our work is that this model can certainly not be generalized because all other organellophagy receptors are anchored at membranes with conventional transmembrane domains lacking the membrane remodeling function.

In a manuscript published during the revision of our work, in the October issue of PNAS, the Dikic/Bhaskara group now recognizes, based on new molecular dynamic simulations that the IDR module of FAM134B (associated with the FAM134B's RHD) eventually facilitates membrane budding (Intrinsically disordered region amplifies membrane remodelling to augment selective ER-phagy, PNAS 2024). Their analyses show that the IDR modules collapse onto the membrane and induce positive membrane curvature. They

write that the charge patterns underlying this behavior are conserved across other ER-phagy receptors, reports that IDRs alone are sufficient to sense curvature and that when combined with RHDs, they enhance membrane remodeling leading to faster budding, thereby amplifying RHD remodeling functions. This offers (*in silico*) support to the mechanism that we originally proposed.

The novelty of our work is that we show *in cellula* that the membrane anchoring modules of ER-phagy and mitophagy receptors are actually dispensable for organelle fragmentation, regardless of their capacity to remodel membranes. Rather, organelle fragmentation is promoted by surface exposure of IDR modules (being them derived from an ER-phagy or a mitophagy receptor). This is important because it explains how organellophagy receptors lacking RHDs (i.e., most ER-phagy receptors and all autophagy receptors at the limiting membrane of other organelles in mammals, yeast and plant cells), do work. In our submission, we also show that the mitochondrial fragmentation promoted by expression of ER-phagy and mitophagy IDR modules at the OMM requires the intervention of the *fissionase* DRP1 (and of its GTPase activity). We predict that a *fissionase* also intervenes to complete the ER fragmentation process promoted by the IDR modules of ER-phagy and mitophagy receptors. However, our tests failed to confirm an intervention of Atlastin proteins (Atlastins were likely candidates as their involvement in ER-phagy has been reported in a study by the Corn's lab, JCB 2018).

In this new submission, we provide schematics and graphs that summarize the properties of organello-phagy receptors IDR modules with lengths ranging from the 47 and the 250 residues, their net negative cumulative charge at physiologic pH, their LIR placed at least 24 residues from the membrane and highlight their conservation in ER-phagy, mitophagy and Golgiphagy receptors in mammalian, yeast, (and plant cells, not shown in the schematics). We also include experiments showing the consequences of a progressive shortening of the IDR modules and of the reduction of the distance of the LIR from the membrane. These modifications eventually inhibits the capacity to fragment the homing organelle and to engage lipidated LC3 proteins when the sizes fall below the length conserved in physiologically active organellophagy receptors.

Moreover, we add a control where the IDR module of REEP1, a membrane remodeling protein of the ER, which has length in the same range as organellophagy receptor's IDR modules, but it is characterized by a net cumulative positive charge, fails to fragment the ER when expressed at the ER membrane or to fragment the mitochondria when expressed at the OMM and to deliver them within endolysosomes upon addition of a LIR.

Are there specific mutations that are disease-linked to some of these IDR segments in ER-phagy receptor proteins? If so, the authors should try a mutation or two to make these results very impactful for the community.

There is no mutation we are aware of, which is linked to a disease. It must be written that the sequences of the various IDR modules are not conserved (the net cumulative negative charge is conserved and also relevant are a minimal length around the 40 residues and a minimal distance of the LIR of about 24 residues from the membrane, as confirmed in our tests, see above).

We think it is important to express to this referee as well that we consider this *conservation of function without conservation of sequence*, throughout organelles and species (note that also Golgiphagy receptors, not investigated in this study display a

cytoplasmic IDR module with the same features), an important paradigm per se, that opens interesting fields of study on the evolution of IDR modules.

Strangely, Figure S5 is presented within the discussion but this should be part of the results. The type of work presented in Figure S5 is important to better understand the function of the IDR region.

This is well taken. The results of the experiments where we assess the consequences of a progressive shortening of the IDR modules (new, Ext. Fig. 10) and of the reduction of the distance of the LIR from the membrane (Ext. Fig. 9) are now presented in the results section.

Are there any concerns with overexpressing these constructs into cells? The authors do not provide overall cell images (for example in Figure 3a,b,d,e,g,h) to highlight differences in cells with and without transfected constructs. Furthermore, one of these constructs should be introduced at endogenous expression levels to observe whether membrane fragmentation occurs.

Please also see referee 2, comment 1.

We now provide whole cell images and analyze non-transfected and transfected cells in the same coverslips where mitochondrial fragmentation is compared in Insets 1 (non-transfected cells) vs. Insets 2 (transfected cells) (see for example Figs. 6, 7, 9, Ext. Figs. 5, 10). In other figures, mock transfections are shown as negative control (Figs. 2, , 3i, 4f, 4g, 5, 6, 8i, 9c, Ext. Figs. 1b, 3, 4, 6.

In the new version of the manuscript, we examine a long list of controls that conclusively show that the phenotypes observed relates to the intrinsic membrane remodeling properties of the IDR modules and not to their overexpression possibly causing a stress-induced organelle fragmentation.

In fact, the membrane expression of various control polypeptides does not induce significant ER or mitochondria fragmentation (as checked by morphometric and by FRAP analyses). Amongst the control polypeptides used in our work there are the transmembrane modules of ER-phagy receptors (including the RHD), ER membrane-bound sfGFP, ER membrane-bound Halo, ER membrane-bound REEP1' IDR, mitochondria membrane-bound REEP1, mitochondria membrane-bound sfGFP, mitochondria membrane-bound short IDR of 23 residues (i.e., shorter than the shortest IDR module conserved in organellophagy receptors (the IDR of FUNDC1, 47 residues) that has lost the capacity to fragment the mitochondria. In some cases, these polypeptides were expressed at higher levels than the membrane tethered IDR modules, yet no organelle fragmentation was induced (these controls are shown in Figs. 3-9, Ext. Figs. 2-6, 10) and variations in mitochondrial size are quantified in Figs. 6l, 9b, Ext. Figs. 5, 10.

On a separate note, the request of “performing the experiments at endogenous/ physiologic expression levels” should take into consideration the fact that organellophagy receptors are “inactive” at steady state. Organellophagy receptors are activated by cues (e.g., nutrient deprivation, accumulation of misfolded proteins and others) that increase their overall (e.g., upon transcriptional induction, de-repression, ...) and/or their local level (via clustering and oligomerization) at organelle subdomains that must be cleared from cells. Some of the cues that trigger these processes (e.g., nutrient restriction)

simultaneously activate multiple organellophagy receptors, others (e.g., expression of a misfolded protein in the ER lumen) may activate only ER-phagy receptors. To assess the role of organellophagy modules *in cellula*, it is crucial to induce organellophagy by-passing the need of activation signals that would activate organelle fragmentation *per se*. Expression of full-length organellophagy receptors has been informative in dissecting their function and regulation (references in the text) and this experimental approach is also used in our work, implemented with the appropriate controls that exclude experimental artifacts as a consequence of protein overexpression.

The AP-LC/MS data presented must be deposited in a proteomics database for sharing – I did not see any information regarding this throughout the manuscript.

We have now deposited the proteomics data in the open database, the access is described in the Methods section. The data can be accessed here:

<https://www.ebi.ac.uk/pride/login>

Username: reviewer_pxd060519@ebi.ac.uk

Password: jkA8yrbIXHnp

Akin to what is shown in Fig S5, gels are required to showcase expression levels of transfected constructs as well as to show the size of transfected constructs in cells for data in Figures 1-4.

This is now shown in Figs. 2, 3, 5, 8, 9, Ext. Figs. 3, 9.

Are the results presented for MEF cells representative in other cell types? This would be important to support the main claim of the paper. Some data are presented in HEK293 cells (only Figure S4).

The results for IDR-induced fragmentation and organello-phagy are reproducible in different cell types, as shown for HEK293 (Ext. Fig. 5 and Ext. Fig. 3 for mitochondrial fragmentation and mitophagy), in MEF cells (e.g., Ext. Fig. 10 for mitochondrial fragmentation and Fig. 5 and, Ext. Fig. 6 for mitophagy). Similarly, IDR-induced ER-phagy is also reproducible in various cell types (HEK293 in Ext. Fig. 3, and MEF in Figs. 3 and Ext. Fig. 2). MEF cells are primarily used for quantitative microscopy analyses with Lysoquant as they have large endolysosomes that can be quantified with high confidence and in a high-throughput manner. Moreover, we have reproduced our findings that IDR modules trigger delivery of ER and mitochondrial portions to ELs in HeLa cells, that we attach here (Ext. Fig. X) but do not include in the manuscript.

Extended Figure X: CLSM analyses of the delivery of ER portions to degradative endolysosomes in HeLa cells. **(a)** CLSM images of delivery of the ER marker CNX (red) to LAMP1-positive EL (LAMP1, cyan) in Mock-transfected HeLa cells treated with 50 nm BafA1. Scale bars: 10 μ m **(b)** Same as (a) for cells expressing GT₁₇ **(c)** Same as (b) for GT₁₇-FAM134B_{IDR}. **(d)** Same as (b) for GT₁₇-SEC62_{IDR}. **(e)** Same as (b) for GT₁₇-TEX264_{IDR}. **(f-h)** Same as (c-e) for the corresponding LIR mutants. **(i-p)** Same as (a-h) for the delivery of the mitochondria marker TOMM20 (red) to LAMP1-positive EL (LAMP1, cyan) upon expression of the transplantation of the IDR modules onto OMM.

Was there a reason for switching between GFP and HaloTag between Figure 3 and Figure S3?

We use Halo-tagged and sfGFP-tagged constructs interchangeably, with consistent results. A switch between sfGFP and Halo-tags is also important to confirm that the phenotypes are not caused by the tag.

Point-by-point rebuttal to the referee comments

Reviewer #1 (Remarks to the Author):

Rudinskiy et al have revised and significantly strengthened the initial conclusions of their manuscript. The manuscript is presented in a way that is lucid and the data remain well-presented

However, since the initial submission of the manuscript, a PNAS paper from the Dikic lab has been published - simulating the effect of the IDR in co-operating with other membrane reshaping activities (e.g. RHD domain) in clustering FAM134B for ER fission. There is partial conceptual overlap with the current submission. Irrespective of this, I find it an omission that the published model is not substantially discussed in the current manuscript and it is not made clear precisely where and how the authors' model fits and/or differs.

We have moved the citation of this paper from the Introduction to the Discussion section (lines 747-751). In the discussion, we also write more explicitly that the published model ascribes to the membrane remodeling function of the RHDs that anchor the ER-phagy receptors of the FAM134 family a crucial role in driving ER fragmentation. Our data clearly show that the RHDs (not conserved amongst organellophagy receptors) are dispensable for organelle fragmentation. IDR modules (conserved in organellophagy receptors) are required and sufficient.

Figures 2 and 3 of the re-organized paper now contain mostly confirmatory data showing that removing the IDR and LIR together, or mutating the LIR, prevents ER-phagy upon overexpression of ER-phagy receptors. This is to be expected.

These data are important because current models imply that only expression of ER-phagy receptors with membrane remodeling capacity (i.e., tethered at the ER membrane via RHD modules) triggers ER fragmentation and ER-phagy (as explicitly mentioned in highly cited reviews, e.g., 10.1038/s41418-019-0444-0 and elsewhere) and that ER-phagy receptors lacking RHD must form complexes with ER-phagy receptors with RHD to function (e.g., 10.15252/embj.2019102608). Figures 2 and 3 (with Ext. Fig. 1) prove that expression of ER-phagy receptors, regardless of the capacity of their membrane tethering modules to remodel membranes do elicit ER fragmentation and ER-phagy.

The interesting new data is that which uncouples the function of the LIR from other functions of the IDR. We get to this at Figure 4 (which shows that ER fragmentation occurs independently of the LIR but requires the IDR for SEC62). I appreciate that these experiments are difficult. However, they are largely confined to SEC62 and their generalizability is unclear. The above-mentioned PNAS paper proposes FAM134B IDR function acting in concert with the RHD membrane reshaping domain in order drive ER fission. Thus, a comprehensive analysis of FAM134B is required in this new manuscript, at the least (the authors do show, unsurprisingly, that FAM134B RHD alone can't fragment the ER - the LIR has already been proposed to assist with this - but what about,

more importantly, the role of the LIR versus the IDR?)

We appreciate the decision of the editors that the experiments suggested by referee 1 are not essential. As a note, it is true that the electron microscopy images in Fig. 4 only reports on the capacity of the IDR of SEC62 to fragment the ER and on the dispensability of the engagement of LC3 lipidation for organelle fragmentation. However, these findings are confirmed also for the IDRs and the LIR motifs of FAM134B and of TEX264 in Figures 3, 5-6, and Ext. figs. 2-7.

Reviewer #2 (Remarks to the Author):

The authors have significantly improved their manuscript by performing extensive revision experiments. However, all validations remain limited to overexpression systems. Although it is understandable that technical challenges make it difficult to verify these results outside of overexpression systems, as it stands, there remains uncertainty regarding whether the observed effects might be artifacts arising specifically from IDR overexpression. If the authors must rely on overexpression systems, they should include additional IDR controls. While the authors chose the IDR of REEP1 as a control, this choice is inappropriate, as REEP1's IDR is positively charged, whereas all receptor-derived IDRs investigated in this study are negatively charged. Additionally, the length of the REEP1 IDR (51 residues) is comparable to the shortest receptor IDR examined. The authors should test at least three non-receptor-derived IDRs that cover the full range of receptor IDRs in terms of charge distribution and amino acid length.

We consider the selection of controls presented in the new version of our manuscript, including the REEP1's IDR, fully appropriate to directly address the important concerns raised by referee 2 when evaluating our original submission (Mai 2024, "The obtained data shown here may simply indicate that overexpression of any IDRs on the surface of ER and mitochondria causes their fragmentation and may not indicate a physiological function for the IDR of the ER-phagy receptors.")

Our new control experiments show that expression of the REEP1's IDR module at the ER or mitochondrial limiting membrane fails to trigger organelle fragmentation/organellophagy (Ext. Fig. 9). This module has a length within the limits of the conserved IDR modules of organellophagy receptors BUT has a net cumulative POSITIVE charge.

Expression of IDR modules that maintain the net cumulative negative charge BUT have progressively shorter length also fails to trigger organelle fragmentation/organellophagy when the length is shorter than the shortest IDR modules of organellophagy receptors (Ext. Fig. 10).

All in all, we are convinced that these controls are appropriate (and important), because they convincingly demonstrate that organelle fragmentation is not "caused by surface expression of any IDR module" (the concern of referee 2 expressed in his/her comment).

In the new version of our manuscript, we include additional tests showing that expression at the ER or mitochondrial limiting membrane of 9 different polypeptides fails to induce a significant organelle fragmentation. These 9 control polypeptides include the transmembrane modules of 3 different ER-phagy receptors expressed at the ER membrane, sfGFP anchored at the ER membrane, sfGFP with a short 23aa-long negatively charged IDR anchored at the mitochondrial membrane, HaloTag anchored at the ER membrane.

Our opinion is that the experiments suggested by referee 2 “The authors should test at least three non-receptor-derived IDRs that cover the full range of receptor IDRs in terms of charge distribution and amino acid length.” goes beyond the scope of our investigation and (independent of the positive or negative outcome) will not add to our observations on “functional conservation on sequence divergency” that characterizes cytoplasmic IDR modules of organellophagy receptors across species to regulate organelle integrity and turnover.

Reviewer #3 (Remarks to the Author):

I thank the authors for addressing comments in this work and performing additional experiments to further strengthen this already-insightful manuscript.

With the discussion on 'functional conservation on sequence divergency' for intrinsically-disordered regions (IDRs), it may be useful to include a reference to Lucia Chemes' recent work in 2022: "Conformational buffering underlies functional selection in intrinsically disordered protein regions" doi: 10.1038/s41594-022-00811-w. The current work on IDR modules of organellophagy receptors reminded me of how IDRs may not be sequence-conserved but still retain certain conservation of biophysical features that are important for function. Lucia's work reports a similar phenomenon where the "effective" length (not actual length) of disordered linkers is conserved for a tethering function in the E1A family.

We thank the referee for this comment. We have now added, at the end of the discussion, a note on the interesting paper by Lucia Chemes highlighting a model, where IDRs characterized by different protein primary structure may retain biophysical features that are important for their function. This is in full agreement with the *functional conservation on sequence divergency* that we report for organellophagy receptors.